# Synergistic roles of aquaporin 5 and intra- and extracellular carbonic anhydrases in promoting $CO_2$ diffusion across the *Xenopus* oocyte plasma membrane

Deng-Ke Wang[1] , Fraser J. Moss[1] and Walter F. Boron[1,2,3]

[1]*Department of Physiology and Biophysics, Case Western Reserve University School of Medicine, Cleveland, Ohio, USA*
[2]*Department of Medicine, Case Western Reserve University School of Medicine, Cleveland, Ohio, USA*
[3]*Department of Biochemistry, Case Western Reserve University School of Medicine, Cleveland, Ohio, USA*

Handling Editors: Peying Fong & Brian Delisle

The peer review history is available in the Supporting Information section of this article (https://doi.org/10.1113/JP289145#support-information-section).

**Deng-Ke Wang** earned his BSc in biological sciences from Henan University, Kaifeng, China, in 2009, and his PhD in biological chemistry from Huazhong University of Science and Technology (HUST), Wuhan, China, in 2015. He completed postdoctoral training in the Boron lab in the Department of Physiology and Biophysics at Case Western Reserve University. His research focuses on acid–base regulation mediated by $CO_2$ and $NH_3$ channels.

This article was first published as a preprint. Wang D-K, Moss FJ, Boron WF. 2025. Synergistic roles of Aquaporin 5 and Intra- and Extracellular Carbonic Anhydrases In facilitating $CO_2$ Diffusion across the Xenopus Oocyte Plasma Membrane. bioRxiv. https://doi.org/10.1101/2025.04.08.647833

**Abstract figure legend** Fick's law: $J_{CO_2} = P_{M^*,CO_2} \cdot ([CO_2]_{os} - [CO_2]_{is})$

$\pm$hAQP5     $\pm$bCA     $\pm$hCA II

Transmembrane $CO_2$ gradient

Fick's law governs the diffusion of $CO_2$ into a cell. The entry of $CO_2$ depletes $CO_2$ at the outer surface but raises it at the inner surface, both tending to decrease the transmembrane gradient. A, we show that bovine carbonic anhydrase (bCA) in the extracellular fluid and human CA II in the intracellular fluid increase the flux by enhancing the gradient, whereas aquaporin 5 (AQP5) increases the flux by increasing membrane $CO_2$ permeability ($P_{M^*,CO_2}$). The three effects are synergistic. B, we assess the $CO_2$ flux from pH changes on the cell surface ($\Delta pH_S$) or intracellular fluid ($dpH_i/dt$). However it is critical that one use pH changes that are 'trans' to the added CA to avoid confusion from the 'cis-side' consumption or production of $H^+$. AQP5 is 'inter' with respect to both pH measurements and is thus not subject to 'cis' errors.

**Abstract**   $CO_2$ diffusion across plasma membranes depends on both membrane $CO_2$ permeability ($P_{M,CO_2}$) and the transmembrane $CO_2$ concentration gradient ($\Delta[CO_2]$) – Fick's law. Human aquaporin-5 (hAQP5) enhances $CO_2$ diffusion by increasing $P_{M,CO_2}$, whereas carbonic anhydrases (CAs) do so by enhancing $CO_2$ consumption/production and thus $\Delta[CO_2]$. Here we systematically assess functional interactions among a gas channel and intra-/extracellular CAs. On Day 1 we inject *Xenopus* oocytes with cRNA encoding hAQP5 (control: $H_2O$). On Day 4 we inject hCA II protein in 'Tris' buffer (control: 'Tris'). We assess $CO_2$ fluxes by introducing extracellular 1.5% $CO_2$/10 mM $HCO_3^-$ and using microelectrodes to measure (1) the maximal increase of extracellular surface pH ($\Delta pH_S$), (2) the maximal rate of $pH_S$ relaxation $(dpH_S/dt)_{Max}$ and (3) the maximal rate of intracellular pH decrease $(dpH_i/dt)_{Max}$. By itself hCA II minimally increases $\Delta pH_S$ – measured on the side of the membrane opposite to the added cytosolic CA ($CA_i$) – even at our highest doses (100 ng/oocyte). However hAQP5 alone triples $\Delta pH_S$, an effect further doubled by increasing hCA II. By itself bovine erythrocyte CA (bCA) in the extracellular fluid doubles $(dpH_i/dt)_{Max}$ magnitude – measured on the side of the membrane opposite to the added extracellular CA ($CA_o$) – an effect further doubled by hAQP5. Our pH measurements (1) confirm synergy between $CA_o$ and $CA_i$; establish synergy between hAQP5 and both (2) $CA_o$ and (3) $CA_i$; and (4) show that the ability of $CA_i$ to enhance $\Delta pH_S$ is a useful tool for assessing the $CO_2$ permeability of membrane proteins (e.g. hAQP5).

(Received 23 April 2025; accepted after revision 11 September 2025; first published online 5 December 2025)

**Corresponding author** W. F. Boron: Department of Physiology and Biophysics, Case Western Reserve University School of Medicine, 10900 Euclid Ave., Robbins Building, E524 Cleveland, OH 44106-4970, USA.    Email: walter.boron@case.edu

### Key points

- According to Fick's law transmembrane $CO_2$ flux ($J_{CO_2}$) is the product of membrane permeability ($P_{M,CO_2}$) and transmembrane concentration gradient ($\Delta[CO_2]$): $J_{CO_2} = P_{M,CO_2} \times \Delta[CO_2]$. Previous work separately showed that (1) human aquaporin-5 (hAQP5) enhances $P_{M,CO_2}$, and (2) intracellular and (3) extracellular carbonic anhydrases (CAs) enhance ($\Delta[CO_2]$) by consuming accumulated or replenishing lost $CO_2$. We now examine interactions among #1–#3.
- We assess $CO_2$ fluxes – produced by addition/removal of extracellular $CO_2$/$HCO_3^-$ – using microelectrodes to monitor extracellular surface pH ($pH_S$) and intracellular pH ($pH_i$) of *Xenopus* oocytes heterologously expressing hAQP5, injected with human CA II (hCA II), and/or exposed to extracellular bovine CA (bCA).
- Enhancing effects on $CO_2$ fluxes are synergistic among hAQP5, hCA II and bCA, any of which can become rate-limiting, depending on the status of the other two.
- $CO_2$/$HCO_3^-$ addition transiently increases $pH_S$ ($\Delta pH_S$), hCA II augments $\Delta pH_S$ ($\Delta\Delta pH_S$) and hAQP5 enhances $\Delta\Delta pH_S$ ($\Delta\Delta\Delta pH_S$) – a novel tool to assess potential $CO_2$ channels.

## Introduction

Aquaporins (AQPs) are integral membrane proteins named for their ability to conduct $H_2O$ across biological membranes. The AQPs occur ubiquitously across all kingdoms of life (Ishibashi et al., 2017; King et al., 2004; Maurel et al., 2015; Ni et al., 2017). To date the cDNAs encoding 13 human AQPs have been cloned and characterized (for review see Li & Wang, 2017). One can divide them into three groups, based on whether they can conduct glycerol in addition to $H_2O$. The first group – the classical aquaporins – comprises AQP0, 1, 2, 4, 5, 6 and 8. Although some classical aquaporins are also permeable to urea (AQP6 and AQP8) or ions (AQP6), all of them have a high permeability to $H_2O$. The second group – the aquaglyceroporins – includes AQP3, 7, 9 and 10 (Heymann & Engel, 1999; Ishibashi et al., 1997; Michalek, 2016; Moss et al., 2020; Zhang et al., 2019). Besides $H_2O$ they can mediate the diffusion of glycerol, urea and other non-volatile solutes. Finally the third group consists of AQP11 and AQP12, which show the lowest homology to the other aquaporins and have uncertain functions (Calvanese et al., 2013; Ishibashi, 2009).

In addition to conducting $H_2O$, glycerol, urea, and other small solutes some AQPs serve as conduits for highly volatile solutes $CO_2$ and $NH_3$ (for review see Boron, 2010), NO (Herrera et al., 2006; Herrera & Garvin, 2007) and probably $O_2$ (Moss et al., 2025; Occhipinti et al., 2025; Zhao et al., 2025). Nakhoul et al. (1998) and Cooper and Boron (1998) used the rate of intracellular acidification ($dpH_i/dt$) to examine the effects of expressing human AQP1 on $CO_2$ diffusion across the plasma membrane of *Xenopus laevis* oocytes. They were the first to observe that a membrane protein can act as a channel for a dissolved gas. Later Endeward et al. (2006) confirmed the $CO_2$ permeability of AQP1 in human red blood cells, using mass spectrometry to monitor the disappearance of $C^{18}O^{16}O$. As part of the Endeward study Musa-Aziz introduced the technique of using the rapid, transient increase of surface pH ($pH_S$) – during the application of $CO_2/HCO_3^-$ – as a semiquantitative index of the $CO_2$ influx across the oocyte membrane. This $pH_S$ approach is technically simpler than the others, and its use led to the elucidation of the roles that several other AQPs – including AQP5 – play as $CO_2$ channels (Geyer et al., 2013; Musa-Aziz et al., 2009). Mathematical modelling has provided quantitative insight into the events surrounding $CO_2$ influx, including the time course of $pH_S$ (Calvetti et al., 2020; Musa-Aziz et al., 2014a, 2014b; Occhipinti et al., 2014; Somersalo et al., 2012).

Raina et al. (1995) were the first to clone the cDNA encoding AQP5, obtaining the clone from a library of rat submandibular gland. Others localized AQP5 to the apical membrane of acinar cells in secretory glands, such as salivary glands (Gresz et al., 2001; Larsen et al., 2011; Ma et al., 1999; Nielsen et al., 1997; Steinfeld et al., 2001; Yoshimura et al., 2016) and lacrimal glands (Ishida et al., 1997; Tsubota et al., 2001). AQP5 also is present in the apical membranes of secretory cells of pyloric glands and duodenal glands (Matsuzaki et al., 2003; Parvin et al., 2002), as well as sweat glands (Nejsum et al., 2002). In the lung AQP5 in rats and humans is localized to apical membrane of alveolar type I pneumocytes but not type II pneumocytes (Funaki et al., 1998; Kreda et al., 2001; Nielsen et al., 1997). However in mice AQP5 is present in the apical membranes of both type I and type II pneumocytes (Krane et al., 2001; Matsuzaki et al., 2009). In the kidney AQP5 co-localizes with pendrin at the apical membrane of $\beta$-intercalated cells ($\beta$-ICs) that secrete $HCO_3^-$ into the lumen of cortical collecting duct (Procino et al., 2011). In the aforementioned cells AQP5 presumably plays an important role in transepithelial fluid and electrolyte transport either because of its role as a $H_2O$ channel or, perhaps in the case of $\beta$-intercalated cells, as a $CO_2$ channel that promotes $HCO_3^-$ secretion.

Besides the above glandular tissues AQP5 is present in astrocytes, where it may play a role during metabolic and traumatic injuries (Chai et al., 2013). In the eye AQP5 is present in the epithelia layer (Funaki et al., 1998) and stromal keratocytes (Kumari et al., 2012) of the cornea, as well as in the epithelial and fibre cells of the lens (Kumari et al., 2012). In the palmar epidermis AQP5 is expressed in keratinocytes (Blaydon et al., 2013). In the aforementioned cells and elsewhere in the body AQP5 could play a role in cell-volume regulation and osmo-sensing.

Governing $CO_2$ diffusion across a membrane is Fick's law, a simplified and integrated form of which was introduced by Wroblewski (1879):

$$J_{CO_2} = P_{M,CO_2} \cdot ([CO_2]_o - [CO_2]_i) \quad . \quad (1)$$

Here $J_{CO_2}$ is flux across the membrane; the $P_{M,CO_2}$ is the macroscopic membrane $CO_2$ permeability; and the term in parentheses indicates the transmembrane $CO_2$ concentration gradient expressed in terms of bulk extracellular and intracellular.

For our purposes it is more informative to consider events immediately adjacent to the membrane:

$$J_{CO_2} = P_{M^*,CO_2} \cdot ([CO_2]_{os} - [CO_2]_{is}) \quad , \quad (2)$$

where $[CO_2]_{os}$ refers to a thin film of aqueous solution on the outer surface of the plasma membrane, $[CO_2]_{is}$ refers to a thin film on the inner surface and $P_{M^*,CO_2}$ (with an *) is the corresponding permeability.

Equation (2) better reflects the reaction-diffusion mathematical models to which we refer above. Stated simply $J_{CO_2}$ depends both on $P_{M^*,CO_2}$, which can

be augmented by certain AQPs and other protein channels, and factors that can influence the nanoscopic concentration gradient near the membrane. Among these latter factors are the carbonic anhydrases (CAs), enzymes that reversibly catalyse the hydration of $CO_2$ and the dehydration of $H_2CO_3$ (for review see Boone et al., 2014). Among the 12 enzymatically active human (h) $\alpha$-carbonic anhydrases (hCA), hCA II has one of the highest catalytic rates (see Purkerson & Schwartz, 2007). It is expressed broadly throughout the body and is especially abundant in cells that engage in substantial $CO_2/HCO_3^-$ transport, including red blood cells, the renal proximal tubule and alveolar type I cells (Chen et al., 2008). Moreover CA II appears to play a role in $CO_2$ elimination by the lung (Heming et al., 1986; Lien & Lai, 1998; Taki et al., 1999).

A long-established role of CAs is in the facilitated diffusion of $CO_2$ within a single compartment (for review see Occhipinti & Boron, 2019). In the case of transmembrane $CO_2$ diffusion, when CA is present on both sides of an artificial membrane, the CA can replenish depleted $CO_2$ on the membrane surface from which $CO_2$ departs and consume newly arriving $CO_2$ on the opposite surface. The result is a magnification of the transmembrane $CO_2$ concentration gradient on a nanoscale (see eqn (2)), and thus an increase in transmembrane $CO_2$ diffusion (Gutknecht et al., 1977). Work from our group on *Xenopus* oocytes – in the absence of exogenously expressed $CO_2$ channels – has shown that hCA II (in the cytosol) and CA IV (mainly on the extracellular surface) can each enhance transmembrane $CO_2$ fluxes, and that together the two enzymes produce a supra-additive stimulation synergism (Musa-Aziz et al., 2014a, 2014b; Occhipinti et al., 2014). Moreover the group quantitatively accounted for these effects using computer simulations based on three-dimensional reaction-diffusion models. Left unanswered is the question of whether CAs can enhance the effect of $CO_2$ channels (the $P_{M^*,CO_2}$ term in eqn (2)), and vice versa, on transmembrane $CO_2$ fluxes. In the present study we use simultaneous measurements of $pH_S$ and intracellular pH ($pH_i$) to determine whether hAQP5, cytosolic hCA II and extracellular bovine CA (bCA; from erythrocytes) augment each other's effects in accentuating $CO_2$ fluxes across *Xenopus* oocyte plasma membranes. We find that the effects of expressing hAQP5 are supra-additive (i.e. synergistic) to those of adding CA to either the intra- or extracellular fluid. Moreover in experiments in which we measure the increase in cell-surface pH ($\Delta pH_S$) caused by the influx of $CO_2$ the introduction of CA to the cytosol augments the $pH_S$ increase – a $\Delta\Delta pH_S$ – by an amount that $CO_2$ channels further augment – a $\Delta\Delta\Delta pH_S$. Thus, this $\Delta\Delta\Delta$ approach could be a diagnostic tool for identifying candidate $CO_2$ channels.

## Methods

### Ethical approval and animal procedures

The Institutional Animal Care and Use Committee at Case Western Reserve University approved all procedures for the housing and handling of *X. laevis* (PHS Assurance number – D16-00089 (A3145-01). Housing, anaesthesia and killing were performed essentially as described by Moss and Boron (2020). To alleviate stress we accommodated a maximum of six frogs, each static 20 gallon aquarium tank, filled with dechlorinated water that was circulated using a charcoal Bio-Bag aquarium power pump (Tetra, Blacksburg, VA, USA). We partially replaced water in the tank as necessary, and every 90 days we transferred the frogs to a newly cleaned tank, which was half-filled with water from the previous tank and half-filled with fresh dechlorinated water. We placed a PVC elbow pipe for environmental enrichment in each tank. We fed the frogs thrice per week with an adult *Xenopus* diet (Zeigler Bros. Inc., Gardners, PA, USA). We sprinkled the food (10 pellets per frog) into the aquarium, and the *Xenopus* were permitted to feed. We used a net to remove surplus food after a few hours.

We anaesthetized the *Xenopus* by immersion in a 0.2% tricaine solution. We then removed the animal from the anaesthetic after it became unresponsive to tactile stimuli. We placed ice into a sterile tray, covered the ice with foil and then the anaesthetized *Xenopus* was placed onto the foil surface to surgically remove the ovaries. We killed the animals before regaining consciousness from anaesthesia through cardiac excision.

### Physiological solutions

Our ND96 solution comprises 93.5 mM NaCl, 2 mM KCl, 1 mM $MgCl_2$, 1.8 mM $CaCl_2$ and 5 mM Hepes (including $\sim$2.5 mM Na-Hepes after titration of solution pH to 7.50 using NaOH). We measured solution pH at room temperature using a benchtop pH meter together with a glass pH electrode with a flowing 3M KCl junction ('Ross' electrode, Cat#13-643-201, ThermoFisher, Waltham, MA, USA). We adjusted the osmolality to 195±5 mOsm by adding NaCl or $H_2O$. Our 1.5% $CO_2$/10 mM $NaHCO_3$ solution was identical to ND96, except that 10 mM $NaHCO_3$ replaced 10 mM NaCl, and we bubbled the solution with 1.5% $CO_2$/balance $O_2$. Our ND48 – identical to ND96, except contained only 48 mM NaCl – was a hypotonic solution ($\sim$100 mOsm) for the osmotic water permeability coefficient ($P_f$) assay. The 0 $Ca^{2+}$ normal saline (NRS) solution used during oocyte isolation was as described by Musa-Aziz et al. (2010).

To study the effect of extracellular carbonic anhydrase on $CO_2$ diffusion we dissolved bCA purified from bovine erythrocytes (Cat#C3934, Sigma-Aldrich, St. Louis, MO, USA) – a mixture of CA I and CA II – in both the ND96 and 1.5% $CO_2$/10 mM $NaHCO_3$ solutions of an experiment to achieve a concentration of 0.1 mg/ml (Musa-Aziz et al., 2014b).

OR3 medium, as described by Musa-Aziz et al. (2010), comprises one sachet of Leibovitz L-15 (Cat#41300, ThermoFisher), 100 ml penicillin-streptomycin (Cat#15140, ThermoFisher) and 1.785 g HEPES, all dissolved in ∼1.6 l of deionized $H_2O$ (d$H_2O$), with pH titrated to 7.50 and osmolality adjusted to 195±5 mOsm by adding an appropriate volume of d$H_2O$.

We confirmed all solution osmolalities using a Wescor model 5520 osmometer.

### Oocyte preparation

For a description of our approach see Musa-Aziz et al. (2010) and Parker et al. (2008). Briefly on Day 0 we anaesthetized a frog (female *X. laevis*, NASCO, Fort Atkinson, WI, USA) in 0.2% tricaine (Cat#A5040, Sigma-Aldrich), surgically removed an ovary from the ventral side of the frog, cut the ovary into small pieces (∼0.5 × 0.5 × 0.5 cm) and washed the pieces ×3 with 0 $Ca^{2+}$ NRS in a 50 ml Falcon tube, using a benchtop end-over-end rotator. We then digested the extracellular matrix of the follicles with 2 mg/ml collagenase (Cat#C5138, Sigma-Aldrich) dissolved in 0 $Ca^{2+}$ NRS, washed ×3 in 0 $Ca^{2+}$ NRS, carefully selected stage V and VI oocytes from the processed batch and then cultured in OR3 medium in an incubator at 18°C for subsequent injection of cRNA.

### cDNA construct encoding human AQP5

The cDNA encoding human AQP5 (GenBank# NM_001651.4), subcloned into pGH19 vector, was the same construct as used by Qin and Boron (2012). After the sequence was confirmed, we linearized the cDNA using Xho I (Cat# R0146, New England Biolabs, Ipswich, MA, USA), transcribed it *in vitro* using a mMESSAGE mMACHINE T7 capped RNA transcription kit (Cat#AM1344, ThermoFisher). We purified the cRNA using an RNeasy MinElute Cleanup Kit (Cat#74204, QIAGEN, Hilden, Germany), eluted the resulting cRNA with DNase/RNase-free $H_2O$, quantified the cRNA concentration by measuring peak absorbance at $\lambda = 260$ nm ($A_{260}$) and assessed its purity by measuring the ratio of the $\lambda = 260$ nm *vs.* $\lambda = 280$ nm absorbance ($A_{260/280}$) using a NanoDrop 2000 UV spectrophotometer (ThermoFisher), and diluted the cRNA into 1000 ng/μl aliquots, which we stored at –80°C.

### Expression in *Xenopus* oocytes

On Day 1 we injected oocytes with 25 ng (25 nl) of cRNA encoding hAQP5 or 25 nl of $H_2O$ as a control. On Day 4 we separated both cRNA-injected and $H_2O$-injected oocytes into two subgroups, one pair of subgroups for injection with 25 nl containing 1, 10 or 100 ng of hCA II protein purified from human erythrocytes (Cat #C6165, Sigma-Aldrich) dissolved in 'Tris' (i.e. 50 mM Tris base, titrated with HCl) at pH 7.40, and the other pair of subgroups for injection with 25 nl of 'Tris' as a control.

We selected healthy oocytes in three ways: (1) we used only oocytes that appeared healthy when observed under a dissecting microscope (see Musa-Aziz et al., 2010) while still in the Petri dish. (2) Although membrane potential ($V_m$; see below) is not a universal indicator of oocyte health, we generally accepted only oocytes in which values were more negative than –40 mV. (3) We discarded oocytes that, as we pushed the $pH_S$ electrode against the membrane (under microscopic observation in the experimental chamber), exhibited a subjectively soft membrane (often accompanied by a splotchy distribution of colours).

### Electrophysiological measurements

**Solution delivery to the chamber.** At room temperature we put each solution into two 140 ml piston syringes (Cat#8881114030, Covidien, Dublin, Ireland) and delivered the solution using a dual syringe pump (Cat#55-2226, Harvard Apparatus, Holliston, MA, USA) at a constant total flow of 4 ml/min. The solutions flowed through Tygon tubing (4.76 mm OD × 1.59 mm ID, Cat#ACF00003, Saint-Gobain, Courbevoie, France) to minimize the leak of $CO_2$. We used custom-manufactured five-way valves (Clippard, Cincinnati, OH, USA) actuated by nitrogen pressure to switch solutions between ND96 and 1.5% $CO_2$/10 mM $NaHCO_3$. We controlled the nitrogen pressure using electrically activated four-way solenoid valves (R481, Clippard).

**Construction of microelectrodes.** We made $V_m$ electrodes from borosilicate tubing (2.00 mm OD × 1.56 mm ID, with filament; Cat#BF200-156-10, Sutter Instrument, Novato, CA, USA), which we pulled on a Sutter puller (model P-97, Sutter Instrument), and filled with 3 M KCl to achieve a resistance of 0.5 to 1.0 MΩ. For $pH_i$ electrodes after pulling we dried the micropipettes at 270°C overnight, treated with *bis*-di-(methylamino)-dimethylsilane (Cat#14755, Sigma-Aldrich) for 20 min, vented the silane vapours, maintained the silanized micropipettes at 270°C overnight, removed them from the oven, filled the tip with $H^+$ ionophore I-cocktail B (Cat#95293, Sigma-Aldrich) and then filled with the backfill solution (see Musa-Aziz et al., 2010). We made $pH_S$ electrodes in a fashion as $pH_i$

electrodes, except that after pulling the tubing (2.00 mm OD × 1.16 mm ID, without filament; Cat#B200-116-10, Sutter Instrument), we used a microforge to break the micropipette tip to give a final inner diameter of 30 μm. The thicker wall for the pH$_S$ electrodes produces a better tip break, and the lack of the filament stabilizes the H$^+$ cocktail.

**Measurement of pH$_S$ and pH$_i$**

*Positioning of electrodes.* We measured pH$_S$ and pH$_i$ as previously described (Musa-Aziz et al., 2010) and illustrated schematically (Musa-Aziz et al., 2014a, 2014b; Occhipinti et al., 2014). Briefly after pre-positioning all electrodes (see below) in the plastic chamber, we introduced an oocyte. The flow of ND96 solution pinned the oocyte against X-shaped nylon fibres immediately downstream, holding the oocyte in place. The final disposition of the microelectrodes (see below) further stabilized the oocyte. We connected the $V_m$ electrode to an OC-725C oocyte clamp (Warner Instruments, Hamden, CT, USA) and the pH$_i$ and pH$_S$ electrodes to a HiZ-223 amplifier (Warner Instruments). The electrical ground of the chamber was a bath clamp in which the same OC-725C as above connected to (a) a platinum wire in the chamber and (b) a reference electrode (identical to a $V_m$ electrode but with its tip broken by dragging on paper) that contacted the ND96 downstream from the oocyte. The reference electrode for the pH$_S$ measurements was a longer version of a $V_m$ microelectrode (i.e. filled with 3M KCl), but with a carefully broken tip, placed downstream from the oocyte, and connected via a plastic electrode holder and calomel half-cell to a model 750 amplifier (WPI, Sarasota, FL, USA). All of the primary electrical signals fed into custom data-acquisition hardware 'Ribbit Box' built in-house by Dale Huffman around a LabJack U6Pro device (LabJack, Lakewood, CO, USA) and connected to a Windows-based computer, controlled by custom software 'The Frog Whisperer', written in-house by Dale Huffman.

*Subtraction of electrical signals.* Our sampling frequency was 3/s. We obtained the pH$_i$ value by digitally subtracting (in the computer) the $V_m$ signal (voltage) from the pH$_i$-electrode signal (voltage), and converted the difference to a pH$_i$ value using the electrode-calibration data (see below). We similarly obtained the pH$_S$ value by subtracting the signal of pH$_S$ reference (calomel) from that of the pH$_S$ electrode. The above custom software performed the calculations and displayed the record of all electrical parameters *vs.* time on a computer monitor.

**Calibration of pH microelectrodes.** We calibrated pH microelectrodes in two steps, first in certified pH standards (not appropriate for physiological experiments)

and then in one of our physiological solutions that we had titrated to pH 7.50 using a commercial glass electrode (the 'gold standard'), as described above.[1]

The first time we used a new pH$_S$ or pH$_i$ electrode – we never used an electrode for more than 1 day – we obtained the electrode slope using the solution-delivery system (see above) to flow sequentially a pH 6.00 or pH 8.00 buffer (Certified, Cat#SB104-1 and SB112-1, Thermo-Fisher) through the chamber – before adding the oocyte – and measuring the voltage signal as described above. This exercise provided the 'electrode slope'. We accepted an electrode only if it had a slope of at least 55 mV/pH. We then flushed the chamber with our ND96 solution before adding the oocyte and performed a one-point calibration for each electrode in ND96. This second exercise provided the 'electrode offset' in the physiological solution, assuming the slope to be the same as in the certified pH standards. We assigned to the subtracted pH$_S$ or pH$_i$ electrode voltage, obtained as described above,[2] the pH value of 7.50 in the ND96 solution. Because the liquid-membrane pH electrodes can be sensitive to CO$_2$, we repeated this calibration during the experiment in the CO$_2$/HCO$_3^-$ solution.

**Recording of electrophysiological data.** In consecutive fashion, with ND96 flowing in the chamber, we impaled the oocyte with a $V_m$ electrode and a pH$_i$ electrode and then positioned the tip of the pH$_S$ electrode ∼300 μm from the oocyte surface using a precision remote-controlled micro-manipulator (Cat#ROE200, Sutter Instrument), all as previously described (Musa-Aziz et al., 2014a). Although maintaining the flow of ND96, we advanced the pH$_S$ electrode tip ∼300 μm to touch the oocyte surface, and an additional 40 μm to create a slight dimple on the oocyte surface. After 1 min, we switched the solution from ND96 to 1.5% CO$_2$/10 mM NaHCO$_3$. After several minutes, we retracted the pH$_S$ electrode 340 μm from the oocyte surface for 1 min to obtain a one-point calibration of the pH$_S$ electrode in the bulk 1.5% CO$_2$/10 mM NaHCO$_3$ solution. We then readvanced the pH$_S$ electrode by 340 μm to resume pH$_S$ measurements before switching back to ND96. Finally after pH$_S$ had stabilized we retracted the pH$_S$ electrode by 340 μm to obtain a one-point calibration in the bulk ND96 solution at the end of the experiment.

**Calculation of ΔpH$_S$.** With its tip in the bulk ND96 solution the pH$_S$ electrode detected, by definition, a pH of 7.50 (i.e. the initial ND96 calibration). Once dimpling

---

[1]See 'Methods' > 'Physiological solutions' > 'Ross'. Throughout the paper, '>' directs the reader to a subsection of the preceding reference.
[2]See 'Methods' > 'Electrophysiological measurements' > 'Measurement of pH$_S$ and pH$_i$' > 'Subtraction of electrical signals'.

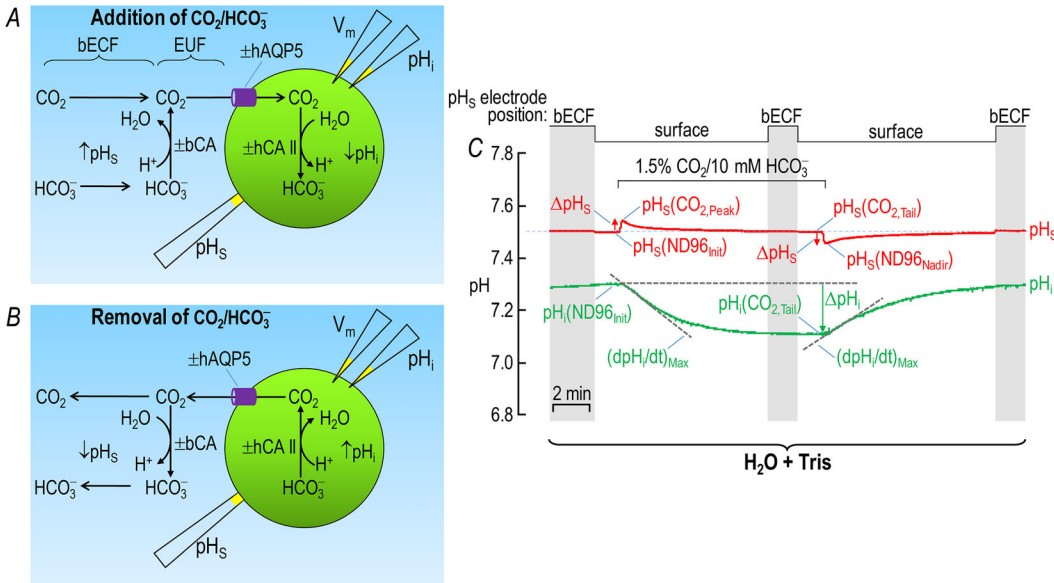

**Figure 1. Design of experiments**

A, model cell showing $CO_2$ influx. The green circle represents a *Xenopus laevis* oocyte. Arrows with flared arrowheads indicate the direction of diffusion. Arrows with triangular arrowheads indicate the direction of a chemical reaction. Upward and downward arrows associated with $pH_S$ and $pH_i$ indicate the direction of a pH change. The purple cylinder represents a human (h) AQP5 tetramer heterologously expressed in the membrane. We impale a *X. laevis* oocyte with both a $pH_i$ and $V_m$ electrode and used a blunt, polished electrode to monitor $pH_S$. ±, presence or absence; bECF, bulk extracellular fluid; bCA, extracellular bovine carbonic anhydrase (a mixture of CA I and CA II); EUF, extracellular unconvected fluid; hCA II, intracellular purified carbonic anhydrase II protein from human erythrocytes. B, model cell showing $CO_2$ efflux. All events occur in a direction opposite to that in A. C, representative experiment. We dissected the oocyte on Day 0, injected it with $H_2O$ (as a control for cRNA encoding hAQP5) on Day 1 and then with 'Tris' (as a control for hCA II protein) on Day 4. The stair-step line at the top of the panel indicates the position of the tip of the $pH_S$ electrode. The three vertical shaded bars indicate the time during which the $pH_S$-electrode tip is in the bECF for a 1-point recalibration at $pH_o = 7.50$; at other times the $pH_S$-electrode tip dimples the cell surface. We used the first and third recalibration periods for assignment of $pH_S$ values for times that the oocyte was exposed to ND96, and the middle recalibration period for assignment of $pH_S$ values for times that the oocyte was exposed to $CO_2/HCO_3^-$. The flowing solution is ND96, except for the period labelled '1.5 $CO_2$/10 mM $HCO_3^-$'. The red record represents the time course of $pH_S$, and the green record represents the time course of $pH_i$. We provide detailed definitions of the red and green labels in Methods.[3]

the membrane with the oocyte exposed to ND96 the $pH_S$ electrode – calibrated for ND96 – detected $pH_S$. The average $pH_S$ over at least the final 30 s – when $pH_S$ was stable – we took as $pH_S(ND96_{Init})$. From the moment that we switched the chamber solution to 1.5% $CO_2$/10 mM $NaHCO_3$ the initial ND96 calibration of the $pH_S$ electrode was no longer valid, which is why we obtained a new one-point calibration in 1.5% $CO_2$/10 mM $NaHCO_3$ (see above); we applied this second $CO_2/HCO_3^-$ calibration to all $pH_S$ data obtained during the $CO_2/HCO_3^-$ exposure, as $pH_S$ rises to a peak and then decays downwards. Using this second calibration we computed the peak $pH_S$ after the switch to $CO_2/HCO_3^-$, that is, $pH_S(CO_{2,Peak})$. We define the 'upward' $\Delta pH_S$ for $CO_2$ addition as $pH_S(CO_{2,Peak}) - pH_S(ND96_{Init})$. At the end of the $CO_2/HCO_3^-$ exposure the average $pH_S$ over at least the final 30 s – when $pH_S$ was stable – we took as $pH_S(CO_{2,Tail})$. From the moment that we switched the chamber solution to ND96 the previous 1.5% $CO_2$/10

mM $NaHCO_3$ calibration of the $pH_S$ electrode was no longer valid, which is why we obtained a final (i.e. third) one-point calibration in ND96 (see above); we applied this final ND96 calibration to all $pH_S$ data obtained during the final ND96 exposure, as $pH_S$ falls to a trough and then decays upwards. Using this third calibration we computed the lowest $pH_S$ after the switch to ND96, namely, $pH_S(ND96_{Nadir})$. We define the 'downward' $\Delta pH_S$ for $CO_2$ removal as $pH_S(ND96_{Nadir}) - pH_S(CO_{2,Tail})$, which is a negative number.

**Calculation of 'maximal' $dpH_S/dt$.** During $CO_2/HCO_3^-$ addition (e.g. see Fig. 1C) $pH_S$ rises rapidly from around 7.50 to a peak (where $dpH_S/dt \cong 0$) and then begins to decline, at first slowly ($dpH_S/dt$ is slightly negative), later

---

[3] See 'Methods' > 'Calculation of $\Delta pH_S$', 'Methods' > "Determination of 'maximal' $dpH_i/dt$", 'Methods' > 'Calculation of intrinsic intracellular buffering power'

rapidly ($dpH_S/dt$ reaches a maximally negative value) and finally more slowly again ($dpH_S/dt$ gradually becomes less and less negative) as $pH_S$ relaxes towards an asymptotic value around 7.50. During $CO_2/HCO_3^-$ removal the changes are in the opposite direction. Here we describe the analysis for $CO_2/HCO_3^-$ addition, but the approach is similar for $CO_2/HCO_3^-$ removal. Using software written in-house by Dale Huffman we obtain a running linear fit of ($pH_S$, time) data points to generate a plot of $(dpH_S/dt)_{Running}$ *vs.* time, which allows us to identify the time (which we define as local time: $t_{Local} = 0$) of the maximally negative $dpH_S/dt$ (i.e. minimal $dpH_S/dt$). We then obtain a double-exponential (DExp) fit of the ($pH_S$, time) data points, beginning at $t_{Local} = 0$ and extending to the time when we removed the $pH_S$ electrode from the cell surface to the bulk extracellular fluid (bECF) (for calibration). Finally we evaluate the derivative of the DExp function at $t_{Local} = 0$ and take this value as $(dpH_S/dt)_{Max}$. During $CO_2/HCO_3^-$ addition this value is negative (i.e. maximal rate of $pH_S$ descent), and during $CO_2/HCO_3^-$ removal (e.g. see Fig. 1*C*) this value is positive (i.e. maximal rate of $pH_S$ rise).

**Calculation of 'maximal' $dpH_i/dt$.** After either the addition or removal of $CO_2/HCO_3^-$ we define $(dpH_i/dt)_{Max}$ as the fastest change in $pH_i$, which occurs several seconds after the new solution reaches the oocytes (Musa-Aziz et al., 2014a). For $CO_2/HCO_3^-$ addition $pH_i$ decreases and thus the 'downward' $(dpH_i/dt)_{Max}$ is a negative number; upon $CO_2/HCO_3^-$ removal the 'upward' $(dpH_i/dt)_{Max}$ is positive. After the time of the fastest $pH_i$ change was visually identified (i.e. corresponding to the most extreme local value of $dpH_i/dt$) we used Origin 2024 to obtain $(dpH_i/dt)_{Max}$ by a linear fit over the next ~10 s of ($pH_i$, time) data.

**Calculation of intrinsic intracellular buffering power.** We defined initial $pH_i$ in ND96 solution – $pH_i(ND96_{Init})$ – as the average of at least 90 data points over a period (i.e. 30 s) when the experimenter (in real experimental time) judged $pH_i$ to be stable just before switching from ND96 to the $CO_2/HCO_3^-$ solution. Similarly we defined the final $pH_i$ in the $CO_2/HCO_3^-$ solution – $pH_i(CO_2,_{Tail})$ – as the average of at least 90 data points over a period when the experimenter (in real experimental time) judged $pH_i$ appeared stable just before switching from the $CO_2/HCO_3^-$ solution to ND96. During $CO_2/HCO_3^-$ application we defined $\Delta pH_i$ (a negative number) as $pH_i(CO_2,_{Tail})$ – $pH_i(ND96_{Init})$.

Based on the decrease in steady-state $pH_i$ produced by the aforementioned application of $CO_2/HCO_3^-$ we calculated the intrinsic intracellular buffering (Boron, 1977) power $\beta_I$ (mM/pH) of oocytes using the following equation:

$$\beta_I = -\frac{\Delta[HCO_3^-]_i}{\Delta pH_i}$$

Here the negative sign arises because we are observing a $pH_i$ decrease produced by an intracellular, $CO_2$-induced acid load. Because we impose the acid load by the introduction of $CO_2/HCO_3^-$, followed by the intracellular reaction $CO_2 + H_2O \rightarrow H^+ + HCO_3^-$, the magnitude of the acid load is $\Delta[HCO_3^-]_i$ (for a discussion see Roos & Boron, 1981). Because we assume that $[HCO_3^-]_i$ is 0 before the exposure to $CO_2/HCO_3^-$, $\Delta[HCO_3^-]_i$ (a positive number) is the same as the final $[HCO_3^-]_i$ after the introduction of $CO_2/HCO_3^-$, determined at a time when $pH_i$ is stable. Because we observed no $pH_i$ recovery during the $CO_2/HCO_3^-$ exposure, nor a $pH_i$ overshoot after the removal of $CO_2/HCO_3^-$, we can conclude that net acid extrusion during the $CO_2/HCO_3^-$ exposure was negligible (Boron & De Weer, 1976). Thus we computed the final $[HCO_3^-]_i$ using $pH_i(CO_2,_{Tail})$. Because we used 1.5% $CO_2$/10 mM $HCO_3^-$ at a pH of 7.5, we can use the shorthand equation:

$$[HCO_3^-]_i = [HCO_3^-]_{o,@pHo=7.50} \cdot 10^{pH_i-7.50}$$

Here we assume that $[CO_2]_i = [CO_2]_o$, where the subscript 'o' denotes the bECF, and that the $pK_a$ in the intracellular fluid is the same as in the extracellular fluid.

## Measurement of $P_f$

After the $pH_S$ and $pH_i$ were recorded we saved the oocyte, equilibrated it in ND96 for at least 10 min, transferred it to a Petri dish containing the hypotonic solution ND48 and a 1.6 mm diameter steel sphere (i.e. a ball bearing), placed the dish under a dissecting microscope (model Stemi 508, ZEISS, Oberkochen, Germany) equipped with a video camera (OptixCam summit series, Microscope LLC, Roanoke, VA, USA) connecting to a computer running proprietary software and recorded 1 image/s of the swelling oocyte and the nearby steel sphere for 1 min. We used Image J software (U. S. National Institutes of Health, Bethesda, MA, USA, https://imagej.nih.gov/ij) to determine the perimeter of the oocyte at each time point, computed the projection area (using the steel sphere as an area standard) and the oocyte volume (assuming the oocyte to be a sphere) and used the following equation to calculate osmotic water permeability:

$$P_f = \frac{V_0 \cdot \frac{d(V/V_0)}{dt}}{S \cdot \Delta Osm \cdot V_w}$$

Here, $V_0$ is the initial oocyte volume, $d(V/V_0)/dt$ is the rate of volume increase during the first minute, $\Delta Osm$ is the osmotic gradient across the membrane (i.e. 195

mOsm – 100 mOsm), $V_W$ is the molar volume of water (18 ml/molar) and S is the oocyte surface area, assumed to be eightfold greater than the idealized area (Chandy et al., 1997).

## Statistical analysis

We present data as mean $\pm$ SD. To compare the difference between two or more groups we use a one-way ANOVA followed by the Tukey *post hoc* analysis. We perform the analyses using Origin 2024, considering $P < 0.05$ as significant.

To compare means of analysed parameters of oocytes exposed to bECF lacking bCA (grey, black, light- and dark-blue bars in Fig. 3*A–D*) with oocytes for which the bECF contained bCA (Fig. 8*A–D*) we perform a one-way ANOVA with Tukey's means comparisons. In Statistics Table 8*E-H* we present descriptive statistics and means comparisons to determine the statistical significance of bCA-dependent differences.

To compare mean values of various other parameters – initial $pH_i$, $\Delta pH_i$, $\beta_I$ and $P_f$ – between oocytes not exposed to bCA (grey, black, light- and dark-blue bars in Fig. 6*A–D*) and those exposed to bCA (Fig. 10*A–D* (bECF +bCA) we use the same approach described in the previous paragraph. In Table 10E–H we present descriptive statistics and means comparisons to determine the statistical significance of bCA-dependent differences.

In Fig. 11 we determine the mean $\Delta\Delta pH_{S,Base}$ by subtracting the mean $\Delta pH_S$ for –hAQP5/–hCA II oocytes from each measured $\Delta pH_S$ of –hAQP5/+hCA II oocytes. We determined the mean $\Delta\Delta pH_{S,hAQP5}$ by subtracting the mean $(\Delta pH_S)$ for +hAQP5/–hCA II oocytes from each measured $\Delta pH_S$ of +hAQP5/+hCA II oocytes. We then performed a one-way ANOVA followed by the Tukey *post hoc* analysis on the data for $\Delta\Delta pH_{S,Base}$ *vs.* $\Delta\Delta pH_{S,hAQP5}$.

In Fig. 12 we determine the mean $\Delta(dpH_i/dt)_{Max,Base}$ by subtracting the mean $(dpH_i/dt)_{Max}$ for –hAQP5/–bCA oocytes from each measured $(dpH_i/dt)_{Max}$ of –hAQP5/+bCA oocytes. We determined the mean $\Delta(dpH_i/dt)_{Max,hAQP5}$ by subtracting the mean $\Delta(dpH_i/dt)_{Max}$ for +hAQP5/–bCA oocytes from each measured $\Delta(dpH_i/dt)_{Max}$ of +hAQP5/+bCA oocytes. We then performed a one-way ANOVA followed by the Tukey *post hoc* analysis on the data for $\Delta(dpH_i/dt)_{Max,Base}$ *vs.* $\Delta(dpH_i/dt)_{Max,hAQP5}$.

## Results

### General protocol

Figure 1*A* and *B* is a schematic representation suggesting how hAQP5, cytosolic hCA II and extracellular bCA would affect the influx (panel A) or efflux of $CO_2$ (panel

B) and thus $pH_S$ and $pH_i$ of an *Xenopus* oocyte. Imagine that we switch the solution – the bECF[4] – that flows through our chamber from one that nominally lacks to one that contains $CO_2/HCO_3^-$. Imagine also that we have a cell whose membrane is initially impermeable to $CO_2$ and $HCO_3^-$. Thus a short time after we switch solutions $CO_2$ and $HCO_3^-$ will have diffused throughout the system, so that $[CO_2]$ and $[HCO_3^-]$ will be uniform through the bECF (an 'infinite reservoir') and extracellular unconvected fluid (EUF), including a thin layer of fluid at the oocyte surface (S). Suddenly we now increase $P_{M,CO_2}$, which is where Fig. 1*A* picks up the narrative. Now $CO_2$ diffuses into the cell (Fig. 1*A*), leading to the depletion of $CO_2$ at the cell surface. The replenishment of this surface $CO_2$ occurs via both $CO_2$ diffusion from bECF and – especially adjacent to the cell surface – the reaction $HCO_3^- + H^+ \rightarrow CO_2 + H_2O$. The reaction causes a rise in $pH_S$, the maximum magnitude of which is $\Delta pH_S$; assuming that cell-surface CA activity is fixed $\Delta pH_S$ is a semiquantitative index of the $CO_2$ influx (see eqn (2)). Meanwhile beneath the membrane the $CO_2$ that has entered the cell undergoes the reaction $CO_2 + H_2O \rightarrow H^+ + HCO_3^-$, thus causing a fall in $pH_i$. The maximal rate of intracellular acidification – assuming that cytosolic CA activity is fixed – is also an index of the $CO_2$ influx. However $(dpH_i/dt)_{Max}$ is a far less sensitive measure than $\Delta pH_S$ (Musa-Aziz et al., 2014b).

If we nominally remove $CO_2/HCO_3^-$ from the solution flowing through the chamber, all of the above processes, including changes in $pH_S$ and $pH_i$, reverse (Fig. 1*B*).

Figure 1*C* shows typical $pH_S$ and $pH_i$ records of a control oocyte – that is, one injected with $H_2O$ (rather than cDNA dissolved in $H_2O$ encoding hAQP5) on Day 1 and then 'Tris' (rather than hCA II dissolved in 'Tris') on Day 4 – during the application and removal of $CO_2/HCO_3^-$. No bCA is present in the bECF. The shaded vertical bars represent periods of time during which the $pH_S$ electrode is ~300 μm from the cell surface, in the bECF, for electrode recalibration (for details see figure legend and Methods[5]). At other times (i.e. between shaded bars) the electrode tip dimples the oocyte surface. Switching the solution from ND96 to 1.5% $CO_2$/10 mM $NaHCO_3$ causes an upward $pH_S$ transient that soon reaches a maximum (left upward red arrow) and a $pH_i$ decrease, the maximum rate of which is indicated by the left dashed line. The $pH_S$ signal (after the $pH_S$ peak) relaxes and $pH_i$ signal declines with similar time constants (Musa-Aziz et al., 2014a, 2014b; Occhipinti et al., 2014). Later we switch the solution back to ND96, which elicits a

---

[4]For simplicity, we refer to concentrations in the bECF using the subscript "o".

[5]See 'Methods' > 'Electrophysiological measurements' > 'Calibration of pH microelectrodes'.

**Statistics Table 3.** For clarity within the figure panel we present tables of *P*-values for one-way ANOVA with Tukey's *post hoc* means comparison for the data presented in Fig. 3. For all tables $\alpha$ is 0.05, and significant *P*-values are highlighted in bold. **A**, *P*-values for means comparisons of $\triangle pH_S$ upon $CO_2$ addition data. **B**, *P*-values for means comparisons of $\triangle pH_S$ upon $CO_2$ removal data. **C**, *P*-values for means comparisons of $(dpH_i/dt)_{Max}$ upon $CO_2$ addition data. **D**, *P*-values for means comparisons of $(dpH_i/dt)_{Max}$ upon $CO_2$ removal data

### 3A, $\triangle pH_S$ upon $CO_2$ addition

| cRNA | | $H_2O$ | hAQP5 | $H_2O$ | hAQP5 | $H_2O$ | hAQP5 | $H_2O$ |
|---|---|---|---|---|---|---|---|---|
| | hCA II (ng) | 'Tris' | 'Tris' | 1 | 1 | 10 | 10 | 100 |
| hAQP5 | 'Tris' | **0.00150** | | | | | | |
| $H_2O$ | 1 | 0.956 | **0.0391** | | | | | |
| hAQP5 | 1 | **4.18×10⁻⁸** | **0.00247** | **7.19×10⁻⁸** | | | | |
| $H_2O$ | 10 | 0.943 | **0.0456** | 1.00 | **7.72×10⁻⁸** | | | |
| hAQP5 | 10 | **3.90×10⁻⁸** | **7.38×10⁻⁴** | **5.43×10⁻⁸** | 1.00 | **5.60×10⁻⁸** | | |
| $H_2O$ | 100 | 0.947 | **0.0437** | 1.00 | **7.56×10⁻⁸** | 1.00 | **5.55×10⁻⁸** | |
| hAQP5 | 100 | **2.21×10⁻⁸** | **1.34×10⁻⁷** | **2.93×10⁻⁸** | 0.0978 | **2.97×10⁻⁸** | 0.214 | **2.95×10⁻⁸** |

### 3B, $\triangle pH_S$ upon $CO_2$ removal

| cRNA | | $H_2O$ | hAQP5 | $H_2O$ | hAQP5 | $H_2O$ | hAQP5 | $H_2O$ |
|---|---|---|---|---|---|---|---|---|
| | hCA II (ng) | 'Tris' | 'Tris' | 1 | 1 | 10 | 10 | 100 |
| hAQP5 | 'Tris' | **2.03×10⁻⁴** | | | | | | |
| $H_2O$ | 1 | 0.992 | **0.00309** | | | | | |
| hAQP5 | 1 | **4.44×10⁻⁸** | **0.0321** | **7.16×10⁻⁸** | | | | |
| $H_2O$ | 10 | 0.995 | **0.00249** | 1.00 | **6.67×10⁻⁸** | | | |
| hAQP5 | 10 | **4.51×10⁻⁸** | **0.0383** | **7.76×10⁻⁸** | 1.00 | **7.15×10⁻⁸** | | |
| $H_2O$ | 100 | 0.994 | **0.00278** | 1.00 | **6.91×10⁻⁸** | 1.00 | **7.44×10⁻⁸** | |
| hAQP5 | 100 | **3.17×10⁻⁸** | **1.50×10⁻⁴** | **3.72×10⁻⁸** | 0.692 | **3.68×10⁻⁸** | 0.650 | **3.70×10⁻⁸** |

### 3C, $(dpH_i/dt)_{Max}$ upon $CO_2$ addition

| cRNA | | $H_2O$ | hAQP5 | $H_2O$ | hAQP5 | $H_2O$ | hAQP5 | $H_2O$ |
|---|---|---|---|---|---|---|---|---|
| | hCA II (ng) | 'Tris' | 'Tris' | 1 | 1 | 10 | 10 | 100 |
| hAQP5 | 'Tris' | 1.00 | | | | | | |
| $H_2O$ | 1 | 0.0803 | 0.0744 | | | | | |
| hAQP5 | 1 | **0.00798** | **0.00726** | 0.989 | | | | |
| $H_2O$ | 10 | **3.62×10⁻⁴** | **3.25×10⁻⁴** | 0.624 | 0.979 | | | |
| hAQP5 | 10 | **3.76×10⁻⁵** | **3.36×10⁻⁵** | 0.244 | 0.753 | 0.998 | | |
| $H_2O$ | 100 | **5.69×10⁻⁴** | **5.10×10⁻⁴** | 0.708 | 0.991 | 1.00 | 0.994 | |
| hAQP5 | 100 | **0.00210** | **0.00189** | 0.905 | 1.00 | 0.999 | 0.938 | 1.00 |

### 3D, $(dpH_i/dt)_{Max}$ upon $CO_2$ removal

| cRNA | | $H_2O$ | hAQP5 | $H_2O$ | hAQP5 | $H_2O$ | hAQP5 | $H_2O$ |
|---|---|---|---|---|---|---|---|---|
| | hCA II (ng) | 'Tris' | 'Tris' | 1 | 1 | 10 | 10 | 100 |
| hAQP5 | 'Tris' | 1.00 | | | | | | |
| $H_2O$ | 1 | 0.154 | 0.141 | | | | | |
| hAQP5 | 1 | **0.0330** | **0.0295** | 0.998 | | | | |
| $H_2O$ | 10 | **0.00381** | **0.00335** | 0.865 | 0.995 | | | |
| hAQP5 | 10 | **4.57×10⁻⁵** | **3.97×10⁻⁴** | 0.479 | 0.862 | 0.998 | | |
| $H_2O$ | 100 | **0.00160** | **0.00140** | 0.721 | 0.972 | 1.00 | 1.00 | |
| hAQP5 | 100 | **0.0460** | **0.0413** | 1.00 | 1.00 | 0.988 | 0.801 | 0.947 |

**Statistics Table 5.** For clarity within the figure panel we present tables of *P*-values for one-way ANOVA with Tukey's *post hoc* means comparison for the data presented in Fig. 5. For all tables $\alpha$ is 0.05, and significant *P*-values are highlighted in bold. A, *P*-values for means comparisons of $(dpH_S/dt)_{Max}$ upon $CO_2$ addition. B, *P*-values for means comparisons of $(dpH_S/dt)_{Max}$ upon $CO_2$ removal. Where the displayed *P*-values are 0.00, this indicates a values less than $<2.22 \times 10^{-308}$.

*5A, $(dpH_S/dt)_{Max}$ upon $CO_2$ addition*

| cRNA | | $H_2O$ | hAQP5 | $H_2O$ | hAQP5 | $H_2O$ | hAQP5 | $H_2O$ |
|---|---|---|---|---|---|---|---|---|
| | hCA II (ng) | 'Tris' | 'Tris' | 1 | 1 | 10 | 10 | 100 |
| hAQP5 | 'Tris' | **$4.79 \times 10^{-4}$** | | | | | | |
| $H_2O$ | 1 | 0.998 | **0.00518** | | | | | |
| hAQP5 | 1 | **$4.73 \times 10^{-8}$** | 0.104 | **$7.84 \times 10^{-8}$** | | | | |
| $H_2O$ | 10 | 0.975 | **0.00655** | 1.00 | **$4.77 \times 10^{-8}$** | | | |
| hAQP5 | 10 | **$3.42 \times 10^{-8}$** | **$7.41 \times 10^{-4}$** | **$4.18 \times 10^{-8}$** | 0.408 | **$4.03 \times 10^{-8}$** | | |
| $H_2O$ | 100 | 0.993 | **0.0134** | 1.00 | **$4.89 \times 10^{-7}$** | 1.00 | **$4.78 \times 10^{-8}$** | |
| hAQP5 | 100 | **$2.85 \times 10^{-8}$** | **0.00** | **$2.85 \times 10^{-8}$** | **$4.74 \times 10^{-8}$** | **0.00** | **$3.95 \times 10^{-4}$** | **0.00** |

*5B, $(dpH_S/dt)_{Max}$ upon $CO_2$ removal*

| cRNA | | $H_2O$ | hAQP5 | $H_2O$ | hAQP5 | $H_2O$ | hAQP5 | $H_2O$ |
|---|---|---|---|---|---|---|---|---|
| | hCA II (ng) | 'Tris' | 'Tris' | 1 | 1 | 10 | 10 | 100 |
| hAQP5 | 'Tris' | **0.0151** | | | | | | |
| $H_2O$ | 1 | 0.999 | 0.0862 | | | | | |
| hAQP5 | 1 | **$7.67 \times 10^{-4}$** | 0.998 | **0.00783** | | | | |
| $H_2O$ | 10 | 0.995 | 0.0768 | 1.00 | **0.00516** | | | |
| AQP5 | 10 | **0.00116** | 0.992 | **0.0125** | 1.00 | **0.00975** | | |
| $H_2O$ | 100 | 0.995 | 0.165 | 1.00 | **0.00228** | 1.00 | **0.0303** | |
| AQP5 | 100 | **$3.68 \times 10^{-8}$** | **0.0114** | **$4.03 \times 10^{-7}$** | **0.0280** | **$1.21 \times 10^{-7}$** | 0.144 | **$2.06 \times 10^{-6}$** |

$2.22 \times 10^{-308}$ is the smallest possible value for double type data that a 64-bit system is able to distinguish.

reversal of the previous events. In Methods and the figure legend we define the red and green labels shown in Fig. 1*C*.

In the following eight sections we describe the effects – on various parameters measured during our standard protocol in Fig. 1*C* – of various combinations of human AQP5 expressed or not (±hAQP5), human CA II injected or not (±hCA II) and bovine CA added to the bulk extracellular fluid or not (±bCA). In the first four sections we examine oocytes in the absence of bCA but with various combinations of ±hAQP5 and ±hCA II. In the second group of four we study oocytes in the presence of extracellular bCA.

### ±hAQP5[6] in absence of exogenous CAs: effects on $\Delta pH_S$ and $(dpH_i/dt)_{Max}$

Our overall goal is to investigate the relative roles of hAQP5, cytosolic hCA II and extracellular bCA in promoting $CO_2$ fluxes across the plasma membrane of a *Xenopus* oocyte. In this first section we begin by verifying previous data that addressed the role of hAQP5 in promoting $CO_2$ diffusion in the absence of added CAs.

We injected oocytes on Day 1 with either '$H_2O$' or $H_2O$ + cRNA encoding 'hAQP5' and then, on Day 4, with 'Tris' (as a control for 'Tris + hCA II').

**$pH_S$.** The red record in Fig. 2*A* shows the $pH_S$ data for a representative '$H_2O$ + Tris' oocyte. The red record in Fig. 2*B* shows the comparable data for an 'hAQP5 + Tris' oocyte. Consistent with previous results (Geyer et al., 2013; Musa-Aziz et al., 2009) the expression of hAQP5 leads to $pH_S$ transients that are substantially greater in magnitude – both with the addition of $CO_2/HCO_3^-$ (0.101 *vs.* 0.028) and with the removal of $CO_2/HCO_3^-$ (−0.129 *vs.* −0.032) – than the mere injection of $H_2O$ into the oocytes (red records in Fig. 2*A vs. B*).

*Summary.* For a larger number of these oocytes studied in the absence of both hCA II and bCA the bars in Fig. 3*A* provide the mean $\Delta pH_S$ data for $CO_2/HCO_3^-$ addition in the absence (grey) and presence (black) of hAQP5, respectively. The grey and black bars in Fig. 3*B* summarize the corresponding data for $CO_2/HCO_3^-$ removal.

---

[6] Here, ±hAQP5 indicates that experiments in this section compare pH responses in the absence vs. presence of hAQP5.

[7] See 'Methods' > 'Statistical analysis' > 'Calibration of pH micro-electrodes'.

**Statistics Table 6.  For clarity within the figure panel we present tables of *P*-values for one-way ANOVA with Tukey's *post hoc* means comparison for the data presented in Fig. 6. For all tables $\alpha$ is 0.05, and significant *P*-values are highlighted in bold. A, *P*-values for means comparisons of initial $pH_i$ data. B, *P*-values for means comparisons of $\triangle pH_i$ data. C, *P*-values for means comparisons of intrinsic buffering power ($\beta_I$). D, *P*-values for means comparisons of $P_f$ data**

**6A Initial $pH_i$**

| cRNA | | $H_2O$ | hAQP5 | $H_2O$ | hAQP5 | $H_2O$ | hAQP5 | $H_2O$ |
|---|---|---|---|---|---|---|---|---|
| | hCA II (ng) | 'Tris' | 'Tris' | 1 | 1 | 10 | 10 | 100 |
| hAQP5 | 'Tris' | 0.701 | | | | | | |
| $H_2O$ | 1 | 0.971 | 0.997 | | | | | |
| hAQP5 | 1 | 0.512 | 1.00 | 0.978 | | | | |
| $H_2O$ | 10 | 1.00 | 0.481 | 0.877 | 0.309 | | | |
| hAQP5 | 10 | 0.703 | 1.00 | 0.998 | 1.00 | 0.484 | | |
| $H_2O$ | 100 | 1.00 | 0.569 | 0.925 | 0.385 | 1.00 | 0.572 | |
| hAQP5 | 100 | 0.997 | 0.972 | 1.00 | 0.898 | 0.970 | 0.973 | 0.987 |

**6B $\triangle pH_i$**

| cRNA | | $H_2O$ | hAQP5 | $H_2O$ | hAQP5 | $H_2O$ | hAQP5 | $H_2O$ |
|---|---|---|---|---|---|---|---|---|
| | hCA II (ng) | 'Tris' | 'Tris' | 1 | 1 | 10 | 10 | 100 |
| hAQP5 | 'Tris' | 1.00 | | | | | | |
| $H_2O$ | 1 | 1.00 | 1.00 | | | | | |
| hAQP5 | 1 | 0.679 | 0.909 | 0.757 | | | | |
| $H_2O$ | 10 | 0.980 | 1.00 | 0.991 | 0.994 | | | |
| hAQP5 | 10 | 0.0737 | 0.199 | 0.0995 | 0.902 | 0.460 | | |
| $H_2O$ | 100 | 1.00 | 1.00 | 1.00 | 0.748 | 0.990 | 0.0963 | |
| hAQP5 | 100 | 0.480 | 0.767 | 0.563 | 1.00 | 0.960 | 0.976 | 0.554 |

**6C $\beta_I$**

| cRNA | | $H_2O$ | hAQP5 | $H_2O$ | hAQP5 | $H_2O$ | hAQP5 | $H_2O$ |
|---|---|---|---|---|---|---|---|---|
| | hCA II (ng) | 'Tris' | 'Tris' | 1 | 1 | 10 | 10 | 100 |
| hAQP5 | 'Tris' | 0.982 | | | | | | |
| $H_2O$ | 1 | 0.995 | 1.00 | | | | | |
| hAQP5 | 1 | 1.00 | 0.869 | 0.934 | | | | |
| $H_2O$ | 10 | 0.876 | 0.326 | 0.434 | 0.984 | | | |
| hAQP5 | 10 | 0.504 | 0.0905 | 0.137 | 0.782 | 0.998 | | |
| $H_2O$ | 100 | 1.00 | 0.928 | 0.970 | 1.00 | 0.960 | 0.685 | |
| hAQP5 | 100 | 0.720 | 0.188 | 0.267 | 0.926 | 1.00 | 1.00 | 0.867 |

**6D $P_f$**

| cRNA | | $H_2O$ | hAQP5 | $H_2O$ | hAQP5 | $H_2O$ | hAQP5 | $H_2O$ |
|---|---|---|---|---|---|---|---|---|
| | hCA II (ng) | 'Tris' | 'Tris' | 1 | 1 | 10 | 10 | 100 |
| hAQP5 | 'Tris' | **$6.86 \times 10^{-8}$** | | | | | | |
| $H_2O$ | 1 | 1.00 | **$8.93 \times 10^{-7}$** | | | | | |
| hAQP5 | 1 | **$1.87 \times 10^{-6}$** | 1.00 | **$2.79 \times 10^{-7}$** | | | | |
| $H_2O$ | 10 | 1.00 | **$8.76 \times 10^{-7}$** | 1.00 | **$8.85 \times 10^{-7}$** | | | |
| hAQP5 | 10 | **$7.30 \times 10^{-8}$** | 0.988 | **$7.23 \times 10^{-8}$** | 0.996 | **$7.16 \times 10^{-8}$** | | |
| $H_2O$ | 100 | 1.00 | **$8.96 \times 10^{-7}$** | 1.00 | **$6.86 \times 10^{-8}$** | 1.00 | **$7.25 \times 10^{-8}$** | |
| hAQP5 | 100 | **$7.08 \times 10^{-8}$** | 1.00 | **$1.86 \times 10^{-6}$** | 1.00 | **$8.93 \times 10^{-7}$** | 0.999 | **$1.87 \times 10^{-6}$** |

**Statistics Table 8. For clarity within the figure panel we present tables of *P*-values for one-way ANOVA with Tukey's *post hoc* means comparison for the data presented in Fig. 8. For all tables $\alpha$ is 0.05, and significant *P*-values are highlighted in bold. A, *P*-values for means comparisons of initial $pH_i$ data. B, *P*-values for means comparisons of $\Delta pH_i$ data. C, *P*-values for means comparisons of $\beta_I$. D, *P*-values for means comparisons of $P_f$ data. Panels E–H provide descriptive statistics and means comparisons (see Methods, Statistical analysis) for comparisons of data represented by the grey, black, light-blue and dark-blue bars in Fig. 3 (–bCA) *vs*. Fig. 8 (+bCA). E, $\Delta pH_i$ during addition of $CO_2/HCO_3^-$. F, $\Delta pH_i$ during removal of $CO_2/HCO_3^-$. G, $(dpH_i/dt)_{Max}$ during addition of $CO_2/HCO_3^-$. H, $(dpH_i/dt)_{Max}$ during addition of $CO_2/HCO_3^-$**

*8A $\Delta pH_i$ $CO_2$ addition*

| cRNA | | $H_2O$ | hAQP5 | $H_2O$ |
|---|---|---|---|---|
| | hCA II (ng) | 'Tris' | 'Tris' | 1 |
| hAQP5 | 'Tris' | 1.00 | | |
| $H_2O$ | 1 | **0.00582** | **0.00663** | |
| hAQP5 | 1 | **$6.13 \times 10^{-4}$** | **$6.99 \times 10^{-4}$** | 0.757 |

*8B $\Delta pH_i$ $CO_2$ removal*

| cRNA | | $H_2O$ | hAQP5 | $H_2O$ |
|---|---|---|---|---|
| | hCA II (ng) | 'Tris' | 'Tris' | 1 |
| hAQP5 | 'Tris' | 0.996 | | |
| $H_2O$ | 1 | **0.0352** | 0.0560 | |
| hAQP5 | 1 | **0.00287** | **0.00478** | 0.674 |

*8C $(dpH_i/dt)_{Max}$ $CO_2$ addition*

| cRNA | | $H_2O$ | hAQP5 | $H_2O$ |
|---|---|---|---|---|
| | hCA II (ng) | 'Tris' | 'Tris' | 1 |
| hAQP5 | 'Tris' | **0.0331** | | |
| $H_2O$ | 1 | **$2.42 \times 10^{-6}$** | 0.00171 | |
| hAQP5 | 1 | **$2.33 \times 10^{-5}$** | 0.0192 | 0.707 |

*8D $(dpH_i/dt)_{Max}$ $CO_2$ removal*

| cRNA | | $H_2O$ | hAQP5 | $H_2O$ |
|---|---|---|---|---|
| | hCA II (ng) | 'Tris' | 'Tris' | 1 |
| hAQP5 | 'Tris' | 0.571 | | |
| $H_2O$ | 1 | **$1.42 \times 10^{-5}$** | **$2.49 \times 10^{-4}$** | |
| hAQP5 | 1 | **$7.08 \times 10^{-4}$** | **0.0135** | 0.318 |

*8E, $\Delta pH_S$ upon $CO_2$ addition ±bCA in bECF (*Fig. *3A vs*. Fig. *8A*)

| Descriptive stats. | n | Mean | SD | SEM | | | |
|---|---|---|---|---|---|---|---|
| bECF | 32 | 0.0689 | 0.0422 | 0.00746 | | | |
| bECF + bCA | 24 | 0.286 | 0.0641 | 0.0131 | | | |
| **Means comparison** | **Mean diff.** | **SEM** | ***Q*-value** | ***P*-value** | **$\alpha$** | **LCL** | **UCL** |
| bECF *vs*. bECF + bCA | 0.217 | 0.0142 | 21.6 | **$4.53 \times 10^{-8}$** | 0.0500 | 0.188 | 0.245 |

*8F, $\Delta pH_S$ upon $CO_2$ removal ±bCA in bECF (*Fig. *3B vs*. Fig. *8B*)

| Descriptive stats. | n | Mean | SD | SEM | | | |
|---|---|---|---|---|---|---|---|
| bECF | 32 | −0.0771 | 0.0487 | 0.00861 | | | |
| bECF + bCA | 24 | −0.386 | 0.0681 | 0.0139 | | | |
| **Means comparison** | **Mean diff.** | **SEM** | ***Q*-value** | ***P*-value** | **$\alpha$** | **LCL** | **UCL** |
| bECF *vs*. bECF + bCA | −0.309 | 0.0156 | 28.0 | **$4.53 \times 10^{-8}$** | 0.0500 | −0.340 | −0.278 |

(*Continued*)

**Statistics Table 8. (Continued)**

*8G, $(dpH_i/dt)_{Max}$ upon $CO_2$ addition $\pm$bCA in bECF (Fig. 3C vs. Fig. 8C)*

| Descriptive stats. | n | Mean | SD | SEM | | | |
|---|---|---|---|---|---|---|---|
| bECF | 32 | −0.00315 | 0.00236 | $4.18\times10^{-4}$ | | | |
| bECF + bCA | 24 | −0.00528 | 0.00225 | $4.60\times10^{-4}$ | | | |
| **Means comparison** | **Mean diff.** | **SEM** | **Q-value** | **P-value** | $\alpha$ | **LCL** | **UCL** |
| bECF *vs.* bECF + bCA | −0.00213 | $6.25\times10^{-4}$ | 4.83 | **0.00122** | 0.0500 | −0.00339 | $-8.81\times10^{-4}$ |

*8H, $(dpH_i/dt)_{Max}$ upon $CO_2$ removal $\pm$bCA in bECF (Fig. 3D vs. Fig. 8D)*

| Descriptive stats. | *n* | Mean | SD | SEM | | | |
|---|---|---|---|---|---|---|---|
| bECF | 32 | 0.00183 | 0.00125 | $2.22\times10^{-4}$ | | | |
| bECF + bCA | 24 | 0.00323 | 0.00161 | $3.29\times10^{-4}$ | | | |
| **Means comparison** | **Mean diff.** | **SEM** | **Q-value** | **P-value** | $\alpha$ | **LCL** | **UCL** |
| bECF *vs.* bECF + bCA | 0.00140 | $3.83\times10^{-4}$ | 5.17 | **$5.82\times10^{-7}$** | 0.0500 | $6.32\times10^{-4}$ | 0.00217 |

*Conclusions.* From the statistical analysis of the data summarized in Fig. 3*A,B* we conclude that – in the absence of both hCA II and bCA both for $CO_2/HCO_3^-$ addition and removal – expression of hAQP5 increases the magnitude of $\Delta pH_S$.

*Interpretation.* $\Delta pH_S$ (grey vs. black bars) in Fig. 3*A,B*; $\pm$hAQP5, –hCA II, –bCA. Expression of hAQP5 increases $P_{M,CO_2}$ in eqn (1).

**$dpH_i$/dt.** The green records in Fig. 2*A* and *B* show the $pH_i$ data that correspond to the $pH_S$ data presented above. The expression of hAQP5 does not produce a remarkable change in either the downward or upward $(dpH_i/dt)_{Max}$.

*Summary.* The grey and black bars in Fig. 3*C* show the mean downward $(dpH_i/dt)_{Max}$ for $CO_2/HCO_3^-$ addition, $\pm$hAQP5 – in the absence of both hCA II and bCA. The bars in Fig. 3*D* show the analogous data for $CO_2/HCO_3^-$ removal.

*Conclusions.* With either $CO_2/HCO_3^-$ addition or removal – in the absence of injected hCA II and extracellular bCA – the expression of hAQP5 does not significantly affect $(dpH_i/dt)_{Max}$.

*Interpretation.* $(dpH_i/dt)_{Max}$ (grey vs. black bars) in Fig. 3*C,D*; $\pm$hAQP5, –hCA II, –bCA. (1) Expression of hAQP5, even though it markedly increases $P_{M,CO_2}$ as indicated by the $\Delta pH_S$ signals in Fig. 3*A,B*, does not increase the magnitude of $(dpH_i/dt)_{Max}$. The reason, in part, is that – without hCA II – the cytosolic $CO_2$ hydration reaction (during $CO_2$ influx) and dehydration reaction (during $CO_2$ efflux) are rate-limiting, thereby limiting the transmembrane $CO_2$ gradient and choking the $CO_2$ fluxes (see eqn (2)). In the second half of Results we will see that the availability of $CO_2$ on the outer surface of the cell (replenished/consumed by bCA) also is rate-limiting under the conditions of Fig. 2*A,B*. (2) Thus

$(dpH_i/dt)_{Max}$ is a relatively insensitive indicator of $CO_2$ fluxes in the absence of bCA in the bECF. And (3) a contributing factor may be the large diameter of oocytes ($\sim$1.2 mm); a prolonged time for diffusion to/from the centre of the cell limits the dynamic range of $(dpH_i/dt)_{Max}$. Note that in the hands of Nakhoul et al. (1998) and Cooper and Boron (1998) AQP1 expression did not produce a statistically significant effect on $(dpH_i/dt)_{Max}$ under similar experimental conditions (i.e. in the absence of injected CA enzyme and with vitelline membrane intact).

## $\pm$hAQP5, $\Delta$hCA II, –bCA[8]: effects on $\Delta pH_S$ and $(dpH_i/dt)_{Max}$

To explore the interaction between hAQP5 and intracellular CA ($CA_i$) in the diffusion of $CO_2$ we injected oocytes on Day 1 with either '$H_2O$' or '$H_2O$ + cRNA encoding hAQP5' and then, on Day 4, we injected oocytes with either 'Tris' as a control or 'Tris + hCA II' at hCA II levels of 1, 10 or 100 ng.

**$pH_s$.** The red record in Fig. 2*C* shows the $pH_S$ data for a representative '$H_2O$ + 1 ng hCA II' oocyte and in Fig. 2*D* the comparable data for an 'hAQP5 + 1 ng hCA II' oocyte. Comparing the $pH_S$ records between Fig. 2*C* vs. Fig. 2*A* we see that – in the absence of hAQP5 – the addition of 1 ng hCA II appears to increase the magnitude of the $pH_S$ transient slightly. Musa-Aziz et al. (2014*a*) had previously observed that hCA II – presumably with a higher specific activity – significantly increases the magnitude of $\Delta pH_S$ with both $CO_2/HCO_3^-$ addition and removal. If we compare Fig. 2*D* vs. Fig. 2*B*, we see that – in the presence

---

[8]Here, '$\Delta$' indicates that in this section we compare a series of injected doses of hCA II; '–bCA' indicates that extracellular bCA was absent in all experiments.

of hAQP5 – the injection of 1 ng hCA II has a much larger effect on the $pH_S$ transient. Finally comparing Fig. 2*D vs.* Fig. 2*C* and Fig. 2*B vs.* Fig. 2*A* we see that the hAQP5 expression has a far greater effect on augmenting the magnitude of $\Delta pH_S$ – both with $CO_2/HCO_3^-$ addition and removal – with the injection of 1 ng hCA II than with no added hCA II.

Note that in the above comparisons the $pH_S$ electrode is 'trans' – or on the opposite side of the membrane – with respect to the hCA II that we added to the cytosol, a concept developed by Musa-Aziz et al. (2014a, 2014b) and Occhipinti et al. (2014). In such cases one can conclude by intuition – but supported by mathematical simulations – that the trans-side increase in $\Delta pH_S$ is indicative of an increased $CO_2$ flux when hCA II is present in the cytosol. Thus based on Fig. 2*A–D* we can conclude that both the expression of hAQP5 alone (a large effect) and the injection of hCA II alone (a small effect), and especially

the two in combination, increase $CO_2$ fluxes into/out of oocytes.

Examining the effects of injecting larger amounts of hCA II we continue to see – in the absence of hAQP5 – very little effect on the $pH_S$ transients (Fig. 2*E,G*) compared to the 'Tris' control (Fig. 2*A*). In the presence of hAQP5 the higher levels of injected hCA II (Fig. 2*F,H*) continue to enhance the $pH_S$ transients greatly compared to the 'AQP5 + Tris' control (Fig. 2*B*), but the stimulatory effect is little more than with our 1 ng dose of hCA II (Fig. 2*D*).

In their earlier work Musa-Aziz et al. (2014a) observed a baseline $\Delta pH_S$ (i.e. no injected CA) of slightly more than 0.04, whereas in the present study our baseline $\Delta pH_S$ is only ~0.03. Thus the baseline values of $CO_2$ permeability, surface CA activity or cytosolic CA activity in that previous study may have been somewhat greater than in the present study. Moreover Musa-Aziz et al.

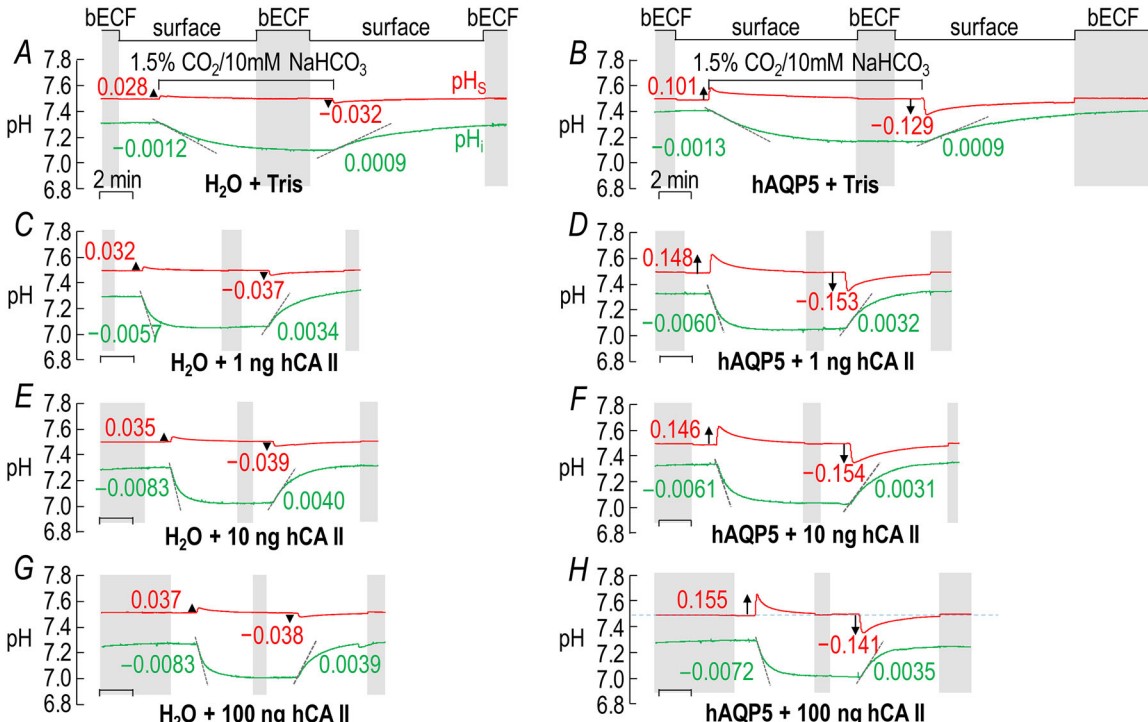

**Figure 2. Representative $pH_S$ and $pH_i$ recordings: ±hAQP5, $\Delta$cytosolic hCA II, all in the absence of extracellular bCA (i.e. −bCA)**
A, 'H$_2$O + Tris'. We injected this oocyte with H$_2$O (as a control for cRNA encoding hAQP5) on Day 1 and with 'Tris' (as a control for hCA II) on Day 4. B, 'hAQP5 + Tris'. We injected this oocyte on Day 1, with cRNA encoding hAQP5, and then on Day 4 with 'Tris'. C, 'H$_2$O + 1 ng hCA II'. Similar to panel A, except that on Day 4, we injected the oocyte with hCA II enzyme dissolved in 'Tris'. D, 'hAQP5 +1 ng hCA II'. Similar to panel B, except that on Day 4, we injected the oocyte with hCA II enzyme dissolved in 'Tris'. E, 'H$_2$O + 10 ng hCA II'. F, 'hAQP5 + 10 ng hCA II'. G, 'H$_2$O + 100 ng hCA II'. H, 'hAQP5 + 100 ng hCA II'. The '$\Delta$' in the title and elsewhere in the paper implies that we choose among several levels of injected hCA II. The numbers in red are $\Delta pH_S$ and those in green are $(dpH_i/dt)_{Max}$ for the oocyte presented here. The stair-step line at the top of A and B indicates the position of the $pH_S$ electrode. The vertical shaded bars indicate times during which the $pH_S$ electrode is in the bulk extracellular fluid (bECF) for recalibration; at other times the $pH_S$ electrode dimples the cell surface for actual $pH_S$ measurements. The colours of the rectangles that surround the panel labels (e.g. 'H$_2$O + Tris') correspond to the colours of the bars in Fig. 3 and later figures.

(2014a) found that injecting 300 ng of recombinant hCA II approximately doubled the $\Delta pH_S$, whereas in the present study injecting 100 ng hCA II enzyme (a dose at which the $\Delta pH_S$ seemingly had already plateaued) increased $\Delta pH_S$ only by about one-third (compare Fig. 2*G vs. A*). We presume that our commercially obtained hCA II, purified from red blood cells (RBCs), had a lower specific activity than the recombinant hCA II in the previous study. We

abandoned attempts at injecting >100 ng/oocyte because these higher doses seemed to have deleterious effects on the oocytes, perhaps because of contaminants.

*Summary.* The lighter-coloured blue, red and green bars in Fig. 3*A* display the mean $\Delta pH_S$ data for $CO_2/HCO_3^-$ addition with increasing doses of hCA II, all in the absence of hAQP5 and bCA. The darker bars in Fig. 3*A* summarize

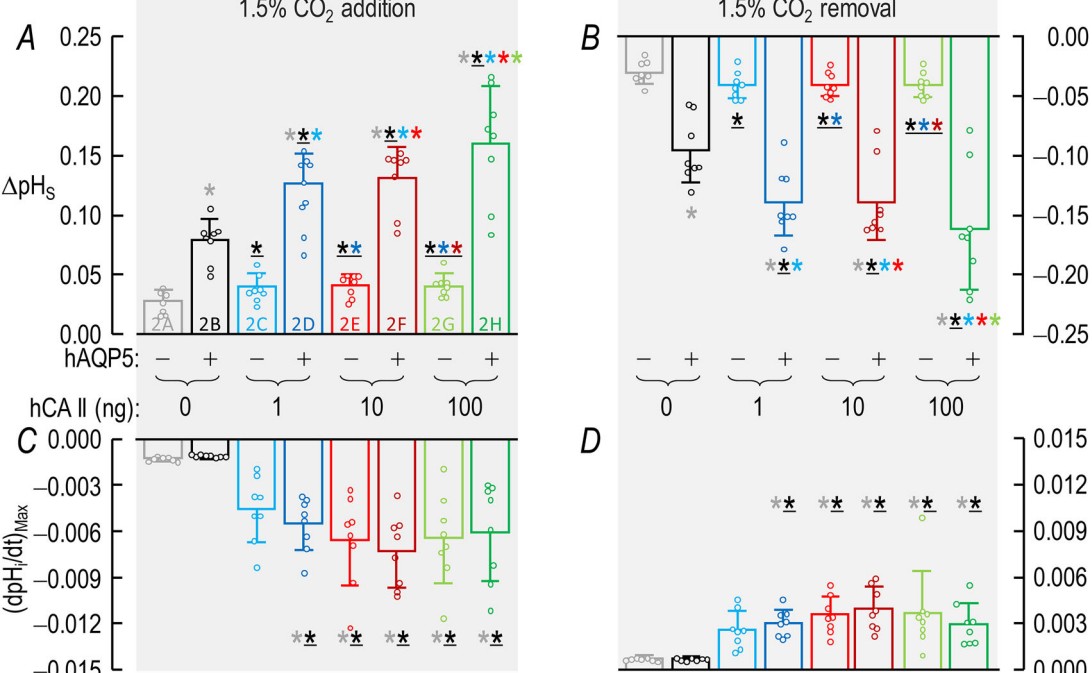

**Figure 3. Summary of $\Delta pH_S$ and $(dpH_i/dt)_{Max}$ data from experiments like those in Fig. 2: ±hAQP5, $\Delta$cytosolic hCA II, all in the absence of extracellular bCA (i.e. −bCA)**
A, summary of $\Delta pH_S$ upon addition of 1.5% $CO_2$/10 mM $HCO_3^-$. We computed individual $\Delta pH_S$ values as outlined in 'Methods' > 'Electrophysiological measurements' > 'Calculation of $\Delta pH_S$'. B, summary of $\Delta pH_S$ upon removal of 1.5% $CO_2$/10 mM $HCO_3^-$. C, summary of $(dpH_i/dt)_{Max}$ upon addition of 1.5% $CO_2$/10 mM $HCO_3^-$. We computed individual $\Delta pH_S$ values as outlined in 'Methods' > 'Electrophysiological measurements' > 'Calculation of "maximal" $dpH_S/dt$'. D, summary of $(dpH_i/dt)_{Max}$ upon removal of 1.5% $CO_2$/10 mM $HCO_3^-$. The '$\Delta$' in the title and elsewhere in the paper implies that we choose among several levels of injected hCA II. Data are presented as mean ± SD. '−' and lighter-coloured bars indicate that we injected oocytes with $H_2O$; '+' and darker-coloured bars, with cRNA encoding hAQP5 on Day 1. '0', '1', '10' and '100' indicate the amount of hCA II (ng) injected into oocytes on Day 4. Grey/black and light-/dark-blue, red and green pairs of bars indicate eight groups of oocytes. These colours correspond to the colours of the rectangles that surround panel labels (e.g. '$H_2O$ + Tris') in Fig. 2. Statistical star/bar conventions: for each panel we used an ANOVA (see Methods[7]) to compare the data underlying each bar with that of every other bar in that panel (i.e. 7! total comparisons for each panel). The bars are meant to be read from right to left, starting with the darker of the two green bars. A coloured star above a bar indicates that the data underlying the light-coloured data to the left differ significantly (see Statistics Table 3 for *P*-values). Thus the light-grey star above the dark-green bar indicates that the dark-green data differ significantly from the data represented by the light-grey bar. The black star (underscored to emphasize that it is darker than grey) indicates that the dark-green data also differ significantly from the data represented by the black bar. The light-blue, light-red and light green stars indicate statistically significant differences between the dark-green data and those of the bars with the same colour as the stars. Similarly the light-green data differ significantly from the black, dark-blue and dark-red data to the left. To reduce the number of displayed stars by half we do not use stars to indicate the reverse statistical significance (i.e. bars to the right). Thus even though the light-grey bar has no stars its underlying data differ significantly from bars to its right with grey stars: the black, dark-blue, dark-red and dark-green bars. The absence of a star in the right-to-left progression indicates a lack of statistical significance. Thus because the dark-green bar lacks dark-red and dark-blue stars, the data underlying the dark-green bar do not differ significantly from those underlying the dark-red or dark-blue bars. The complete statistical analyses are presented in Table A3*A-D*

comparable data but in the presence of hAQP5. For $CO_2/HCO_3^-$ removal Fig. 3*B* reveals patterns that are similar to those in Fig. 3*A*.

*Conclusions.* For oocytes examined in the absence of bCA both for the addition and removal of $CO_2/HCO_3^-$: (a)[9] expressing hAQP5 (darker *vs.* lighter bars) increases the magnitude of $\Delta pH_S$ at every level of injected hCA II. (b) In the absence of hAQP5 (lighter bars) injecting oocytes with increasing amounts of hCA II, despite a modest upward trend, does not have a statistically significant effect on $\Delta pH_S$ magnitudes, neither for $CO_2/HCO_3^-$ addition nor removal. (c) In the presence of hAQP5 (darker bars), injection of increasing amounts of hCA II tends to cause graded increases of $\Delta pH_S$ magnitudes, both for $CO_2$ influx and efflux. Relative to no injected hCA II the effects reach statistical significance at all three levels of injected hCA II, although these three bars are not significantly different from each other.

*Interpretation.* $\Delta pH_S$ (colourful bars) in Fig. 3*A,B*; $\pm$hAQP5, $\Delta$hCA II, –bCA. (1) In the absence of hAQP5 (four lighter bars) $P_{M,CO_2}$ is the major rate-limiting factor in both directions of $CO_2$ diffusion and at all hCA II levels. Expression of hAQP5 augments $P_{M,CO_2}$ and thereby increases $\Delta pH_S$ (four darker bars). (2) During $CO_2/HCO_3^-$ addition in control cells (–hAQP5, –hCA II; grey bars) disposal of incoming $CO_2$ (in this case by cytosolic CA) is borderline rate-limiting (i.e. with greater $CA_i$ activity the flux would have been modestly greater), as noted previously by Musa-Aziz et al. (2014a). Conversely during the subsequent $CO_2/HCO_3^-$ removal replenishment of outgoing $CO_2$ (here by hCA II), likewise, is borderline rate-limiting. And (3) as we will see below in our presentation of data on bCA in the bECF for control cells (grey bars) the replenishment of $CO_2$ on the extracellular surface is also rate-limiting during $CO_2$ influx. Conversely disposal of exiting $CO_2$ at the cell surface during $CO_2$ efflux is also rate-limiting. (4) The greater effect of expressing hAQP5 in oocytes injected with 1 ng of hCA II (light- *vs.* dark-blue bars) than in oocytes injected only with 'Tris' (grey *vs.* black bars) is an example of synergism.

**dpH_i/dt.** The green records in Fig. 2*C,E,G* and *D,F,H* show the $pH_i$ data that correspond to the $pH_S$ data presented above, with incremental amounts of injected hCA II without hAQP5 (left side of Fig. 2) and with hAQP5 (right side). We observe that injecting even 1

ng hCA II into the cytosol produces a striking increase in $(dpH_i/dt)_{Max}$, both for $CO_2/HCO_3^-$ application and removal. Increasing the injected hCA II to 10 ng further increases the downward $(dpH_i/dt)_{Max}$ but has only a modest effect on the upward $(dpH_i/dt)_{Max}$ in this example. Increasing injected hCA II to 100 ng has no additional effect. Comparing the left and right sides of Fig. 2 we see that hAQP5 expression seemingly has little effect on downward or upward $(dpH_i/dt)_{Max}$ regardless of the amount of injected hCA II.

Note that in the above $(dpH_i/dt)_{Max}$ analyses the $pH_i$ electrode is 'cis' – or on the same side of the membrane – with respect to the hCA II that we added to the cytosol, again, a concept developed by Musa-Aziz et al. (2014a, 2014b) and Occhipinti et al. (2014). In such cases one cannot use intuition to arrive at conclusions that the cis-side acceleration of a $pH_i$ change by hCA II is indicative of an increased $CO_2$ flux. The reason is that – near the intracellular surface of the plasma membrane – the added hCA II greatly accelerates $H^+$ formation during $CO_2$ influx and $H^+$ consumption during $CO_2$ efflux. The $pH_i$ electrode directly senses the changes in $[H^+]$, which only indirectly reflect $CO_2$ fluxes. However previous mathematical simulations predict that the $CO_2$ fluxes must have increased under these conditions (Musa-Aziz et al., 2014a; Occhipinti et al., 2014).

*Summary.* The lighter/darker pairs of bars (–/+ hAQP5) in Fig. 3*C,D* précis the mean $(dpH_i/dt)_{Max}$ data for $CO_2/HCO_3^-$ addition and removal.

*Conclusions.* Both for $CO_2/HCO_3^-$ addition and removal: (a) expression of hAQP5 does not significantly affect $(dpH_i/dt)_{Max}$ at any level of injected hCA II (four sets of lighter *vs.* darker bars). And (b) in the absence of hAQP5 (four lighter bars) increasing amounts of injected hCA II cause $(dpH_i/dt)_{Max}$ values to trend towards greater magnitudes, although the effects do not reach statistical significance until 10 ng (light red) and 100 ng (light green).

*Interpretation.* $(dpH_i/dt)_{Max}$ (colourful bars) in Fig. 3*C,D*; $\pm$hAQP5, $\Delta$hCA II, –bCA. (1) Because the injected hCA II is 'cis' to the $pH_i$ electrode, we cannot draw intuitive conclusions about $CO_2$ fluxes as we compare the grey $(dpH_i/dt)_{Max}$ bar (i.e. –hAQP5) to the other three light-coloured bars (i.e. increasing amounts of hCA II) or as we compare the black bar (i.e. +hAQP5) to the other three darker-coloured bars. (2) Expression of hAQP5 (compare lighter *vs.* darker blue, red and green bars) does not increase $(dpH_i/dt)_{Max}$ magnitudes. As noted in our interpretation of $\Delta pH_S$ data

---

[9]In lists of points in the Summary and Conclusion we use (a), (b), (c) …. The (a) in one list corresponds to the (a) in another list, etc. However in the Interpretation, we list points as (1), (2), (3) … because the interpretations do not necessarily correspond to lettered points in a one-to-one fashion.

immediately above[10] this lack of effect probably reflects limited availability of $CO_2$ at the extracellular surface during influx and disposal of $CO_2$ at the extracellular surface during efflux. Such choking could be mitigated by introducing an extracellular carbonic anhydrase ($CA_o$), as illustrated by the grey *vs.* black bars in Fig. 8*C*. As already mentioned[11] a compounding factor may be the large diameter of the oocyte. (3) Nevertheless we know that expression of hAQP5 leads to a sizeable increase in $P_{M,CO_2}$ because of the parallel $\Delta pH_S$ data summarized by the lighter/darker pairs of bars in Fig. 3*A,B*.

**$V_m$.** Figure *A*1 (in the Appendix) summarizes the end-of-experiment $V_m$ data for the oocytes in Fig. 3. Note that oocytes expressing hAQP5 are modestly depolarized compared to the $H_2O$-injected control oocytes.

### ±hAQP5, ΔhCA II, –bCA: effects on $pH_S$ relaxation

**Theoretical considerations.** Let us assume that the oocyte is a sphere, containing only an aqueous pH buffer, surrounded by a membrane permeable only to $CO_2$ and initially devoid of $CO_2/HCO_3^-$. If we now expose the oocyte to a $CO_2/HCO_3^-$ solution, we could compute – given $[CO_2]_o$, initial $pH_i$, buffering power and cell volume – the net number of $CO_2$ molecules that would diffuse into the cell before the system came into equilibrium. The speed of this equilibration depends on the parameters that contribute to the description of Fick's law in eqns (1) and (2), including $[CO_2]_{os}$, $[CO_2]_{is}$ and $P_{M^*,CO_2}$.

In their study of the impact of cytosolic CA II and CA IV (predominantly extracellular) on $CO_2$ equilibration Musa-Aziz et al. (2014a, 2014b) noted that, following the rapid upswing in $pH_S$ triggered by $CO_2/HCO_3^-$ application (or the rapid downswing in $pH_S$ triggered by $CO_2/HCO_3^-$ removal), (1) $pH_S$ decayed following an approximately single-exponential (SExp) time course, and that the time constant ($\tau$) of this decay decreased markedly either with (2) injection of recombinant hCA II into the cytosol (see fig. 13 in Musa-Aziz et al. (2014a) or (3) expression of hCA IV (see fig. 21 in Musa-Aziz et al. (2014b). Moreover mathematical simulations based on a reaction-diffusion model (Somersalo et al., 2012) corroborated nearly all of the essential observations (Occhipinti et al., 2014). In other words CA II and CA IV increased the speed (reflected by the rate constant = $1/\tau$)

of $CO_2$ equilibration, and thus $1/\tau$ is an indirect measure of the transmembrane $CO_2$ flux.

In the present study we note that the $pH_S$ relaxations do not so nearly approximate an SExp decay as in the work by Musa-Aziz et al. (2014a, 2014b), probably due to (1) minor differences in the tip of the $pH_S$ electrodes, (2) the angles with which the $pH_S$ electrode contacted the cell surface and (3) the precise location of the $pH_S$ electrode on the oocyte surface (see Fig. 1 insets in Musa-Aziz et al. (2014a). Rather we were able to obtain excellent fits of the present $pH_S$-decay data using a DExp function (i.e. one with two, not one, time constants). Note also that the aforementioned mathematical simulations suggest that the $pH_S$ decays should not be perfectly exponential in the first place (Musa-Aziz et al., 2014a, 2014b; Occhipinti et al., 2014). We regard the subtle differences in $pH_S$-decay waveforms between the previous and present work as the result of a divergence in technique that requires appropriate flexibility in data analysis. In the present study, where the $pH_S$ decay is not a single exponential, we chose to use the maximal initial rate of $pH_S$ relaxation as a surrogate for the single-exponential $1/\tau$.

**Exemplar data.** Figure 4*A* reproduces the $pH_S$ time course for $CO_2/HCO_3^-$ addition in Fig. 2*B* (i.e. 'hAQP5 + Tris'). The jittery black record in Fig. 4*B* shows the time course of the actual $pH_S$ relaxation, beginning at a local time ($t_{Local}$) of 0, which we judged by eye to be the time of the fastest $pH_S$ decay. The smooth orange curve is the result of a DExp curve fit. The dashed orange line is the tangent to the DExp function, evaluated at $t_{Local} = 0$. The blue curve in Fig. 4*C* is a plot of $dpH_S/dt$, computed as the derivative of the best-fit DExp function, *vs.* $pH_S$. Here time flows from right to left. The break in slopes at $pH_S \cong 7.54$ is the result of the second slower exponential process. The dashed blue line is the derivative of the DExp function, again calculated at $t_{Local} = 0$, but now plotted as a function of $pH_S$. Although the choice of $t_{Local} = 0$ is subject to human error, the effect of misjudgement is likely to be minimal inasmuch as the DExp fit encompasses so many points, and because both the initial time course and the initial plot of $dpH_S/dt$ *vs.* $pH_S$ are nearly linear.

Figure 4*D–F* is analogous to the top row of panels, except that here we analyse the $CO_2/HCO_3^-$-removal step in Fig. 2*B*. Note that all of the $pH_S$ records are inverted, and that, in Fig. 4*F*, time runs from left to right.

*Summary.* The layout of the synopses in Fig. 5*A,B* for $(dpH_S/dt)_{Max}$ data – $CO_2$ addition/removal, ±hAQP5, with increasing amounts of injected hCA II – are the same as in Fig. 3*A* for the $\Delta pH_S$ data.

*Conclusions.* Both for $CO_2/HCO_3^-$ addition and removal: (a) in the absence of hCA II expression of hAQP5 (grey *vs.* black bars) increases the magnitude of $(dpH_S/dt)_{Max}$, and the difference in mean values

---

[10] See 'Results' > '±hAQP5, ΔhCA II, –bCA: effects on $\Delta pH_S$ and $(dpH_i/dt)_{Max}$' > '$\Delta pH_S$' > 'Interpretation: $\Delta pH_S$ (colourful bars) in Figure 3*A,B*; ±hAQP5, ΔhCA II, –bCA'.

[11] See 'Results' > '±hAQP5 in absence of exogenous CAs: effects on $\Delta pH_S$ and $(dpH_i/dt)_{Max}$' > '$(dpH_i/dt)_{Max}$' > 'Interpretation: $(dpH_i/dt)_{Max}$ (grey *vs.* black bars) in Figure 3*C,D*; ±hAQP5, –hCA II, –bCA'.

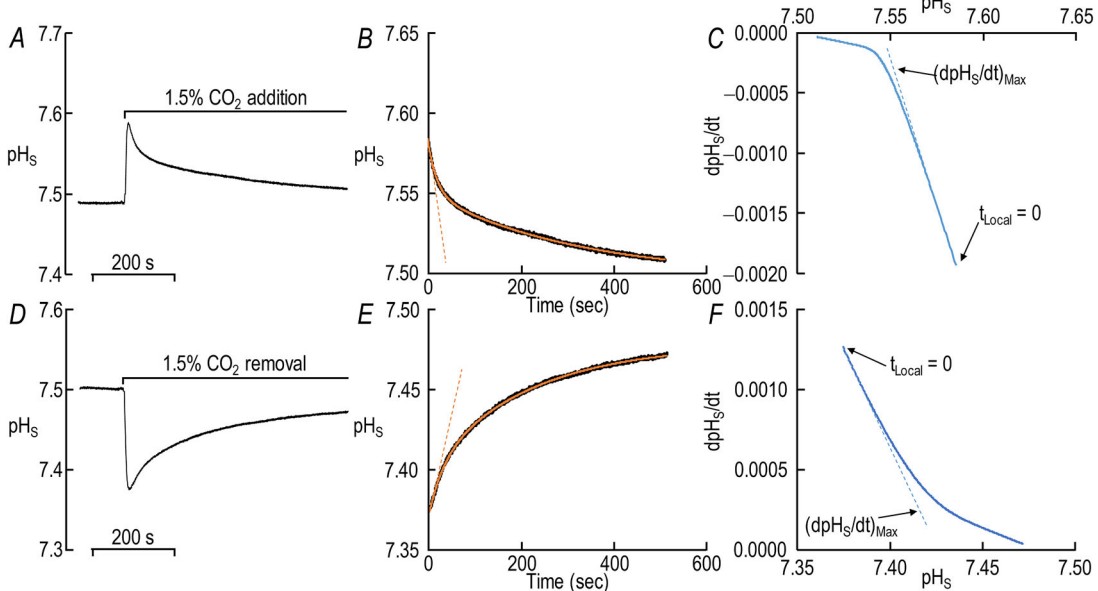

**Figure 4. Analysis of $(dpH_S/dt)_{Max}$, the magnitude of maximal rate of relaxation of $pH_S$**

A, time course of surface pH ($pH_S$) during addition of 1.5% $CO_2$/10 mM $HCO_3^-$. This is a reproduction – with a magnified *y*-axis – of the red record (during $CO_2$ addition) in Fig. 2B. B, detail of panel A, showing only the $pH_S$ relaxation, beginning from the time ($t_{Local} = 0$ s) of fastest $pH_S$ descent. The jittery black record is the actual $pH_S$ recording; the smooth orange curve is the result of a double-exponential (DExp) curve fit. The dashed red line has a $dpH_S/dt$-*vs.*-time slope that is the derivative of the fitted DExp function at $t_{Local} = 0$ s. C, dependence of $dpH_S/dt$ on $pH_S$ during addition of $CO_2$/$HCO_3^-$ application. We obtained $dpH_S/dt$ by computing the derivative of the DExp best-fit function at the time that corresponds to the indicated $pH_S$. Note that real time runs from right to left. The dashed blue line has a $dpH_S/dt$-*vs.*-$pH_S$ slope that corresponds to the value at $t_{Local} = 0$ s. A perfect single-exponential fit of $pH_S$ *vs.* time would have produced a straight line in this $dpH_S/dt$-*vs.*-$pH_S$ plot. Thus the break in the blue curve near $pH_S = 7.54$ is the demarcation between the dominance of a rapid/initial process and a slower/later process. D, time course of $pH_S$ during removal of 1.5% $CO_2$/10 mM $HCO_3^-$. This is a reproduction – with a magnified *y*-axis – of the red record (during $CO_2$ removal) in Fig. 2B. E, detail of panel D, processed as in panel B. F, dependence of $dpH_S/dt$ on $pH_S$ during $CO_2$/$HCO_3^-$ removal, processed as in panel C. Note that real time runs from left to right.

is statistically significant. The same is true for the expression of hAQP5 at each of the increasing hCA II levels (compare light *vs.* dark blue, red, and green bars). (b) In the absence of hAQP5, injecting hCA II in ever-greater amounts (lighter bars) causes $(dpH_S/dt)_{Max}$ to trend upward but is without a statistically significant effect on $(dpH_S/dt)_{Max}$. And (c) in the presence of hAQP5, injecting increasing amounts of hCA II tends to produce ever-greater magnitudes of $(dpH_S/dt)_{Max}$, with several of the differences reaching statistical significance, especially for $CO_2$/$HCO_3^-$ addition. And (d) the effects on $(dpH_S/dt)_{Max}$ are in the same direction but statistically more robust than for $\Delta pH_S$ in Fig. 3A,B.

Interpretation: $(dpH_S/dt)_{Max}$ in Fig. 5A,B; ±hAQP5, ΔhCA II, –bCA. Our analysis of these data is similar to those for $\Delta pH_S$ in sections above.[12]

## ±hAQP5, ΔhCA II, –bCA: effects on other oocyte parameters

**Initial $pH_i$.** Figure 6A summarizes the initial $pH_i$ values of oocytes subjected to the eight protocols of Fig. 2. All of the mean values are within 0.1 pH units of each other. Statistical analyses reveal no significant differences.

**$\Delta pH_i$.** Figure 6B summarizes the $\Delta pH_i$ values – that is, the decreases in $pH_i$ – elicited by application of 1.5% $CO_2$/NaHCO$_3$ for the eight protocols. These values are all well within 0.1 of each other, and statistical analyses show no statistically significant differences among the eight $\Delta pH_i$ mean values.

**$\beta_I$.** Figure 6C summarizes the intrinsic buffering power values of the eight groups of oocytes. Statistical analyses show no significant differences among the groups. The

---

[12]See Results > '±hAQP5 in absence of exogenous CAs: effects on $\Delta pH_S$ and $(dpH_i/dt)_{Max}$' > '$pH_S$' > 'Interpretation: $\Delta pH_S$ (grey *vs.* black bars) in Fig. 3A,B; ±hAQP5, –hCA II, –bCA'. Also see Results > '±hAQP5, ΔhCA II, –bCA: effects on $\Delta pH_S$ and $(dpH_i/dt)_{Max}$' > '$pH_S$'

> 'Interpretation: $\Delta pH_S$ (colourful bars) in Fig. 3A,B; ±hAQP5, –hCA II, –bCA'.

modest downward trend from the leftmost bars to the rightmost bars may reflect a faster equilibration of $CO_2$ within oocytes with higher injected levels of hCA II, especially with hAQP5 expression. With a slower $CO_2$ equilibration (see Fig. 2*A*) the measured $\Delta pH_i$ would tend to underestimate the value that we would have obtained at $t = \infty$ and thus lead to an artificially elevated computed $\beta_I$ value.

**$P_f$.** After $pH_S$ and $pH_i$ were monitored we performed the $P_f$ assay with each oocyte. Figure 6*D* summarizes our $P_f$ data. The injection of cRNA encoding hAQP5 significantly increases $P_f$ for each of the four groups of oocytes previously injected with no or increasing amounts of hCA II (compare lighter *vs.* darker bars in blue, red and green). Thus we can conclude that the oocytes express hAQP5 that traffics normally to the plasma membrane. Because the increase in $P_f$ induced by hAQP5 was virtually the same for the four doses of hCA II, it is most likely that the CA has no effect on the monomeric pores that are responsible for the osmotic water permeability of hAQP5.

### ±hAQP5, –hCA II, +bCA: effects on $\Delta pH_S$ and $(dpH_i/dt)_{Max}$

In the final four sections of Results we examine oocytes studied in the presence of extracellular bCA and thereby explore the functional interaction between hAQP5 and $CA_o$ in the transmembrane diffusion of $CO_2$.

In this first of the final four sections we study oocytes in the absence of injected hCA II. On Day 1 we injected oocytes with either 'H₂O' or 'H₂O + cRNA' encoding 'hAQP5' and then on Day 4 we injected all oocytes with 'Tris'. During the experiment we augment the extracellular solutions with 0.1 mg/ml of bCA, the same level as used previously by Musa-Aziz et al. (2014b). This protocol is identical to that of Fig. 2*A*,*B* (–bCA) except for the presence of bCA.

**$pH_S$.** The red record in Fig. 7*A* shows the $pH_S$ data for a representative 'H₂O + Tris' oocyte and in Fig. 7*B* the comparable data for an 'hAQP5 + Tris' oocyte. Notice that the $pH_S$ transients in Fig. 7*A* (+bCA, extracellular) are many fold larger – both with $CO_2/HCO_3^-$ addition and removal – than their counterparts in Fig. 2*A* (–bCA). Musa-Aziz et al. (2014b) had previously made a similar observation with extracellular hCA II (compare their figs. 13g and 15g).

*Theoretical considerations.* Note that in comparing $pH_S$ transients in Fig. 7*A* *vs.* Fig. 2*A* the $pH_S$ electrode is 'cis' to the added bCA in Fig. 7. A major reason that bCA increases the magnitude of $\Delta pH_S$ is that – near the extracellular surface of the plasma membrane – the added bCA greatly accelerates $H^+$ consumption during $CO_2$ influx (see Fig. 1*A*) and $H^+$ production during $CO_2$ efflux (see Fig. 1*B*). Although the $pH_S$ electrode directly senses changes in $[H^+]_S$, these only indirectly reflect $CO_2$ fluxes. Thus by comparing ±bCA (cis to the $pH_S$ electrode) we can reach no intuitive conclusions about the possible

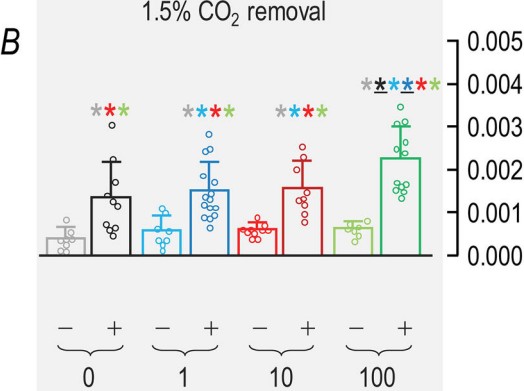

**Figure 5. Summary of $(dpH_S/dt)_{Max}$ data from experiments like those in Fig. 2: ±hAQP5, Δcytosolic hCA II, all in the absence of extracellular bCA (i.e. –bCA)**
A, summary of $(dpH_S/dt)_{Max}$ upon addition of 1.5% $CO_2$/10 mM $HCO_3^-$. We computed individual $(dpH_S/dt)_{Max}$ values as described in Fig. 4. B, summary of $(dpH_S/dt)_{Max}$ upon removal of 1.5% $CO_2$/10 mM $HCO_3^-$. The 'Δ' in the title and elsewhere in the paper implies that we choose among several levels of injected hCA II. Data are presented as mean ± SD. '−' and lighter-coloured bars indicate that we injected oocytes with H₂O; '+' and darker-coloured bars, with cRNA encoding hAQP5 on Day 1. '0', '1', '10' and '100' indicate the amount of hCA II (ng) injected into oocytes on Day 4. Grey/black and light-/dark-blue, red and green pairs of bars indicate eight groups of oocytes. These colours correspond to the colours of the rectangles that surround panel labels (e.g. 'H₂O + Tris') in Fig. 2. Statistical star/bar conventions. See legend of Fig. 3 for a description of statistical significance. See Statistics Table 5 for *P*-values. The complete statistical analyses are presented in Table A5*A* & *B*.

effect of bCA on transmembrane $CO_2$ fluxes from $pH_S$ data. However previous mathematical simulations predict that the $CO_2$ fluxes must have increased under these conditions (Musa-Aziz et al., 2014b; Occhipinti et al., 2014), even though in some cases the effects may be too small to measure.

Exemplar data in Fig. 7*A* vs. Fig. 7*B*. The $pH_S$ records show that in the presence of extracellular bCA the expression of hAQP5 does not produce an apparent increase in the magnitudes of the $pH_S$ transients. This apparent lack of effect is quite different from the large fractional increases in $\Delta pH_S$ magnitudes that we observed in the absence of extracellular bCA (i.e. Fig. 2*B* vs. Fig. 2*A*), where the $\Delta pH_S$ magnitudes are so much smaller.

*Summary.* The grey and black bars in Fig. 8*A* show the mean $\Delta pH_S$ data for $CO_2/HCO_3^-$ addition, ±hAQP5, all in the absence of hCA II but presence of extracellular bCA. Fig. 8*B* shows corresponding data for $CO_2/HCO_3^-$ removal.

*Conclusions.* Considering oocytes examined in the presence of bCA: in the absence of hCA II expression of hAQP5 (black *vs.* grey bars) does not significantly affect $\Delta pH_S$, either for $CO_2/HCO_3^-$ addition or removal.

*Interpretation.* $\Delta pH_S$ (grey vs. black bars) in Fig. 8*A,B*: ±hAQP5, –hCA II, +bCA. (1) bCA promotes large transmembrane $CO_2$ gradients. During $CO_2/HCO_3^-$ addition bCA maintains relatively high $CO_2$ levels near the extracellular face of the membrane and markedly increases $CO_2$ influxes. During the subsequent $CO_2/HCO_3^-$ removal bCA maintains relatively low $CO_2$ levels at the cell surface. Thus even in the absence of hAQP5 (grey bars) the $CO_2$ fluxes enabled by bCA are very high. (2) In parallel in a 'cis-side' effect, bCA greatly increases the magnitudes of $\Delta pH_S$, as we can see by comparing Fig. 8*A,B* vs. Fig. 3*A,B*. Thus during $CO_2/HCO_3^-$ addition bCA enhances $H^+$ consumption, whereas during $CO_2/HCO_3^-$ removal bCA enhances $H^+$

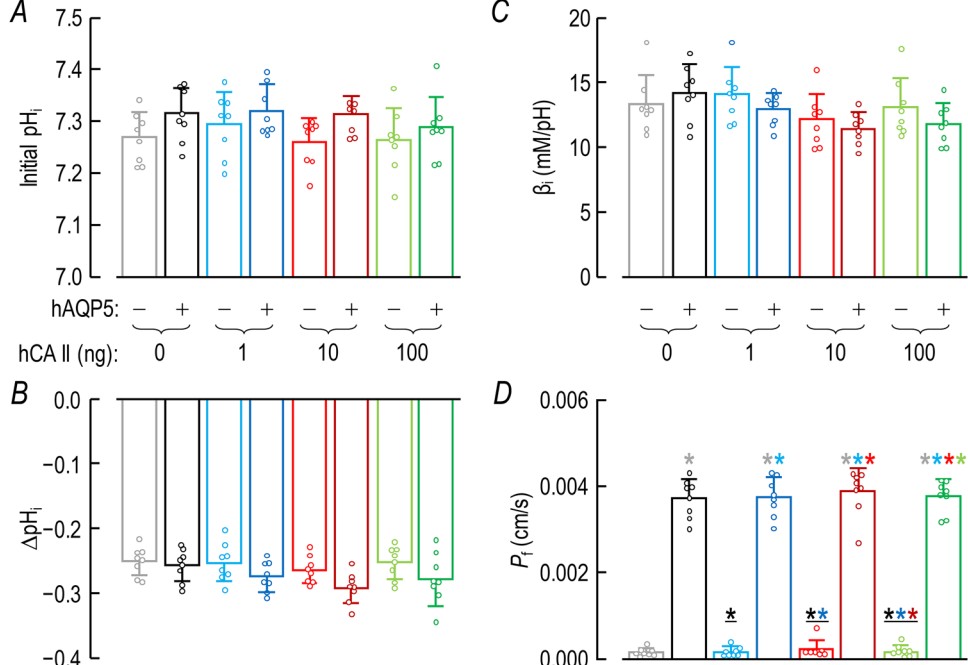

**Figure 6. Summary of other oocyte parameters from experiments like those in Fig. 2: ±hAQP5, Δcytosolic hCA II, all in the absence of extracellular bCA (i.e. –bCA)**
A, summary of initial $pH_i$ values. We computed individual initial $pH_i$, $\Delta pH_i$ (see panel B) and $\beta_I$ (see panel C) values as outlined in 'Methods' > 'Electrophysiological measurements' > 'Calculation of intrinsic intracellular buffering power'. B, summary of $\Delta pH_i$ elicited by addition of 1.5% $CO_2$/10 mM $HCO_3^-$. C, summary of intrinsic buffering power ($\beta_I$). D, summary of $P_f$. We computed individual initial $pH_i$, $\Delta pH_i$ (see panel B) and $\beta_I$ (see panel C) values as outlined in 'Methods' > 'Electrophysiological measurements' > 'Measurement of $P_f$'. The 'Δ' in the title and elsewhere in the paper implies that we choose among several levels of injected hCA II. Data are presented as mean ± SD. '−' and lighter-coloured bars indicate that we injected oocytes injected with $H_2O$; '+' and darker-coloured bars, with cRNA encoding hAQP5 on Day 1. '0', '1', '10' and '100' indicate the amount of hCA II (ng) injected into oocytes on Day 4. Grey/black and light-/dark-blue, red and green pairs of bars indicate eight groups of oocytes. These colours correspond to the colours of the rectangles that surround panel labels (e.g. '$H_2O$ + Tris') in Fig. 2. Statistical star/bar conventions. See legend of Fig. 3 for a description of statistical significance. See Statistics Table 6 for *P*-values. The complete statistical analyses are presented in Table A6*A-D*.

production. (3) With expression of hAQP5 (black bars) – combining an increased $P_{M,CO_2}$ with the capacity of bCA to replenish/consume cell-surface $CO_2$ – cytosolic CA activity becomes rate-limiting. Thus during $CO_2$ addition $CO_2$ rapidly builds up near the inner surface of the membrane, limiting the transmembrane gradient and choking $CO_2$ influx – especially as assessed by a $pH_S$ electrode that is 'cis' to the bCA. The same is true in reverse during $CO_2/HCO_3^-$ removal. (4) The evidence that hAQP5 does indeed increase $P_{M,CO_2}$ will come below in our analysis of Fig. 8C.

**$dpH_i/dt$.** The green records in Fig. 7A and Fig. 7B show the $pH_i$ data collected simultaneously with the red $pH_S$ transients (presented above) in these same panels. Note that $(dpH_i/dt)_{Max}$ values in Fig. 7A (+bCA) are substantially greater – both with $CO_2/HCO_3^-$ addition and removal – than their counterparts in Fig. 2A (–bCA). Musa-Aziz et al. (2014b) had previously made a similar observation with extracellular

hCA II (compare their figures 13e and 15e). Note that in comparing $pH_i$ changes in Fig. 7A *vs.* Fig. 2A the $pH_i$ electrode is 'trans' to the bCA that we add to the bECF. During $CO_2/HCO_3^-$ addition bCA accelerates extracellular $CO_2$ formation and thus maintains a relatively high $[CO_2]$ near the extracellular surface of the membrane. Similarly during $CO_2/HCO_3^-$ removal bCA accelerates $CO_2$ consumption and thus maintains a relatively low $[CO_2]$ at the extracellular membrane surface. The greater transmembrane $CO_2$ gradients lead to greater $CO_2$ fluxes and thus faster 'trans-side' $pH_i$ changes. Thus comparing Fig. 7A *vs.* Fig. 2A we can conclude that bCA increases transmembrane $CO_2$ fluxes.

If we now compare the $pH_i$ records in Fig. 7A and Fig. 7B, we see that hAQP5 expression produces a substantial acceleration in the rate of $pH_i$ decrease during $CO_2/HCO_3^-$ addition, consistent with an hAQP5-dependent increase in $CO_2$ influx. During $CO_2/HCO_3^-$ removal the presence of hAQP5 does not substantially speed the $pH_i$ increase.

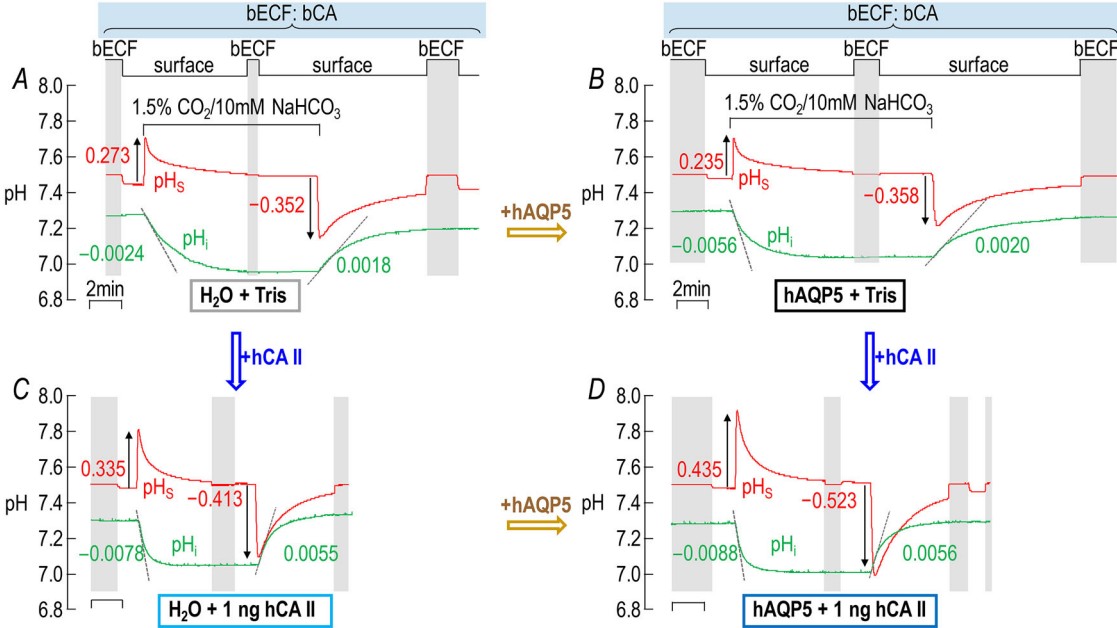

**Figure 7. Representative $pH_S$ and $pH_i$ recordings: ±hAQP5, ±cytosolic hCA II, all in the presence of extracellular bCA (i.e. +bCA)**
A, 'H₂O + Tris'. We injected this oocyte with H₂O (as a control for cRNA encoding hAQP5) on Day 1 and with 'Tris' (as a control for bCA II) on Day 4. B, 'hAQP5 + Tris'. We injected this oocyte on Day 1 with cRNA encoding hAQP5 and then on Day 4 with 'Tris'. C, H₂O + 1 ng hCA II'. Similar to panel A, except that on Day 4, we injected the oocyte with hCA II enzyme dissolved in 'Tris'. D, 'hAQP5 + 1 ng hCA II'. Similar to panel B, except that on Day 4, we injected the oocyte with hCA II enzyme dissolved in 'Tris'. The numbers in red are ΔpH_S and those in green are $(dpH_i/dt)_{Max}$ for the oocytes presented here. The stair-step line at the top of A and B indicates the position of the $pH_S$ electrode. The vertical shaded bars indicate times during which the $pH_S$ electrode is in the bulk extracellular fluid (bECF) for recalibration; at other times the $pH_S$ electrode dimples the cell surface for actual $pH_S$ measurements. The colours of the rectangles that surround the panel labels (e.g. 'H₂O + Tris') correspond to the colours of the bars in Fig. 8 and later figures. ECF, bCA indicates that we obtained all measurements while exposing oocytes to solutions containing 0.1 mg/ml of bCA.

*Summary.* The grey and black bars in Fig. 8*C* summarize the downward $(dpH_i/dt)_{Max}$ for $CO_2/HCO_3^-$ addition, $\pm$hAQP5, all in the presence of extracellular bCA but absence of hCA II. Fig. 8*D* summarizes the corresponding data for $CO_2/HCO_3^-$ removal.

*Conclusions.* Considering oocytes examined in the absence of hCA II but presence of bCA (Fig. 8*C,D*): during $CO_2/HCO_3^-$ addition expression of hAQP5 (black *vs.* grey bars) significantly increases the magnitude of $(dpH_i/dt)_{Max}$. During $CO_2/HCO_3^-$ removal expression of hAQP5 causes $(dpH_i/dt)_{Max}$ to trend faster, although the difference is not statistically significant.

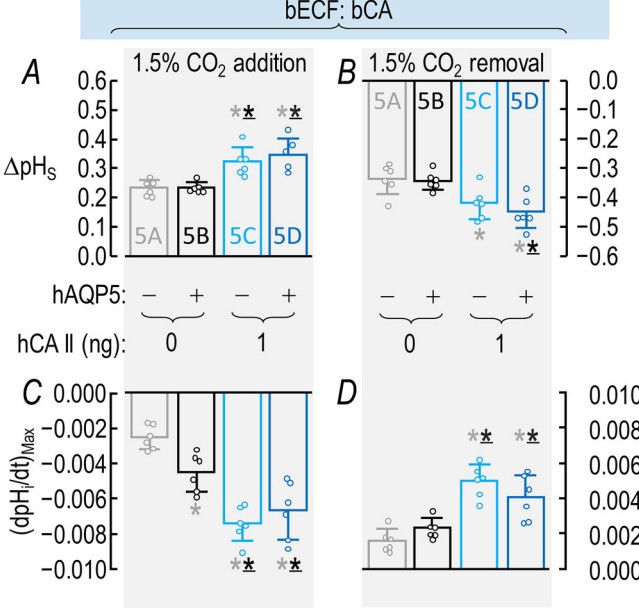

**Figure 8. Summary of $\Delta pH_S$ and $(dpH_i/dt)_{Max}$ data from experiments like those in Fig. 7: $\pm$hAQP5, $\pm$ cytosolic hCA II, all in the presence of extracellular bCA (i.e. +bCA)**
This figure is analogous to Fig. 3, which summarized $\Delta pH_S$ and $(dpH_i/dt)_{Max}$ data obtained from oocytes in the absence of bCA. A, summary of $\Delta pH_S$ upon addition of 1.5% $CO_2/10$ mM $HCO_3^-$. We computed individual $\Delta pH_S$ values as outlined in 'Methods' > 'Electrophysiological measurements' > 'Calculation of $\Delta pH_S$'. B, summary of $\Delta pH_S$ upon removal of 1.5% $CO_2/10$ mM $HCO_3^-$. C, summary of $(dpH_i/dt)_{Max}$ upon addition of 1.5% $CO_2/10$ mM $HCO_3^-$. We computed individual $\Delta pH_S$ values as outlined in 'Methods' > 'Electrophysiological measurements' > 'Calculation of 'maximal' $dpH_S/dt$'. D, summary of $(dpH_i/dt)_{Max}$ upon removal of 1.5% $CO_2/10$ mM $HCO_3^-$. Data are presented as mean $\pm$ SD. '$-$' and lighter-coloured bars indicate oocytes injected with $H_2O$; '+' and darker-coloured bars indicate oocytes injected with cRNA encoding hAQP5 on Day 1. '0' and '1' indicate the amount of hCA II (ng) injected into oocytes on Day 4. Grey/black and light-/dark-blue pairs of bars indicate four groups of oocytes. These colours correspond to the colours of the rectangles that surround panel labels (e.g. '$H_2O$ + Tris') in Fig. 7. Statistical star/bar conventions. See legend of Fig. 3 for a description of statistical significance. See Statistics Table 8 for *P*-values. The complete statistical analyses are presented in Table A8*A-D*.

*Interpretation.* $(dpH_i/dt)_{Max}$ (grey vs. black bars) in Fig. 8*C,D*; $\pm$hAQP5, $-$hCA II, +bCA. See our $\Delta pH_S$ 'Interpretation' immediately above.[13] (1) bCA promotes large transmembrane $CO_2$ gradients. (2) In a 'trans-side' effect bCA greatly increases the magnitudes of $(dpH_i/dt)_{Max}$, as we can see by comparing grey and black bars in Fig. 8*C,D* (+bCA) *vs.* Fig. 3*C,D* ($-$bCA) – consequences of the gradient effect in point #1. (3) Here in Fig. 8*C* (+bCA) hAQP5 increases the magnitude of $(dpH_i/dt)_{Max}$ during $CO_2/HCO_3^-$ addition, demonstrating that the hAQP5 increases $P_{M,CO_2}$. In Fig. 3*C* ($-$bCA) the effect of expressing hAQP5 was nil because the limited availability of extracellular $CO_2$ choked influx.

## $\pm$hAQP5, +hCA II, +bCA: effects on $\Delta pH_S$ and $(dpH_i/dt)_{Max}$

To continue our exploration of the functional interaction of hAQP5 with CAs in the diffusion of $CO_2$ in this next series of experiments we not only augment the extracellular solutions with 0.1 mg/ml bCA but also inject hCA II. On Day 1 we injected all oocytes with either '$H_2O$' or '$H_2O$ + cRNA' encoding 'hAQP5' and then on Day 4 we injected all oocytes with 1 ng hCA II.

**$pH_S$.** The red records in Fig. 7*C* ($-$hAQP5, +hCA II, +bCA) and Fig. 7*D* (+hAQP5, +hCA II, +bCA) are analogous to those in panels A and B, except for the injection of hCA II. We now make three sets of comparisons:

(a) Comparing the red $pH_S$ records between Fig. 7*C* (+hCA II) and Fig. 7*A* ($-$hCA II) – having in common $-$hAQP5, +bCA – we see that the addition of 1 ng hCA II increases the magnitudes of the $pH_S$ transients – both with $CO_2/HCO_3^-$ addition and removal. Recall that Musa-Aziz et al. (2014a) made similar $\Delta pH_S$ observations $\pm$cytosolic hCA II, though in the absence of added $CA_o$. Because the $pH_S$ electrode is 'trans' to the added hCA II (in comparing Fig. 7*C vs.* Fig. 7*A*), we can conclude that the hCA II has increased the transmembrane $CO_2$ flux in the continued presence of extracellular bCA.

(b) Comparing the $pH_S$ records in Fig. 7*D* (+hCA II) *vs.* Fig. 7*B* ($-$hCA II) – having in common +hAQP5, +bCA – we see that the addition of 1 ng hCA II again increases the magnitudes of the $pH_S$ transients.

(c) Comparing Fig. 7*D* (+hAQP5) *vs.* Fig. 7*C* ($-$hAQP5) we see that on a background of injected hCA II

---

[13]See Results > '$\pm$hAQP5, $-$hCA II, +bCA: effects on $\Delta pH_S$ and $(dpH_i/dt)_{Max}$' > '$pH_S$' > 'Interpretation: $\Delta pH_S$ (grey vs. black bars) in Fig. 8*A,B*: $\pm$hAQP5, $-$hCA II, +bCA'.

and extracellular bCA hAQP5 expression has limited effect.

(d) Note that the $pH_S$ transients here in Fig. 7*C,D* (±hAQP5, +hCA II, +bCA) are much larger than their counterparts in Fig. 2*C,D* (±hAQP5, +hCA II, –bCA).

*Summary.* (a) The grey *vs.* light-blue bars in Fig. 8*A* show the mean $\Delta pH_S$ data for $CO_2/HCO_3^-$ addition, ± hCA II, all in the absence of hAQP5 but presence of bCA. Fig. 8*B* shows the corresponding data for $CO_2/HCO_3^-$ removal. (b) The black *vs.* dark-blue bars compare ±hCA II but in the presence of both hAQP5 and bCA. (c) The light-blue *vs.* dark-blue bars compare ±hAQP5, all in the presence of both hCA II and bCA.

*Conclusions.* (a) The difference in mean values represented by the grey and light-blue bars (±hCA II in the absence of hAQP5) is statistically significant, as is (b) the difference between the black and dark-blue bars (±hCA II in the presence of hAQP5). Note that the effects in 'a' and 'b' correspond to the previously described synergistic effects of $CA_i$ and $CA_o$ (Musa-Aziz et al., 2014a, 2014b; Occhipinti et al., 2014). (c) The difference between the light- *vs.* dark-blue bars (±hAQP5 in the presence of hCA II and bCA) is not significant. (d) The statistical analysis summarized in Statistics Table 8E,F compares $\Delta pH_S$ amplitudes summarized in Fig. 8*A,B* (+bCA) *vs.* the grey/black/light-blue/dark-blue bars in Fig. 3*A,B* (–bCA). The result is that the effect of bCA is significant.

*Interpretation.* $\Delta pH_S$ (colourful bars) in Fig. 8*A,B*; ±hAQP5, +hCA II, +bCA. See 'Interpretation' for $\Delta pH_S$ in the previous section[14]: (1) the bCA enhances transmembrane $CO_2$. (2) The $\Delta pH_S$ data summarized by the light/dark-blue bars in Fig. 8*A,B* (+bCA) are in stark contrast to the analogous data in Fig. 3*A,B* (–bCA) – note that this is a 'cis' comparison (i.e. ±bCA→$pH_S$) – where the $\Delta pH_S$ values are only about $\frac{1}{8}$ to $\frac{1}{2}$ as large during $CO_2/HCO_3^-$ addition. (3) Expression of hAQP5 in the presence of 1 ng of injected hCA II (light- *vs.* dark-blue bars), even with the increase in $P_{M,CO_2}$, does not increase $\Delta pH_S$ magnitudes because $CO_2$ fluxes are choked by insufficient cytosolic CA activity (i.e. an insufficient transmembrane $CO_2$ gradient). And (4) recall that hAQP5 does indeed increase $P_{M,CO_2}$ (see Fig. 8*C*). In addition to points #1 – #4, which are analogous to the corresponding points made in the previous section, (5) in a 'trans-side' effect injection of 1 ng of hCA II – either in the absence of hAQP5 (grey *vs.* light-blue bars) or in the presence of hAQP5 (black *vs.* dark-blue bars) – increases the

$\Delta pH_S$ magnitudes because the hCA II is able to increase transmembrane $CO_2$ gradients sufficiently under these conditions.

**$dpH_i/dt$.** The green records in Fig. 7*C* and Fig. 7*D* show the $pH_i$ data that correspond to the $pH_S$ data presented above and lead to three comparisons:

(a) Comparing the $pH_i$ records between Fig. 7*C* (+hCA II) and Fig. 7*A* (–hCA II) – both in the absence of hAQP5 – we see that the addition of 1 ng hCA II substantially increases the rates of $pH_i$ changes, both with $CO_2/HCO_3^-$ addition and removal. Musa-Aziz et al. (2014a) made similar $(dpH_i/dt)_{Max}$ observations ±hCA II, though in the absence of added $CA_o$.

(b) Comparing the $pH_i$ records in Fig. 7*D* (+hCA II) *vs.* Fig. 7*B* (–hCA II) – now both in the presence of hAQP5 – we again see that the addition of 1 ng hCA II increases the magnitudes of $(dpH_i/dt)_{Max}$.

(c) Comparing the $pH_i$ records in Fig. 7*D* (+hAQP5) *vs.* Fig. 7*C* (–hAQP5) we see that – on a background of injected hCA II and extracellular bCA – hAQP5 expression has little effect.

(d) Finally the magnitudes of $(dpH_i/dt)_{Max}$ in Fig. 7*C,D* are modestly larger than those in Fig. 2*C,D*.

*Summary.* The layout and meaning of the bars in Fig. 8*C,D* are the same as in Fig. 8*A,B*.

*Conclusions.* As we observed for the $\Delta pH_S$ data the differences between (a) the two lighter-coloured bars (– hAQP5, ±hCA II) and (b) the two darker-coloured bars (+hAQP5, ±hCA II) are statistically significant. In contrast, (c) the difference between the light- and dark-blue bars (±hAQP5, + hCA II, +bCA) is not significant. (d) The statistical analysis summarized in Statistics Table 8G,H compares $(dpH_i/dt)_{Max}$ magnitudes from Fig. 8*C,D* (+bCA) *vs.* the grey/black/light-blue/dark-blue bars in Fig. 3*C,D* (–bCA). The result is that the effect of bCA is significant.

Interpretation: $(dpH_i/dt)_{Max}$ (colourful bars) in Fig. 8*C,D*; ±hAQP5, +hCA II, +bCA. See 'Interpretation' for $\Delta pH_S$ immediately above[15]: (1) bCA magnifies transmembrane $CO_2$ gradients. (2) In a 'trans' effect the magnitudes of the light- and dark-blue bars in Fig. 8*C,D* (+hCA II, +bCA) are modestly larger than their counterparts in Fig. 3*C,D* (+hCA II, –bCA) – a specific example of enhanced transmembrane $CO_2$ gradients noted in point #1. (3) The light- and dark-blue bars (i.e. ±hAQP5, +hCA II, +bCA) are not significantly different, presumably because 1 ng of injected hCA II does not raise

---

[14]See Results > '±hAQP5, –hCA II, +bCA: effects on $\Delta pH_S$ & $(dpH_i/dt)_{Max}$' > '$pH_S$' > 'Interpretation: $\Delta pH_S$ (grey vs. black bars) in Fig. 8*A,B*: ±hAQP5, –hCA II, +bCA'.

[15]See Results > '±hAQP5, +hCA II, +bCA: effects on $\Delta pH_S$ and $(dpH_i/dt)_{Max}$' > '$pH_S$' > 'Interpretation: $\Delta pH_S$ (colourful bars) in Fig. 8*A,B*: ±hAQP5, +hCA II, +bCA'.

cytosolic CA activity sufficiently to prevent choking the dominant effects of hAQP5 ($\uparrow P_{M,CO_2}$) and bCA ($\uparrow CO_2$ gradient). We predict that greater $CA_i$ activities (e.g. 100 ng) would have alleviated the choke and revealed a much taller +hAQP5 bar. And (4) hAQP5 does indeed increase $P_{M,CO_2}$ as evidenced by the comparison of grey *vs.* black bars in Fig. 8*C*. In addition to points #1–#4 (analogous to those made in previous section) (5) we note that – because of a 'cis' effect – we cannot intuitively interpret the effects of injecting hCA II on $(dpH_i/dt)_{Max}$, either in the absence of hAQP5 (grey *vs.* light-blue bars) or in the presence of hAQP5 (black *vs.* dark-blue bars). Even though we cannot intuitively assess ±hCA II effects from $(dpH_i/dt)_{Max}$ we know from the corresponding $\Delta pH_S$ data in Fig. 8*A,B* that the injection of hCA II – by enhancing transmembrane $CO_2$ gradients – must have increased the $CO_2$ fluxes.

### ±hAQP5, ±hCA II, +bCA: effects on $pH_S$ relaxation

**Summary.** The layout of Fig. 9*A,B* is similar to that of Fig. 8*A,B* (i.e. $\Delta pH_S$), except that here in Fig. 9*A,B* we examine $(dpH_S/dt)_{Max}$ as we did in Fig. 5*A,B*.

**Conclusions.** For oocytes examined in the presence of bCA:

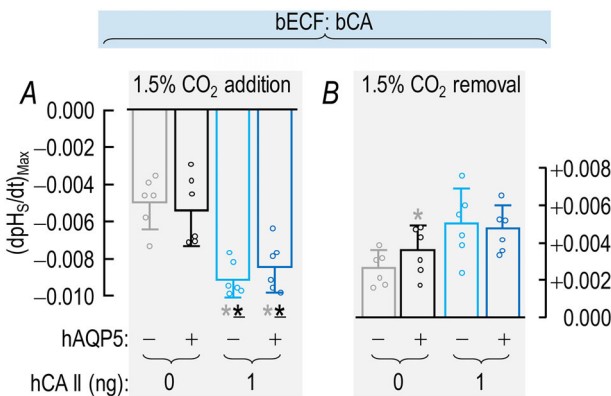

**Figure 9. Summary of $(dpH_S/dt)_{Max}$ data from experiments like those in Fig. 8: ±hAQP5, ±cytosolic hCA II, all in the presence of extracellular bCA (i.e. +bCA)**
This figure is analogous to Fig. 5, which summarized $(dpH_S/dt)_{Max}$ data obtained from oocytes in the absence of bCA. A, summary of $(dpH_S/dt)_{Max}$ upon addition of 1.5% $CO_2$/10 mM $HCO_3^-$. We computed individual $(dpH_S/dt)_{Max}$ values as described in Fig. 4. B, summary of $(dpH_S/dt)_{Max}$ upon removal of 1.5% $CO_2$/10 mM $HCO_3^-$. Data are presented as mean ± SD. '−' and lighter-coloured bars indicate oocytes injected with $H_2O$; '+' and darker-coloured bars indicate oocytes injected with cRNA encoding hAQP5 on Day 1. '0' and '1' indicate the amount of hCA II (ng) injected into oocytes on Day 4. Grey/black and light-/dark-blue pairs of bars indicate four groups of oocytes. These colours correspond to the colours of the rectangles that surround panel labels (e.g. '$H_2O$ + Tris') in Fig. 7. Statistical star/bar conventions. See legend of Fig. 3 for a description of statistical significance. See Statistics Table 9 for *P*-values. The complete statistical analyses are presented in Table A9*A & B*.

**Statistics Table 9.** For clarity within the figure panel we present tables of *P*-values for one-way ANOVA with Tukey's *post hoc* means comparison for the data presented in Fig. 9. For all tables $\alpha$ is 0.05, and significant *P*-values are highlighted in bold. A, *P*-values for means comparisons of $(dpH_S/dt)_{Max}$ on $CO_2$ addition. B, *P*-values for means comparisons of $(dpH_S/dt)_{Max}$ on $CO_2$ removal. C, *P*-values for means comparisons of $\beta_i$. D, *P*-values for means comparisons of $P_f$ data

*9A ($(dpH_S/dt)_{Max}$ on $CO_2$ addition)*

| cRNA | hCA II (ng) | $H_2O$ 'Tris' | hAQP5 'Tris' | $H_2O$ 1 |
|---|---|---|---|---|
| hAQP5 'Tris' | | 0.960 | | |
| $H_2O$ | 1 | **$3.38 \times 10^{-4}$** | **0.00103** | |
| hAQP5 | 1 | **0.00221** | **0.00674** | 0.842 |

*9B ($(dpH_S/dt)_{Max}$ on $CO_2$ removal)*

| cRNA | hCA II (ng) | $H_2O$ 'Tris' | hAQP5 'Tris' | $H_2O$ 1 |
|---|---|---|---|---|
| hAQP5 'Tris' | | **0.0288** | | |
| $H_2O$ | 1 | 0.625 | 0.281 | |
| hAQP5 | 1 | 0.0604 | 0.984 | 0.461 |

(a) In the absence of hAQP5 injecting hCA II (grey *vs.* light-blue bars in Fig. 9*A*) produces an increase in $(dpH_S/dt)_{Max}$ during $CO_2$/$HCO_3^-$ addition; the difference is statistically significant. During $CO_2$/$HCO_3^-$ removal the results trend towards a greater magnitude but do not reach statistical significance (Statistics Table 9*B*). These results in Fig. 9*A* (–hAQP5, ±hCA II, +bCA) are in contrast to those in Fig. 5*A* (–hAQP5, ±hCA II, –bCA), where injected hCA II is without statistically significant effect, and even the trends were mild.

And (b) in oocytes expressing hAQP5 injecting hCA II (black *vs.* dark blue bars) increases the magnitude of $(dpH_S/dt)_{Max}$ during $CO_2$/$HCO_3^-$ addition with a statistically significant difference. This treatment produces a slight upward trend during $CO_2$/$HCO_3^-$ removal. Note that these increments (due to injecting hCA II) are no greater than for oocytes not expressing hAQP5. Viewed differently expressing hAQP5 significantly increases the magnitude of $(dpH_S/dt)_{Max}$ in only 1 of 4 cases ($CO_2$/$HCO_3^-$ removal in the absence of hAQP5, grey *vs.* black bars in Fig. 9*B*). These results contrast to those in Fig. 5 (– bCA) where – in the presence of hAQP5 – increasingly large hCA II injections tended to produce graded increases $(dpH_S/dt)_{Max}$ and for both $CO_2$/$HCO_3^-$ addition and removal. We note, however, that the hCA II injections in Fig. 9 were either 0 or 1 ng, whereas in the Fig. 5 (–bCA) study those were as high as 100 ng.

(c) Expression of hAQP5 in the absence of hCA II (grey *vs.* black bars in Fig. 9*A,B*) or in the presence

of hCA II (light- *vs.* dark-blue bars) has no effect on $(dpH_S/dt)_{Max}$ except during $CO_2/HCO_3^-$ removal in the absence of hCA II (grey *vs.* black bars, Fig. 9*B*). These results contrast with those of Fig. 5 (– bCA), where hAQP5 expression increases $(dpH_S/dt)_{Max}$ magnitudes robustly and consistently.

(d) The $(dpH_S/dt)_{Max}$ magnitudes in Fig. 9 are ∼2- to ∼3-fold greater than the analogous ones in Fig. 5. The differences are greatest in the absence of hCA II, where the lighter-coloured bars in Fig. 9 are ∼10-fold greater.

Interpretation: $(dpH_S/dt)_{Max}$ in Fig. 9*A,B*; ±hAQP5, ±hCA II, +bCA. Our analysis of these $(dpH_S/dt)_{Max}$ is similar to that for the $\Delta pH_S$ data in the previous section.[16]

**±hAQP5, ±hCA II, +bCA: effects on other oocyte parameters.** The four bars in each of the panels of Fig. 10 are analogous to the four leftmost bars of the four panels in Fig. 6; the difference is that in Fig. 10 we summarize experiments in which we exposed oocytes to 0.1 mg/ml extracellular bCA.

*Initial pHi.* Statistical analyses, as in the case of Fig. 6*A*, reveal no significant differences among the four mean initial $pH_i$ values in Fig. 10*A*. Moreover Statistics Table 10E, which summarizes an analysis of values in Fig. 10*A* (+bCA) *vs.* the grey/black/light-blue/dark-blue bars in Fig. 6*A* (–bCA), reveals no significant effect of adding bCA.

**ΔpHi.** We observe no significant differences among mean $\Delta pH_i$ values in Fig. 10*B*, as observed among groups in Fig. 6*B*. However Statistics Table 10F, which summarizes an analysis of values in Fig. 10*B* (+bCA) *vs.* the grey/black/light-blue/dark-blue bars in Fig. 6*B* (–bCA), reveals a small but significant difference due to the addition of bCA. The magnitudes of $\Delta pH_i$ values in Fig. 6*B* are somewhat smaller, presumably reflecting the slower transmembrane equilibration of $CO_2$ due to the absence of bCA, particularly at lower levels of injected hCA II.

**βI.** Continuing the trend from the previous two panels we observe no significant differences among mean $\beta_I$ in Fig. 10*C*, as described among analogous conditions in Fig. 6*C*. However Statistics Table 10G, which summarizes an analysis of values in Fig. 10*C* (+bCA) *vs.* the grey/black/light-blue/dark-blue bars in Fig. 6*C* (–bCA), reveals a small but significant difference – similar to the analysis of the $\Delta pH_i$ data above. Because their $\Delta pH_i$ magnitudes tend to be smaller, the $\beta_I$ values in Fig. 6*C*

(– bCA) are somewhat larger than those in Fig. 10*C* (+bCA).

***P*f.** As for the previous three panels the mean values summarized in Fig. 10*D* (+bCA) – now for $P_f$ – are indistinguishable from the comparable values in Fig. 6*D* (–bCA). Our statistical analyses in Statistics Table 10H reveal no significant effect of bCA on $P_f$. Thus we can conclude that extracellular bCA – a mixture of CA I and CA II – does not interfere with the monomeric pores of hAQP5. Moreover as we did in our analysis of Fig. 6*D* we can conclude, from a comparison of the black and dark-blue bars, that the hCA II does not functionally interfere with the monomeric pores.

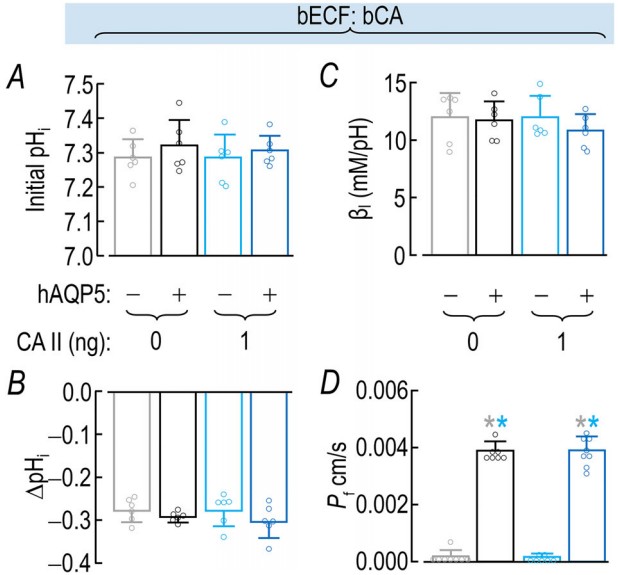

**Figure 10. Summary of other oocyte parameters from experiments like those in Fig. 7: ±hAQP5, ±cytosolic hCA II, all in the presence of extracellular bCA (i.e. +bCA)**

This figure is analogous to Fig. 6, which summarized comparable data obtained from oocytes in the absence of bCA. A, summary of initial $pH_i$ values. We computed individual initial $pH_i$, $\Delta pH_i$ (see panel B) and $\beta_I$ (see panel C) values as outlined in 'Methods' > 'Electrophysiological measurements' > 'Calculation of intrinsic intracellular buffering power'. B, summary of $\Delta pH_i$ elicited by addition of 1.5% $CO_2$/10 mM $HCO_3^-$. C, summary of intrinsic buffering power ($\beta_I$). D, summary of $P_f$. We computed individual initial $pH_i$, $\Delta pH_i$ (see panel B) and $\beta_I$ (see panel C) values as outlined in 'Methods' > 'Electrophysiological measurements' > 'Measurement of $P_f$'. Data are presented as mean ± SD. '–' and lighter-coloured bars indicate oocytes injected with $H_2O$; '+' and darker-coloured bars indicate oocytes injected with cRNA encoding hAQP5 on Day 1. '0' and '1' indicate the amount of hCA II (ng) injected into oocytes on Day 4. Grey/black and light-/dark-blue pairs of bars indicate eight groups of oocytes. These colours correspond to the colours of the rectangles that surround panel labels (e.g. '$H_2O$ + Tris') in Fig. 7. Statistical star/bar conventions. See legend of Fig. 3 for a description of statistical significance. See Statistics Table 10 for *P*-values. The complete statistical analyses are presented in Table A10*A-D*.

---

[16] See Results > '±hAQP5, +hCA II, +bCA: effects on $\Delta pH_S$ and $(dpH_i/dt)_{Max}$' > 'pH$_S$' > 'Interpretation: $\Delta pH_S$ (colourful bars) in Fig. 8*A,B*: ±hAQP5, +hCA II, +bCA'.

**Statistics Table 10.** **For clarity within the figure panel we present tables of *P*-values for one-way ANOVA with Tukey's *post hoc* means comparison for the data presented in Fig. 10. For all tables** $\alpha$ **is 0.05, and significant *P*-values are highlighted in bold. A, *P*-values for means comparisons of initial pH$_i$ data. B, *P*-values for means comparisons of $\triangle$pH$_i$ data. C, *P*-values for means comparisons of** $\beta_I$**. D, *P*-values for means comparisons of** $P_f$ **data. Panels E–H provide descriptive statistics and means comparisons (see Methods, Statistical analysis) for comparisons of data represented by the grey, black, light-blue and dark-blue bars in Fig. 6 (–bCA) *vs.* Fig. 10 (+bCA). E, Initial pH$_i$. F, $\triangle$pH$_i$ amplitude on addition of** $CO_2/HCO_3^-$**. G,** $\beta_I$**. H,** $P_f$**.**

*10A initial pH$_i$*

| cRNA | hCA II (ng) | $H_2O$ 'Tris' | hAQP5 'Tris' | $H_2O$ 1 |
|---|---|---|---|---|
| hAQP5 | 'Tris' | 0.752 | | |
| $H_2O$ | 1 | 1.00 | 0.727 | |
| hAQP5 | 1 | 0.939 | 0.974 | 0.925 |

*10B $\triangle$pH$_i$ data*

| cRNA | hCA II (ng) | $H_2O$ 'Tris' | hAQP5 'Tris' | $H_2O$ 1 |
|---|---|---|---|---|
| hAQP5 | 'Tris' | 0.798 | | |
| $H_2O$ | 1 | 1.00 | 0.779 | |
| hAQP5 | 1 | 0.486 | 0.949 | 0.465 |

*10C Buffering power ($\beta_I$)*

| cRNA | hCA II (ng) | $H_2O$ 'Tris' | hAQP5 'Tris' | $H_2O$ 1 |
|---|---|---|---|---|
| hAQP5 | 'Tris' | 0.994 | | |
| $H_2O$ | 1 | 1.00 | 0.994 | |
| hAQP5 | 1 | 0.652 | 0.799 | 0.659 |

*10D $P_f$ data*

| cRNA | hCA II (ng) | $H_2O$ 'Tris' | hAQP5 'Tris' | $H_2O$ 1 |
|---|---|---|---|---|
| hAQP5 | 'Tris' | **$2.10 \times 10^{-12}$** | | |
| $H_2O$ | 1 | 0.999 | **$2.10 \times 10^{-12}$** | |
| hAQP5 | 1 | **$2.10 \times 10^{-12}$** | 0.999 | **$2.10 \times 10^{-12}$** |

*10E, initial pH$_i$ values ±bCA in bECF (Fig. 6A vs. Fig. 10A)*

| Descriptive stats. | $n$ | Mean | SD | SEM | | | |
|---|---|---|---|---|---|---|---|
| bECF | 32 | 7.30 | 0.0551 | 0.00975 | | | |
| bECF + bCA | 24 | 7.31 | 0.0603 | 0.0123 | | | |
| **Means comparison** | **Mean diff.** | **SEM** | **Q-value** | **P-value** | $\alpha$ | **LCL** | **UCL** |
| bECF *vs.* bECF + bCA | 0.00669 | 0.0155 | 0.610 | 0.668 | 0.0500 | −0.0244 | 0.0378 |

*10F, $\triangle$pH$_i$ upon $CO_2$ addition ±bCA in bECF (Fig. 6B vs. Fig. 10B)*

| Descriptive stats. | $n$ | Mean | SD | SEM | | | |
|---|---|---|---|---|---|---|---|
| bECF | 32 | −0.258 | 0.0258 | 0.00456 | | | |
| bECF + bCA | 24 | −0.285 | 0.0308 | 0.00628 | | | |
| **Means comparison** | **Mean diff.** | **SEM** | **Q-value** | **P-value** | $\alpha$ | **LCL** | **UCL** |
| bECF *vs.* bECF + bCA | −0.0270 | 0.00756 | 5.04 | **$7.74 \times 10^{-4}$** | 0.0500 | 0.0118 | 0.0421 |

(*Continued*)

**Statistics Table 10. (Continued)**

*10G, $\beta_i$ values for oocytes $\pm$bCA in bECF (Fig. 6C vs. Fig. 10C)*

| Descriptive stats. | *n* | Mean | SD | SEM | | | |
|---|---|---|---|---|---|---|---|
| bECF | 32 | 13.6 | 1.93 | 0.341 | | | |
| bECF + bCA | 24 | 11.6 | 1.73 | 0.353 | | | |
| **Means comparison** | **Mean diff.** | **SEM** | **Q-value** | **P-value** | **$\alpha$** | **LCL** | **UCL** |
| bECF *vs.* bECF + bCA | −2.03 | 0.498 | 5.75 | **$1.55 \times 10^{-4}$** | 0.0500 | −3.02 | −1.03 |

*10H, $P_f$ $\pm$bCA in bECF (Fig. 6D vs. Fig. 10D)*

| Descriptive stats. | *n* | Mean | SD | SEM | | | |
|---|---|---|---|---|---|---|---|
| bECF | 32 | 0.00196 | 0.00184 | $3.25 \times 10^{-4}$ | | | |
| bECF + bCA | 30 | 0.00204 | 0.00192 | $3.50 \times 10^{-4}$ | | | |
| **Means comparison** | **Mean diff.** | **SEM** | **Q-value** | **P-value** | **$\alpha$** | **LCL** | **UCL** |
| bECF *vs.* bECF + bCA | $8.03 \times 10^{-5}$ | $4.77 \times 10^{-4}$ | 0.238 | 0.867 | 0.0500 | $-8.74 \times 10^{-4}$ | 0.00103 |

## Discussion

### Historical context

**Permeability *vs*. gradient.** Although previous papers on $CO_2$ diffusion across membranes have addressed the role of channels or the role of CAs, the present paper is the first to undertake a systematic examination of both channels and CAs, as well as their synergistic interaction to enhance $CO_2$ fluxes. The underlying principle is Fick's law of diffusion, which we reproduce from eqn (2):

$$J_{CO_2} = \underset{\pm hAQP5}{P_{M^*,CO_2}} \cdot \left( \underset{\pm bCA}{[CO_2]_{os}} - \underset{\pm hCA\,II}{[CO_2]_{is}} \right) \qquad (3)$$

Transmembrane $CO_2$ gradient

and embellish to emphasize that the key parameters in the present study increase $P_{M^*,CO_2}$ (i.e. hAQP5) and the transmembrane $CO_2$ gradient. During $CO_2$ influx bCA (when present) in the extracellular fluid maintains a relatively high $[CO_2]_{os}$, whereas hCA II (when present) maintains a relatively low $[CO_2]_{is}$, the result being an enhanced inwardly directed $CO_2$ gradient. Conversely during $CO_2$ efflux bCA in the extracellular fluid maintains a relatively low $[CO_2]_{os}$, whereas hCA II maintains a relatively high $[CO_2]_{is}$, the result being an enhanced outwardly directed $CO_2$ gradient.

It is perhaps worth noting that similar principles are at work during $O_2$ fluxes across the erythrocyte membrane. In that case, according to papers not yet peer reviewed (Moss et al., 2025; Occhipinti et al., 2025; Zhao et al., 2025), AQP1, the Rh complex and as-yet-unidentified channel(s) make the dominant contribution to $P_{M,CO_2}$. Moreover it is haemoglobin that maximizes transmembrane $O_2$ gradients (ignoring haemoglobin diffusion within the cytoplasm) by serving as a sink or source of $O_2$ near the plasma membrane. In the laboratory experiments on $O_2$ efflux from RBCs one can use an extracellular $O_2$ scavenger like sodium dithionite to consume $O_2$ near the extracellular face of the membrane in an action that is analogous to that of bCA in the present experiments on $CO_2$ efflux.

Working on artificial lipid bilayers Gutknecht et al. (1977) showed that mobile (more or less freely diffusible) CA enhances $CO_2$ fluxes (measured using $^{14}C$-labeled $CO_2$) but only in the presence of sufficient non-$CO_2$/$HCO_3^-$ buffers (phosphate, HEPES, 'Tris'). Their interpretation is that the CA increases the transmembrane $CO_2$ gradient by replenishing $CO_2$ on one side of the membrane and consuming it on the other. This replenishment (or consumption) requires that $HCO_3^-$ diffuse through the unconvected layers on opposite sides of the membrane to serve as either the source or product of the CA reactions. Moreover the buffers act as either a source of or sink for $H^+$ in these reactions.

The trio of papers by Musa-Aziz et al. (2014a, 2014b) and Occhipinti et al. (2014) extended this line of reasoning to *Xenopus* oocytes, which they injected with recombinant hCA II (to an apparent concentration ~50% higher than in RBCs) and/or cRNA encoding hCA IV (which contributes to both extracellular-surface and cytosolic CA activity). In addition they systematically varied $[CO_2]_o$ and $[HEPES]_o$ and performed three-dimensional reaction-diffusion mathematical modelling to enable a quantitative interpretation of the data. As anticipated from the earlier artificial lipid-bilayer work they found that each CA alone augmented $CO_2$ diffusion by increasing transmembrane $CO_2$ gradients. However the combination of the two is not simply additive but highly synergistic. The reason is that, with a CA on just one side of the membrane, net $CO_2$ fluxes are limited by the build-up or depletion of $CO_2$ on the other – the 'choking' (or throttling) effect to which we allude in the Results section

of the present paper. They also demonstrated an additional synergism involving an extracellular non-$HCO_3^-$ buffer (i.e. HEPES), assessed by 'trans-side' measurements of $pH_i$, and examined the impact of increasing $[CO_2]_o$ during $CO_2/HCO_3^-$ application. In their papers they did not address the issue of $CO_2$ permeability.

The first studies to address the $P_{M,CO_2}$ term in eqn (3) were those by Nakhoul et al. (1998) and Cooper and Boron (1998), who showed that AQP1 – in addition to being a $H_2O$ channel – is an effective $CO_2$ channel. Musa-Aziz et al. (2007) and Geyer et al. (2013) later examined a range of mammalian AQPs and showed that AQP5 has the highest $CO_2/H_2O$ permeability ratio. The only study to broach channel-CA interactions was that by Nakhoul et al. (1998), who studied $CO_2$-induced $pH_i$ changes in oocytes with the vitelline membrane intact. They found that the stimulatory effect of AQP1 on $CO_2$ influx into oocytes occurs only with bCA injected into the cytoplasm. In the present paper the analogous experiment is summarized by the light- *vs.* dark-blue bars in Fig. 3*C*, where hAQP5 expression causes $(dpH_i/dt)_{Max}$ to trend faster. Perhaps a difference, as suggested by others (Vilas et al., 2015), is that CA II binds to AQP1 but not AQP5.

The present paper builds on the work of Musa-Aziz and Occhipinti (Musa-Aziz et al., 2014a, 2014b; Occhipinti et al., 2014) and generally uses identical approaches, with three important differences:

First rather than recombinant hCA II in the present study we use hCA II purified commercially from RBCs.

Second rather than heterologously expressing hCA IV we add bCA to the extracellular fluid. The reasons for this switch are fourfold: (1) hCA IV expression in oocytes increases not only cell-surface CA activity but also cytosolic CA activity. The oocyte membrane confines bCA to the outside of the cell. (2) The co-expression of hCA IV and hAQP5 adds an extra burden of heterologous expression and also introduces potential competition between the injected cRNAs encoding the two proteins ('ribosome steal'). (3) We also chose to add purified CA protein to the bECF because in principle we know the precise increase in $CA_o$ activity, which is difficult to know in the case of protein expression. And (4) we chose bCA because it is a less-expensive combination of bCA I (less active) and bCA II (more active) *vs.* hCA II; this is an important practical consideration during continuous-flow experiments, which consume considerable volumes of experimental solutions.

Third rather than working only with oocytes having a background $P_{M,CO_2}$ we alternated between injections of $H_2O$ and cRNA encoding hAQP5.

**Buffering.** One might ask whether the introduction of a CA increases buffering power ($\beta$). The short answer is 'no'. We divided $\beta$ for $CO_2/HCO_3^-$ into two components: (1)

$CO_2/HCO_3^-$ buffering power and (2) non-$CO_2/HCO_3^-$ buffering power (see Thornell et al., 2025). In the present study we see that increasing $[CO_2]_o$ causes $pH_i$ to fall – an example of intracellular 'respiratory acidosis'. The $CO_2/HCO_3^-$ buffer system does contribute to the $CO_2$-induced acid load, and thus CAs have no impact on $\beta$, even though they may markedly increase the rate at which $pH_i$ reaches a new steady state (see Figs. 6*C* and 10*C*). It is the non-$CO_2/HCO_3^-$ buffers that limit the extent of a $CO_2$-induced pH change. If the intracellular acid load had been 'metabolic' in nature (e.g. the iontophoretic injection of $KHCO_3$ or HCl as pioneered by the late Roger Thomas in 1976), then both $CO_2/HCO_3^-$ and non-$CO_2/HCO_3^-$ buffers would have contributed to $\beta_{Total}$. However even here CAs would only have speeded the attainment of $CO_2/HCO_3^-$ equilibration and not increased the magnitude of buffering.

## pH measurements made 'cis' *vs.* 'inter' *vs.* 'trans' to the altered parameter

The papers of Musa-Aziz et al. (2014a, 2014b) and Occhipinti et al. (2014) introduced the concept of pH measurements, used as an indirect measure of something else (e.g. $CO_2$ flux), being made 'cis' or 'trans' to a compartment with an alteration that directly affects pH (e.g. CA activity, non-$HCO_3^-$ buffering power). Thus if one wishes to assess the effects on $CO_2$ flux of injecting (or not injecting) hCA II into the cytosol, the measurement of $pH_i$ – 'cis' to the added hCA II – does not provide intuitive insight into the effects of hCA II' on transmembrane $CO_2$ fluxes. Of course at the same time as making the $pH_i$ measurement one may also monitor $pH_S$ – 'trans' to the added hCA II in the same cell. Interpreting the data from such a 'trans' perspective does provide intuitive insight into $CO_2$ fluxes. The same principles apply in the opposite sense if one wishes to assess the effects of adding (or not adding) bCA to the bECF. The measurement of $pH_S$ – 'cis' to the added bCA – does not provide intuitive insight into the effects of bCA on $CO_2$ fluxes, whereas the measurement of $pH_i$ – 'trans' – does.

The reason for the 'cis-side' prohibition is that the enzymatic activity of the CA produces or consumes $H^+$ and thus can produce large pH changes that are independent of – and could conflated with the interpretation of – changes in transmembrane $CO_2$ fluxes. Thus intuitive interpretations can be extremely difficult, even though mathematical modelling can unravel alterations in 'cis-side' pH from alterations in the flux of $CO_2$ or other buffer components. Such unraveling was a major component of the studies of Musa-Aziz et al. (2014a, 2014b) and Occhipinti et al. (2014), which demonstrated that both cytosolic and extracellular CA increase transmembrane $CO_2$ gradients and thus increase

$CO_2$ fluxes. Of course avoiding a prohibited 'cis-side' interpretation requires proper experimental design. However the 'cis-side' prohibition is more an issue of data interpretation: when altering any parameter that affects acid–base reaction rates in Fig. 1*A*, an intuitive interpretation requires that one examine pH 'trans' to the alteration.

Stated simply 'trans-side' pH measurements are valuable for intuitive interpretations because trans-side pH change can occur only as the result of altered $CO_2$ fluxes across the membrane.

The experiments summarized by Fig. 8 are a useful case study. bCA is present continuously in the bECF. Thus when we inject hCA II, the $pH_S$ measurements – 'trans' to the hCA II – do provide intuitive insight into the effect of hCA II on transmembrane $CO_2$ fluxes because in this particular comparison the bCA status is unchanging. Basing one's intuition on $pH_i$ measurements – 'cis' to the added hCA II – would be unwise. However if we compare Fig. 8 (+bCA) with Fig. 3 (–bCA), an intuitive assessment of bCA effects must come from $pH_i$ measurements – 'trans' to the ±bCA condition. Thus 'cis' *vs.* 'trans' is not a matter of data *per se* but a matter of perspective during data analysis.

Finally in Figs. 3 and 8 we also assess the effects of expressing the integral membrane protein hAQP5. Both $pH_S$ and $pH_i$ measurements are neither 'cis' nor 'trans' to the altered expression of hAQP5 and thus the altered $P_{M,CO_2}$. One might use the term 'inter' (Latin, 'within', 'inside' or 'between') to indicate the position of the protein relative to the membrane. In such an 'inter' situation one could use both $pH_S$ and $pH_i$ measurements to assess the data intuitively.

## Molecular mechanism of $CO_2$ permeability

Given the suggestion that AQPs may conduct dissolved gases via the hydrophobic central pore of the tetramer (Boron, 2010; Wang et al., 2007) it is not surprising that molecular dynamics simulations are consistent with the hypothesis that $O_2$ (Zhang & Chen, 2013) and $CO_2$ (Alishahi & Kamali, 2019) can diffuse through the central pore of hAQP5. Preliminary data on hAQP5 suggest that (1) mutating amino acid residues near the outer mouth of the central pore to residues with bulky side chains (e.g. T41F) or (2) creating a divalent-cation binding site (T41H) and then adding $Ni^{2+}$ or $Zn^{2+}$ greatly reduces $\Delta pH_S$, as supported by crystal structures and molecular dynamics (Shinn et al., 2024). In addition a preliminary report suggests that with hAQP1 the mercurial pCMBS can block one component of $P_{M,CO_2}$, the stilbene derivative DIDS can block an equally large component, and together the two can eliminate the $CO_2$ permeability of hAQP1 (Musa-Aziz et al., 2025). Thus the emerging picture is that some $CO_2$ can permeate the four hydrophilic monomeric

pores (at least of hAQP1), whereas another component – presumably the major component of $CO_2$ – moves through the central pore of hAQP5.

## Diagnostic power of $\Delta\Delta\Delta pH_S \pm hCA\ II$ in identifying enhanced $CO_2$ permeability

In Fig. 11 we rearrange $CO_2$-influx bars from Fig. 3*A* so that we can easily compare the effects – on $\Delta pH_S$ – of adding a CA, namely hCA II, on the side of the membrane 'trans' to the $pH_S$ measurement. The pair of bars on the left of Fig. 11*A* shows that in the absence of hAQP5 injecting 1 ng of hCA II into oocytes has only a minor effect on $\Delta pH_S$. That is the baseline $\Delta\Delta pH_S$ – or $\Delta\Delta pH_{S,Base}$ – is small. However in the presence of a $CO_2$ channel hAQP5 $\Delta\Delta pH_S$ – or $\Delta\Delta pH_{S,hAQP5}$ – is substantially greater. The difference between the two $\Delta\Delta pH_S$ values – the $\Delta\Delta\Delta pH_S$ due to the presence of hAQP5 or $\Delta\Delta\Delta pH_{S,hAQP5}$ – is statistically significant (Table A11*A*).

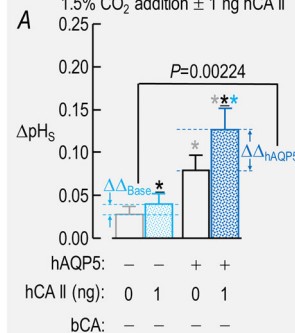
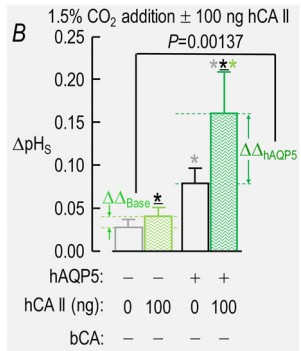

**Figure 11. Effect of expressing hAQP5 on $\Delta\Delta pH_S$: comparison of selected $\Delta pH_S$ bars extracted from Fig. 3*A* (–bCA)**
Here we rearrange bars from Fig. 3A to juxtapose two bars representing the same hAQP5 status ('–', absent; '+', heterologously expressed): the left bar in a pair representing the absence of hCA II and the right bar representing the presence of injected hCA II. A, $\Delta\Delta pH_S$ determined with ±1 ng of injected hCA II during addition of $CO_2/HCO_3^-$. B, $\Delta\Delta pH_S$ determined with ±100 ng of injected hCA II during addition of $CO_2/HCO_3^-$. For a pair of bars the difference between $\Delta pH_S$ values is $\Delta\Delta pH_S$. Thus $\Delta\Delta_{Base}$ is difference between $\Delta pH_S$ values under baseline conditions (i.e. no hAQP5) induced by the addition of the trans-side CA (i.e. CA II); $\Delta\Delta_{hAQP5}$ is the corresponding difference between $\Delta pH_S$ values in AQP5-expressing oocytes. Comparing two pairs of bars the difference between $\Delta\Delta_{hAQP5}$ and $\Delta\Delta_{Base}$ values is $\Delta\Delta\Delta pH_{S,hAQP5}$. Although no bCA is present in any of these experiments, we include bCA status ('–', absent from bECF) to facilitate comparisons with Fig. 12. Statistical star/bar conventions. See legend of Fig. 3 for a description of statistical significance. The stars, which indicate statistical significance among the individual bars, have the same meanings as in Fig. 3A. The *P*-values indicate statistical significance between the respective $\Delta\Delta_{Base}$ and $\Delta\Delta_{hAQP5}$ bar pairs and reflect the effect of adding 1 ng hCA II (panel *A*) or 100 ng hCA II (Panel *B*) to the 'trans' side of the membrane during the $pH_S$ measurement. See Table A11*A* and Table A11*B* for the statistics summary (including all *P*-values reported by the stars).

Figure 11*B* is a rearrangement of two pairs of $CO_2$-influx bars, but with 100 ng of injected hCA II. The $\Delta\Delta pH_{S,Base}$ is similar to the value in Fig. 11*A*. However here the effect of hAQP5 expression, namely $\Delta\Delta pH_{S,hAQP5}$, is even greater. Thus it appears that the greater the level of hCA II the greater is the effect of hAQP5 expression on $\Delta\Delta pH_S$. Although this set of comparisons is of $CO_2$ influx, we could reach a similar set of conclusions for $CO_2$ efflux (i.e. by rearranging bars in Fig. 3*B*).

If we did not know the identity of the membrane protein expressed in these experiments, we could deduce that the protein must have a significant $CO_2$ conductance, assuming that the protein itself lacks significant CA activity. We can imagine that the cytosolic CA, by serving as a sink for $CO_2$ during influx, sucks $CO_2$ into the cell. If $P_{M,CO_2}$ is rate-limiting, the addition of a $CO_2$ channel will lead to increased $CO_2$ influx, a greater decrease in $[CO_2]$ at the extracellular cell surface and thus a greater $\Delta pH_S$.

Note that in a cell in which it is impractical to inject or express a cytosolic CA one might use a drug like acetazolamide to block endogenous CA II and thereby perform an analogous $\Delta\Delta\Delta pH_S$ assay.

### Diagnostic power of $\Delta\Delta(dpH_i/dt)_{Max}$ ±bCA in identifying enhanced $CO_2$ permeability

In Fig. 12 we perform the inverse analysis of Fig. 11: we rearrange $CO_2$-influx bars from Fig. 3*C* and Fig. 8*C* so that we can easily compare the effects – now on $\Delta(dPH_i/dt)_{Max}$ – of adding a CA – now bCA – on the side of the membrane 'trans' to the $pH_i$ measurement. The pair of bars on the left of Fig. 12*A* shows that, in the absence of hAQP5, adding bCA to the bECF produces a modest increase in the magnitude of $(dpH_i/dt)_{Max}$ – the $\Delta(dpH_i/dt)_{Max,Base}$. On the contrary in the presence of the $CO_2$ channel hAQP5 the magnitude of $(dpH_i/dt)_{Max}$ – $\Delta(dpH_i/dt)_{Max,hAQP5}$ – is substantially greater. The difference between the two $\Delta(dpH_i/dt)_{Max}$ values – the $\Delta\Delta(dpH_i/dt)_{Max}$ due to the presence of hAQP5 or $\Delta\Delta(dpH_i/dt)_{Max,hAQP5}$ – is statistically significant (Table A12*A*).

Figure 12*B* shows a similar analysis for $CO_2/HCO_3^-$ removal. Because the $(dpH_i/dt)_{Max}$ values are smaller, the $\Delta(dpH_i/dt)_{Max}$ values also are of smaller magnitude than in Fig. 12*B*. In this case the difference between the two $\Delta(dpH_i/dt)_{Max}$ values is not statistically significant. Thus this assay may not be practical as shown. A future approach could be to raise $[CO_2]_o$, raise $[HEPES]_o$, inject 100 ng of hCA II and raise the bCA level (albeit at the increased cost of the enzyme in a continuously flowing solution). Some combination of these changes would increase the rates of $pH_i$ change, and thereby make it easier to detect $(dpH_i/dt)_{Max}$ differences in both the influx and efflux assays.

## Advances and Limitations

**Synergisms.** In the present study we confirm the strong synergy between extracellular and cytosolic CAs (see Figs. 3 and 8). We also identify a strong synergy between a $CO_2$ channel (i.e. hAQP5) and a cytosolic CA (i.e. hCA II; see $pH_S$ data in Fig. 3). Finally we identify a strong synergy between a $CO_2$ channel and an extracellular CA (i.e. bCA; see $dpH_i/dt$ data in Fig. 8).

**'Cis-' *vs*. 'trans-side' effects.** Our group has previously pointed out the challenges in pH measurements made 'cis' to a CA manipulation (Lu et al., 2006). For example 'cis-side' $pH_i$ measurements led to the erroneous conclusion that cytosolic CA II binds to and thereby stimulates the Cl-$HCO_3$ exchanger AE1, while, in fact, the added CA II was catalysing the reaction $HCO_3^- + H^+ \rightarrow$

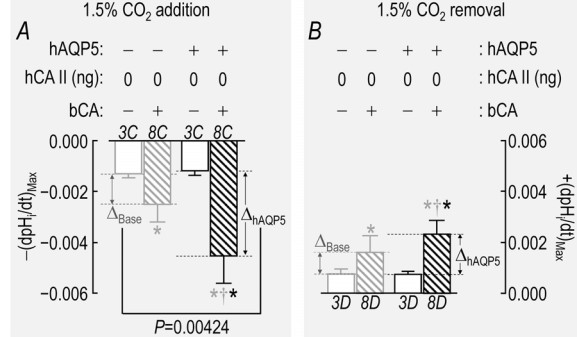

**Figure 12. Effect of expressing hAQP5 on $\Delta(dpH_i/dt)_{Max}$: comparison of selected $(dpH_i/dt)_{Max}$ bars extracted from Fig. 3*C vs*. Fig. 8*C* and from Fig. 3*D vs*. Fig. 8*D*.**
Here we rearrange the grey and black bars from Fig. 3*C* (–bCA) and the grey and black bars from Fig. 8*C* (+bCA) to juxtapose two bars representing the same hAQP5 status ('–', absent; '+', heterologously expressed): the left bar in a pair representing the absence of bCA and the right bar in a pair representing the presence of bCA in the bECF. A, summary of $\Delta(dpH_i/dt)_{Max}$ ±bCA upon addition of $CO_2/HCO_3^-$. B, summary of $\Delta(dpH_i/dt)_{Max}$ ±bCA upon removal of $CO_2/HCO_3^-$. For a pair of bars the difference between $(dpH_i/dt)_{Max}$ values is $\Delta(dpH_i/dt)_{Max}$. Thus $\Delta_{Base}$ is the difference between $(dpH_i/dt)_{Max}$ values under baseline conditions (i.e. no hAQP5) induced by the addition of the trans-side CA (i.e. bCA); $\Delta_{hAQP5}$ is the corresponding difference between $(dpH_i/dt)_{Max}$ values in hAQP5-expressing oocytes. Comparing two pairs of bars the difference between $\Delta_{hAQP5}$ and $\Delta_{Base}$ values is $\Delta\Delta(dpH_i/dt)_{Max,hAQP5}$. Although no hCA II is present in any of these experiments, we include hCA II status ('0 ng', not injected) to facilitate comparisons with Fig. 11. Statistical star/bar conventions. See legend of Fig. 3 for a description of statistical significance. The stars indicate statistical significance among individual bars; the grey star associated with the grey- and black-hatched bars indicates a significant difference compared to the open grey bar (–hAQP5, –bCA); the black star associated with the black-hatched bars (+AQP5, +bCA) indicates a significant difference compared to the open black bar (+hAQP5, –bCA); the grey dagger symbol associated with the black-hatched bars (+AQP5, +bCA) indicates a significant difference compared to the open grey-hatched bar (–hAQP5, +bCA). See Table A12*A* and Table A12*B* for the statistics summary (including all the *P*-values reported by the stars).

$CO_2 + H_2O$ and producing a rapid $pH_i$ increase because of catalysis, not transport.

The aforementioned trio of papers (Musa-Aziz et al., 2014a, 2014b; Occhipinti et al., 2014) systematically deals with the issue of pH measurements made 'cis' to manipulations that impact acid–base chemistry (e.g. modulation of CA activity, alterations in non-$HCO_3^-$ buffering power) and the necessity to focus on 'trans' effects. The present paper builds upon and expands these concepts. Note that our modulation of $P_{M,CO_2}$ – an 'inter' situation – is neither 'cis' nor 'trans' to our electrodes and is thus immune from these considerations.

**'Inter' effect.** We introduce this term to describe a manoeuvre that impacts the membrane separating two solutions (e.g. intra- *vs.* extracellular). For example the act of introducing, deleting, mutating or otherwise changing the activity of an integral membrane protein is 'in between' 'cis' and 'trans'. Likewise altering lipid composition or otherwise altering the chemistry of membrane lipids or their interaction with proteins or other substances would be an 'inter' effect.

**$pH_S$ relaxation assessed as $(dpH_S/dt)_{Max}$.** Although we have previously assessed $pH_S$ relaxation as the time constant of a single-exponential decay, we now introduce a more general tool for describing the initial rate of $pH_S$ decay, especially one that deviates substantially from an SExp time course. Although we implement this tool using a double-exponential curve fit, in principle we could use any function that near $t_{Local} = 0$ fits $pH_S$ *vs.* time with a monotonic decay.

**Novel diagnostic paradigms.** Our approaches for assessing $\Delta\Delta\Delta pH_S$ and $\Delta\Delta(dpH_i/dt)_{Max}$ are potentially valuable for evaluating the $CO_2$ permeability of candidate membrane proteins.

**Limitations.** A potential drawback of using bCA rather than hCA IV is that, with bCA added to the bECF, chemical or physical factors near the membrane may limit its access to the nanodomain at the membrane surface. On the contrary the use of bCA avoids the issue of 'ribosome steal' noted above. Moreover bCA ideally provides CA activity throughout the extracellular unconvected layer. Indeed Musa-Aziz et al. (2014b) and Occhipinti et al. (2014) showed that bCA II together with hCA IV (the extracellular presence of which is presumably confined to the nanodomain at the membrane surface) is more effective than hCA IV alone.

In the present study, recognizing an already large matrix of experimental conditions, we did not explore bCA levels higher than 0.1 mg/ml, levels that may have generated greater $(dpH_i/dt)_{Max}$ signals. In our bCA study we did not explore hCA II levels greater than 1 ng, levels that presumably would have generated larger $\Delta pH_S$ signals.

Unlike the trio of papers noted above the present study does not include assessments of altered extracellular concentrations of (1) $CO_2$ (1.5%, 5%, 10% *vs.* 1.5% in present study) or (2) non-$HCO_3^-$ buffers (1, 5, 25 mM HEPES *vs.* 5 mM here). Such investigations are important because – especially together with the accompanying mathematical model – they provide valuable insights into the impact and interactions of various acid–base buffer members and CAs on transmembrane $CO_2$ fluxes. Including such analyses in the present study together with ±hAQP5 would have expanded our experimental matrix to unrealistic levels.

Musa-Aziz et al. (2014b) found that (1) increasing $[CO_2]_o$ markedly increased both $\Delta pH_S$ and $(dpH_i/dt)_{Max}$, especially in the presence of hCA IV (their fig. 10). The effects of increasing $[HEPES]_o$ were more complex: (2) with 1.5% $CO_2$ raising $[HEPES]_o$ predictably decreased $\Delta pH_S$ (a 'cis-side' effect) but did not significantly affect $(dpH_i/dt)_{Max}$ (their fig. 13). (3) When they supplemented the hCA IV with extracellular bCA II, still with 1.5% $CO_2$, the effect of raising $[HEPES]_o$ on $(dpH_i/dt)_{Max}$ became stronger but still not significant (their fig. 15). However (4) when they worked with 10% $CO_2$, the effects of raising $[HEPES]_o$ on $(dpH_i/dt)_{Max}$ were far stronger (their fig. 17).

Based on the above observations we suggest the following for future $(dpH_i/dt)_{Max}$ experiments on oocytes intended to examine the synergy among hAQP5, $CA_i$ and the following: in $\Delta\Delta(dpH_i/dt)_{Max}$ protocols it would be helpful to (a) raise $[CO_2]_o$ (e.g. to 10%), (b) explore increasing bCA beyond 0.1 mg/ml and (c) raise $[HEPES]_o$ (e.g. to 25 mM). We also suggest that in future $\Delta\Delta\Delta pH_S$ protocols it would be helpful to (a) raise the $[CO_2]_o$ and (b) increase $CA_i$ activity (e.g. employing recombinant hCA II, injecting 100 ng or more of purified hCA II).

## Appendix

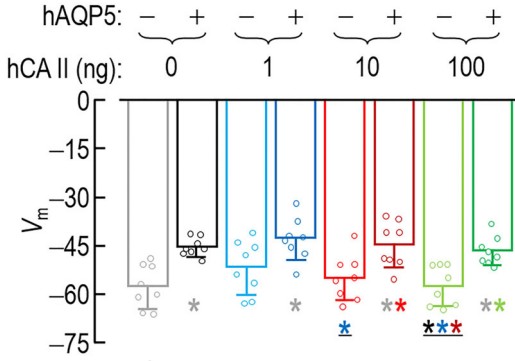

**Figure A1. Summary of membrane-potential ($V_m$) values for oocytes used in Fig. 3**

The bar graph summarizes $V_m$ values – obtained near the end of experiments (e.g. the rightmost grey band in Fig. 1*C*) – for the oocytes used in Fig. 3. Statistical star/bar conventions: see legend of Fig. 3 for a description of statistical significance. See Table A13 for the statistics summary (including all *P*-values reported by the stars).

**Table A3A. Statistics summary for Fig. 3*A*: (i) descriptive statistics for data presented in the figure panel, $\triangle pH_S$ upon $CO_2$ addition; (ii) overall one-way ANOVA results; (iii) Tukey's means comparison analysis**

### i. Descriptive statistics

| | n | Mean | SD | SEM |
|---|---|---|---|---|
| $H_2O$ + Tris | 8 | 0.0285 | 0.00895 | 0.00316 |
| hAQP5 + Tris | 8 | 0.0787 | 0.0181 | 0.00641 |
| $H_2O$ + 1 ng hCA II | 8 | 0.0411 | 0.0115 | 0.00406 |
| hAQP5 + 1 ng hCA II | 8 | 0.127 | 0.0246 | 0.00869 |
| $H_2O$ + 10 ng hCA II | 8 | 0.0418 | 0.00938 | 0.00332 |
| hAQP5 + 10 ng hCA II | 8 | 0.131 | 0.0262 | 0.00925 |
| $H_2O$ + 100 ng hCA II | 8 | 0.0416 | 0.00948 | 0.00335 |
| hAQP5 + 100 ng hCA II | 8 | 0.160 | 0.0478 | 0.0169 |

### ii. Overall ANOVA

| Variation | DF | Sum of squares | Mean square | *F*-value | *P*-value |
|---|---|---|---|---|---|
| Between groups | 7 | 0.147 | 0.0210 | 39.2 | $2.44 \times 10^{-19}$ |
| Within groups | 56 | 0.0301 | $5.37 \times 10^{-4}$ | | |
| Total | 63 | 0.177 | | | |

### iii. Tukey's test

| Condition A | Condition B | Mean diff. | *Q*-value | *P*-value | Sig. | LCL | UCL |
|---|---|---|---|---|---|---|---|
| hAQP5 + Tris | $H_2O$ + Tris | 0.0502 | 6.13 | 0.00150 | ✓ | 0.0137 | 0.0867 |
| $H_2O$ + 1 ng hCA II | $H_2O$ + Tris | 0.0126 | 1.54 | 0.956 | – | −0.0238 | 0.0491 |
| $H_2O$ + 1 ng hCA II | hAQP5 + Tris | −0.0376 | 4.59 | 0.0391 | ✓ | −0.0741 | −0.00110 |
| hAQP5 + 1 ng hCA II | $H_2O$ + Tris | 0.0986 | 12.0 | $4.18 \times 10^{-8}$ | ✓ | 0.0622 | 0.135 |
| hAQP5 + 1 ng hCA II | hAQP5 + Tris | 0.0484 | 5.91 | 0.00247 | ✓ | 0.0120 | 0.0849 |
| hAQP5 + 1 ng hCA II | $H_2O$ + 1ng hCA II | 0.0860 | 10.5 | $7.19 \times 10^{-8}$ | ✓ | 0.0495 | 0.122 |
| $H_2O$ + 10 ng hCA II | $H_2O$ + Tris | 0.0133 | 1.63 | 0.943 | – | −0.0232 | 0.0498 |
| $H_2O$ + 10 ng hCA II | hAQP5 + Tris | −0.0369 | 4.50 | 0.0456 | ✓ | −0.0734 | $-4.10 \times 10^{-4}$ |
| $H_2O$ + 10 ng hCA II | $H_2O$ + 1ng hCA II | $6.88 \times 10^{-4}$ | 0.0839 | 1.00 | – | −0.0358 | 0.0372 |
| $H_2O$ + 10 ng hCA II | hAQP5 + 1ng hCA II | −0.0853 | 10.4 | $7.72 \times 10^{-8}$ | ✓ | −0.122 | −0.0488 |
| hAQP5 + 10 ng hCA II | $H_2O$ + Tris | 0.103 | 12.6 | $3.90 \times 10^{-8}$ | ✓ | 0.0664 | 0.139 |
| hAQP5 + 10 ng hCA II | hAQP5 + Tris | 0.0527 | 6.43 | $7.38 \times 10^{-4}$ | ✓ | 0.0162 | 0.0892 |
| hAQP5 + 10 ng hCA II | $H_2O$ + 1 ng hCA II | 0.0902 | 11.0 | $5.43 \times 10^{-8}$ | ✓ | 0.0538 | 0.127 |
| hAQP5 + 10 ng hCA II | hAQP5 + 1 ng hCA II | 0.00424 | 0.518 | 1.00 | – | −0.0322 | 0.0407 |
| hAQP5 + 10 ng hCA II | $H_2O$ + 10 ng hCA II | 0.0896 | 10.9 | $5.60 \times 10^{-8}$ | ✓ | 0.0531 | 0.126 |
| $H_2O$ + 100 ng hCA II | $H_2O$ + Tris | 0.0131 | 1.60 | 0.947 | – | −0.0234 | 0.0496 |
| $H_2O$ + 100 ng hCA II | hAQP5 + Tris | −0.0371 | 4.53 | 0.0437 | ✓ | −0.0736 | $-6.01 \times 10^{-4}$ |
| $H_2O$ + 100 ng hCA II | $H_2O$ + 1 ng hCA II | $4.96 \times 10^{-4}$ | 0.0606 | 1.00 | – | −0.0360 | 0.0370 |
| $H_2O$ + 100 ng hCA II | hAQP5 + 1 ng hCA II | −0.0855 | 10.4 | $7.56 \times 10^{-8}$ | ✓ | −0.122 | −0.0490 |
| $H_2O$ + 100 ng hCA II | $H_2O$ + 10 ng hCA II | $-1.91 \times 10^{-4}$ | 0.0233 | 1.00 | – | −0.0367 | 0.0363 |
| $H_2O$ + 100 ng hCA II | hAQP5 + 10 ng hCA II | −0.0898 | 11.0 | $5.55 \times 10^{-8}$ | ✓ | −0.126 | −0.0533 |
| hAQP5 +100 ng hCA II | $H_2O$ + Tris | 0.132 | 16.1 | $2.21 \times 10^{-8}$ | ✓ | 0.0955 | 0.168 |
| hAQP5 +100 ng hCA II | hAQP5 + Tris | 0.0817 | 9.98 | $1.34 \times 10^{-7}$ | ✓ | 0.0453 | 0.118 |
| hAQP5 +100 ng hCA II | $H_2O$ + 1 ng hCA II | 0.119 | 14.6 | $2.93 \times 10^{-8}$ | ✓ | 0.0828 | 0.156 |
| hAQP5 +100 ng hCA II | hAQP5 + 1 ng hCA II | 0.0333 | 4.06 | 0.0978 | – | −0.00318 | 0.0698 |
| hAQP5 +100 ng hCA II | $H_2O$ + 10 ng hCA II | 0.119 | 14.5 | $2.97 \times 10^{-8}$ | ✓ | 0.0821 | 0.155 |
| hAQP5 +100 ng hCA II | hAQP5 + 10 ng hCA II | 0.0291 | 3.55 | 0.214 | – | −0.00742 | 0.0655 |
| hAQP5 +100 ng hCA II | $H_2O$ + 100 ng hCA II | 0.119 | 14.5 | $2.95 \times 10^{-8}$ | ✓ | 0.0823 | 0.155 |

*Notes*: Threshold for significance a = 0.05. Standard error of the mean = 0.0116.
Abbreviations: DF, degrees of freedom; LCL, lower confidence limit; mean diff., mean difference for each comparison; UCL, upper confidence limit.

**Table A3B. Statistics summary for Fig. 3B: (i) descriptive statistics for data presented in the figure panel, $\triangle pH_S$ upon $CO_2$ removal; (ii) overall one-way ANOVA results; (iii) Tukey's means comparison analysis**

### i. Descriptive statistics

|  | $n$ | Mean | SD | SEM |
|---|---|---|---|---|
| $H_2O$ + Tris | 8 | −0.03093 | 0.00994 | 0.00352 |
| hAQP5 + Tris | 8 | −0.09598 | 0.02688 | 0.0095 |
| $H_2O$ + 1 ng hCA II | 8 | −0.04167 | 0.01131 | 0.004 |
| hAQP5 + 1 ng hCA II | 8 | −0.13982 | 0.02817 | 0.00996 |
| $H_2O$ + 10 ng hCA II | 8 | −0.04078 | 0.01032 | 0.00365 |
| hAQP5 + 10 ng hCA II | 8 | −0.13894 | 0.03249 | 0.01149 |
| $H_2O$ + 100 ng hCA II | 8 | −0.04123 | 0.0107 | 0.00378 |
| hAQP5 + 100 ng hCA II | 8 | −0.16218 | 0.05069 | 0.01792 |

### ii. Overall ANOVA

| Variation | DF | Sum of squares | Mean square | $F$-value | $P$-value |
|---|---|---|---|---|---|
| Between groups | 7 | 0.165 | 0.0236 | 33.8 | $6.93 \times 10^{-18}$ |
| Within groups | 56 | 0.0391 | $6.99 \times 10^{-4}$ | | |
| Total | 63 | 0.204 | | | |

### iii. Tukey's test

| Condition A | Condition B | Mean diff. | $Q$-value | $P$-value | Sig. | LCL | UCL |
|---|---|---|---|---|---|---|---|
| hAQP5 + Tris | $H_2O$ + Tris | −0.0650 | 6.96 | $2.03 \times 10^{-4}$ | ✓ | −0.107 | −0.0234 |
| $H_2O$ + 1 ng hCA II | $H_2O$ + Tris | −0.0107 | 1.15 | 0.992 | – | −0.0524 | 0.0309 |
| $H_2O$ + 1 ng hCA II | hAQP5 + Tris | 0.0543 | 5.81 | 0.00309 | ✓ | 0.0127 | 0.0959 |
| hAQP5 + 1 ng hCA II | $H_2O$ + Tris | −0.109 | 11.7 | $4.44 \times 10^{-8}$ | ✓ | −0.150 | −0.0673 |
| hAQP5 + 1 ng hCA II | hAQP5 + Tris | −0.0438 | 4.69 | 0.0321 | ✓ | −0.0854 | −0.00223 |
| hAQP5 + 1 ng hCA II | $H_2O$ + 1 ng hCA II | −0.0981 | 10.5 | $7.16 \times 10^{-8}$ | ✓ | −0.140 | −0.0565 |
| $H_2O$ + 10 ng hCA II | $H_2O$ + Tris | −0.00985 | 1.05 | 0.995 | – | −0.0515 | 0.0318 |
| $H_2O$ + 10 ng hCA II | hAQP5 + Tris | 0.0552 | 5.91 | 0.00249 | ✓ | 0.0136 | 0.0968 |
| $H_2O$ + 10 ng hCA II | $H_2O$ + 1 ng hCA II | $8.96 \times 10^{-4}$ | 0.0959 | 1.00 | – | −0.0407 | 0.0425 |
| $H_2O$ + 10 ng hCA II | hAQP5 + 1 ng hCA II | 0.0990 | 10.6 | $6.67 \times 10^{-8}$ | ✓ | 0.0574 | 0.141 |
| hAQP5 + 10 ng hCA II | $H_2O$ + Tris | −0.108 | 11.6 | $4.51 \times 10^{-8}$ | ✓ | −0.150 | −0.0664 |
| hAQP5 + 10 ng hCA II | hAQP5 + Tris | −0.0430 | 4.60 | 0.0383 | ✓ | −0.0846 | −0.00136 |
| hAQP5 + 10 ng hCA II | $H_2O$ + 1 ng hCA II | −0.0973 | 10.4 | $7.76 \times 10^{-8}$ | ✓ | −0.139 | −0.0557 |
| hAQP5 + 10 ng hCA II | hAQP5 + 1 ng hCA II | $8.74 \times 10^{-4}$ | 0.0935 | 1.00 | – | −0.0407 | 0.0425 |
| hAQP5 + 10 ng hCA II | $H_2O$ + 10 ng hCA II | −0.0982 | 10.5 | $7.15 \times 10^{-8}$ | ✓ | −0.140 | −0.0566 |
| $H_2O$ + 100 ng hCA II | $H_2O$ + Tris | −0.0103 | 1.10 | 0.994 | – | −0.0519 | 0.0313 |
| $H_2O$ + 100 ng hCA II | hAQP5 + Tris | 0.0547 | 5.86 | 0.00278 | ✓ | 0.0131 | 0.0964 |
| $H_2O$ + 100 ng hCA II | $H_2O$ + 1 ng hCA II | $4.40 \times 10^{-4}$ | 0.0471 | 1.00 | – | −0.0412 | 0.0420 |
| $H_2O$ + 100 ng hCA II | hAQP5 + 1 ng hCA II | 0.0986 | 10.5 | $6.91 \times 10^{-8}$ | ✓ | 0.0570 | 0.140 |
| $H_2O$ + 100 ng hCA II | $H_2O$ + 10 ng hCA II | $-4.56 \times 10^{-4}$ | 0.0488 | 1.00 | – | −0.0421 | 0.0412 |
| $H_2O$ + 100 ng hCA II | hAQP5 + 10ng hCA II | 0.0977 | 10.5 | $7.44 \times 10^{-8}$ | ✓ | 0.0561 | 0.139 |
| hAQP5 +100 ng hCA II | $H_2O$ + Tris | −0.131 | 14.0 | $3.17 \times 10^{-8}$ | ✓ | −0.173 | −0.0896 |
| hAQP5 +100 ng hCA II | hAQP5 + Tris | −0.0662 | 7.08 | $1.50 \times 10^{-4}$ | ✓ | −0.108 | −0.0246 |
| hAQP5 +100 ng hCA II | $H_2O$ + 1 ng hCA II | −0.121 | 12.9 | $3.72 \times 10^{-8}$ | ✓ | −0.162 | −0.0789 |
| hAQP5 +100 ng hCA II | hAQP5 + 1 ng hCA II | −0.0224 | 2.39 | 0.692 | – | −0.0640 | 0.0192 |
| hAQP5 +100 ng hCA II | $H_2O$ + 10 ng hCA II | −0.121 | 13.0 | $3.68 \times 10^{-8}$ | ✓ | −0.163 | −0.0798 |
| hAQP5 +100 ng hCA II | hAQP5 + 10 ng hCA II | −0.0232 | 2.49 | 0.650 | – | −0.0648 | 0.0184 |
| hAQP5 +100 ng hCA II | $H_2O$ + 100 ng hCA II | −0.121 | 12.9 | $3.70 \times 10^{-8}$ | ✓ | −0.163 | −0.0793 |

*Notes*: Threshold for significance a = 0.05. Standard error of the mean = 0.0132.
Abbreviations: DF, degrees of freedom; LCL, lower confidence limit; mean diff., mean difference for each comparison; UCL, upper confidence limit.

**Table A3C. Statistics summary for Fig. 3C: (i) descriptive statistics for data presented in the figure panel, $(dpH_i/dt)_{Max}$ upon $CO_2$ addition; (ii) overall one-way ANOVA results. (iii) Tukey's means comparison analysis**

i. Descriptive statistics

|  | $n$ | Mean | SD | SEM |
|---|---|---|---|---|
| $H_2O$ + Tris | 8 | −0.00126 | $2.08 \times 10^{-4}$ | $7.35 \times 10^{-5}$ |
| hAQP5 + Tris | 8 | −0.00122 | $1.57 \times 10^{-4}$ | $5.55 \times 10^{-5}$ |
| $H_2O$ + 1 ng hCA II | 8 | −0.00458 | 0.00214 | $7.57 \times 10^{-4}$ |
| hAQP5 + 1 ng hCA II | 8 | −0.00553 | 0.00171 | $6.06 \times 10^{-4}$ |
| $H_2O$ + 10 ng hCA II | 8 | −0.00660 | 0.00293 | 0.00104 |
| hAQP5 + 10 ng hCA II | 8 | −0.00732 | 0.00236 | $8.36 \times 10^{-4}$ |
| $H_2O$ + 100 ng hCA II | 8 | −0.00645 | 0.0029 | 0.00102 |
| hAQP5 + 100 ng hCA II | 8 | −0.00601 | 0.00318 | 0.00113 |

ii. Overall ANOVA

| Variation | DF | Sum of squares | Mean square | $F$-value | $P$-value |
|---|---|---|---|---|---|
| Between groups | 7 | $3.17 \times 10^{-4}$ | $4.53 \times 10^{-5}$ | 8.98 | $2.25 \times 10^{-7}$ |
| Within groups | 56 | $2.82 \times 10^{-4}$ | $5.04 \times 10^{-6}$ |  |  |
| Total | 63 | $5.99 \times 10^{-4}$ |  |  |  |

iii. Tukey's test

| Condition A | Condition B | Mean diff. | $Q$-value | $P$-value | Sig. | LCL | UCL |
|---|---|---|---|---|---|---|---|
| hAQP5 + Tris | $H_2O$ + Tris | $3.53 \times 10^{-5}$ | 0.0445 | 1.00 | − | −0.00350 | 0.00357 |
| $H_2O$ + 1 ng hCA II | $H_2O$ + Tris | −0.00332 | 4.18 | 0.0803 | − | −0.00685 | $2.14 \times 10^{-4}$ |
| $H_2O$ + 1 ng hCA II | hAQP5 + Tris | −0.00335 | 4.23 | 0.0744 | − | −0.00689 | $1.79 \times 10^{-4}$ |
| hAQP5 + 1 ng hCA II | $H_2O$ + Tris | −0.00427 | 5.38 | 0.00798 | √ | −0.00780 | $−7.35 \times 10^{-4}$ |
| hAQP5 + 1 ng hCA II | hAQP5 + Tris | −0.00430 | 5.42 | 0.00726 | √ | −0.00784 | $−7.71 \times 10^{-4}$ |
| hAQP5 + 1 ng hCA II | $H_2O$ + 1 ng hCA II | $−9.50 \times 10^{-4}$ | 1.20 | 0.989 | − | −0.00448 | 0.00258 |
| $H_2O$ + 10 ng hCA II | $H_2O$ + Tris | −0.00534 | 6.73 | $3.62 \times 10^{-4}$ | √ | −0.00887 | −0.00180 |
| $H_2O$ + 10 ng hCA II | hAQP5 + Tris | −0.00537 | 6.77 | $3.25 \times 10^{-4}$ | √ | −0.00891 | −0.00184 |
| $H_2O$ + 10 ng hCA II | $H_2O$ + 1 ng hCA II | −0.00202 | 2.54 | 0.624 | − | −0.00555 | 0.00152 |
| $H_2O$ + 10 ng hCA II | hAQP5 + 1 ng hCA II | −0.00107 | 1.35 | 0.979 | − | −0.00460 | 0.00247 |
| hAQP5 + 10 ng hCA II | $H_2O$ + Tris | −0.00606 | 7.63 | $3.76 \times 10^{-5}$ | √ | −0.00959 | −0.00252 |
| hAQP5 + 10 ng hCA II | hAQP5 + Tris | −0.00609 | 7.68 | $3.36 \times 10^{-5}$ | √ | −0.00963 | −0.00256 |
| hAQP5 + 10 ng hCA II | $H_2O$ + 1 ng hCA II | −0.00274 | 3.45 | 0.244 | − | −0.00627 | $7.97 \times 10^{-4}$ |
| hAQP5 + 10 ng hCA II | hAQP5 + 1 ng hCA II | −0.00179 | 2.25 | 0.753 | − | −0.00532 | 0.00175 |
| hAQP5 + 10 ng hCA II | $H_2O$ + 10 ng hCA II | $−7.19 \times 10^{-4}$ | 0.906 | 0.998 | − | −0.00425 | 0.00281 |
| $H_2O$ + 100 ng hCA II | $H_2O$ + Tris | −0.00519 | 6.54 | $5.69 \times 10^{-4}$ | √ | −0.00872 | −0.00166 |
| $H_2O$ + 100 ng hCA II | hAQP5 + Tris | −0.00522 | 6.58 | $5.11 \times 10^{-4}$ | √ | −0.00876 | −0.00169 |
| $H_2O$ + 100 ng hCA II | $H_2O$ + 1ng hCA II | −0.00187 | 2.36 | 0.708 | − | −0.00540 | 0.00166 |
| $H_2O$ + 100 ng hCA II | hAQP5 + 1ng hCA II | $−9.20 \times 10^{-4}$ | 1.16 | 0.991 | − | −0.00445 | 0.00261 |
| $H_2O$ + 100 ng hCA II | $H_2O$ + 10ng hCA II | $1.48 \times 10^{-4}$ | 0.187 | 1.00 | − | −0.00339 | 0.00368 |
| $H_2O$ + 100 ng hCA II | hAQP5 + 10ng hCA II | $8.67 \times 10^{-4}$ | 1.09 | 0.994 | − | −0.00267 | 0.00440 |
| hAQP5 +10 0ng hCA II | $H_2O$ + Tris | −0.00475 | 5.98 | 0.00210 | √ | −0.00828 | −0.00121 |
| hAQP5 +100 ng hCA II | hAQP5 + Tris | −0.00478 | 6.03 | 0.00189 | √ | −0.00832 | −0.00125 |
| hAQP5 +100 ng hCA II | $H_2O$ + 1ng hCA II | −0.00143 | 1.80 | 0.905 | − | −0.00496 | 0.00210 |
| hAQP5 +100 ng hCA II | hAQP5 + 1ng hCA II | $−4.79 \times 10^{-4}$ | 0.604 | 1.00 | − | −0.00401 | 0.00305 |
| hAQP5 +100 ng hCA II | $H_2O$ + 10ng hCA II | $5.89 \times 10^{-4}$ | 0.743 | 0.999 | − | −0.00294 | 0.00412 |
| hAQP5 +100 ng hCA II | hAQP5 + 10ng hCA II | 0.00131 | 1.65 | 0.938 | − | −0.00223 | 0.00484 |
| hAQP5 +100 ng hCA II | $H_2O$ + 100ng hCA II | $4.41 \times 10^{-4}$ | 0.556 | 1.00 | − | −0.00309 | 0.00397 |

*Notes*: Threshold for significance a = 0.05. Standard error of the mean = 0.0112.
Abbreviations: DF, degrees of freedom; LCL, lower confidence limit; mean diff., mean difference for each comparison; UCL, upper confidence limit.

**Table A3D. Statistics summary for Fig. 3*D*: (i) descriptive statistics for data presented in the figure panel, $(dpH_i/dt)_{Max}$ upon $CO_2$ removal; (ii) overall one-way ANOVA results. (iii) Tukey's means comparison analysis**

### i. Descriptive statistics

|  | n | Mean | SD | SEM |
|---|---|---|---|---|
| $H_2O$ + Tris | 8 | $8.35 \times 10^{-4}$ | $1.34 \times 10^{-4}$ | $4.74 \times 10^{-5}$ |
| hAQP5 + Tris | 8 | $8.07 \times 10^{-4}$ | $6.85 \times 10^{-5}$ | $2.42 \times 10^{-5}$ |
| $H_2O$ + 1 ng hCA II | 8 | 0.00263 | 0.00119 | $4.19 \times 10^{-4}$ |
| hAQP5 + 1 ng hCA II | 8 | 0.00306 | $8.73 \times 10^{-4}$ | $3.08 \times 10^{-4}$ |
| $H_2O$ + 10 ng hCA II | 8 | 0.00355 | 0.00121 | $4.29 \times 10^{-4}$ |
| hAQP5 + 10 ng hCA II | 8 | 0.00399 | 0.00139 | $4.92 \times 10^{-4}$ |
| $H_2O$ + 100 ng hCA II | 8 | 0.00373 | 0.00266 | $9.39 \times 10^{-4}$ |
| hAQP5 + 100 ng hCA II | 8 | 0.00297 | 0.00134 | $4.74 \times 10^{-4}$ |

### ii. Overall ANOVA

| Variation | DF | Sum of squares | Mean square | *F*-value | *P*-value |
|---|---|---|---|---|---|
| Between groups | 7 | $8.57 \times 10^{-5}$ | $1.22 \times 10^{-5}$ | 6.78 | $7.89 \times 10^{-6}$ |
| Within groups | 56 | $1.01 \times 10^{-4}$ | $1.81 \times 10^{-6}$ | | |
| Total | 63 | $1.87 \times 10^{-4}$ | | | |

### iii. Tukey's test

| Condition A | Condition B | Mean diff. | *Q*-value | *P*-value | Sig. | LCL | UCL |
|---|---|---|---|---|---|---|---|
| hAQP5 + Tris | $H_2O$ + Tris | $-2.75 \times 10^{-5}$ | 0.0579 | 1.00 | – | −0.00214 | 0.00209 |
| $H_2O$ + 1 ng hCA II | $H_2O$ + Tris | 0.00179 | 3.77 | 0.154 | – | $-3.22 \times 10^{-4}$ | 0.00391 |
| $H_2O$ + 1 ng hCA II | hAQP5 + Tris | 0.00182 | 3.83 | 0.141 | – | $-2.95 \times 10^{-4}$ | 0.00394 |
| hAQP5 + 1 ng hCA II | $H_2O$ + Tris | 0.00222 | 4.68 | 0.0329 | ✓ | $1.07 \times 10^{-4}$ | 0.00434 |
| hAQP5 + 1 ng hCA II | hAQP5 + Tris | 0.00225 | 4.74 | 0.0295 | ✓ | $1.34 \times 10^{-4}$ | 0.00437 |
| hAQP5 + 1 ng hCA II | $H_2O$ + 1 ng hCA II | $4.29 \times 10^{-4}$ | 0.903 | 0.998 | – | −0.00169 | 0.00255 |
| $H_2O$ + 10 ng hCA II | $H_2O$ + Tris | 0.00272 | 5.72 | 0.00381 | ✓ | $6.02 \times 10^{-4}$ | 0.00483 |
| $H_2O$ + 10 ng hCA II | hAQP5 + Tris | 0.00275 | 5.78 | 0.00335 | ✓ | $6.29 \times 10^{-4}$ | 0.00486 |
| $H_2O$ + 10 ng hCA II | $H_2O$ + 1ng hCA II | $9.24 \times 10^{-4}$ | 1.94 | 0.865 | – | −0.00119 | 0.00304 |
| $H_2O$ + 10 ng hCA II | hAQP5 + 1ng hCA II | $4.95 \times 10^{-4}$ | 1.04 | 0.995 | – | −0.00162 | 0.00261 |
| hAQP5 + 10 ng hCA II | $H_2O$ + Tris | 0.00315 | 6.63 | $4.57 \times 10^{-4}$ | ✓ | 0.00103 | 0.00527 |
| hAQP5 + 10 ng hCA II | hAQP5 + Tris | 0.00318 | 6.69 | $3.97 \times 10^{-4}$ | ✓ | 0.00106 | 0.00529 |
| hAQP5 + 10 ng hCA II | $H_2O$ + 1 ng hCA II | 0.00136 | 2.85 | 0.479 | – | $-7.59 \times 10^{-4}$ | 0.00347 |
| hAQP5 + 10 ng hCA II | hAQP5 + 1 ng hCA II | $9.28 \times 10^{-4}$ | 1.95 | 0.862 | – | −0.00119 | 0.00304 |
| hAQP5 + 10 ng hCA II | $H_2O$ + 10 ng hCA II | $4.33 \times 10^{-4}$ | 0.911 | 0.998 | – | −0.00168 | 0.00255 |
| $H_2O$ + 100ng hCA II | $H_2O$ + Tris | 0.00290 | 6.10 | 0.00160 | ✓ | $7.84 \times 10^{-4}$ | 0.00502 |
| $H_2O$ + 100ng hCA II | hAQP5 + Tris | 0.00293 | 6.16 | 0.00140 | ✓ | $8.11 \times 10^{-4}$ | 0.00504 |
| $H_2O$ + 100ng hCA II | $H_2O$ + 1 ng hCA II | 0.00111 | 2.33 | 0.721 | – | −0.00101 | 0.00322 |
| $H_2O$ + 100ng hCA II | hAQP5 + 1 ng hCA II | $6.77 \times 10^{-4}$ | 1.42 | 0.972 | – | −0.00144 | 0.00279 |
| $H_2O$ + 100ng hCA II | $H_2O$ + 10 ng hCA II | $1.82 \times 10^{-4}$ | 0.383 | 1.00 | – | −0.00193 | 0.00230 |
| $H_2O$ + 100ng hCA II | hAQP5 + 10 ng hCA II | $-2.51 \times 10^{-4}$ | 0.528 | 1.00 | – | −0.00237 | 0.00187 |
| hAQP5 +100ng hCA II | $H_2O$ + Tris | 0.00214 | 4.50 | 0.0460 | ✓ | $2.19 \times 10^{-5}$ | 0.00425 |
| hAQP5 +100ng hCA II | hAQP5 + Tris | 0.00217 | 4.56 | 0.0413 | ✓ | $4.94 \times 10^{-5}$ | 0.00428 |
| hAQP5 +100ng hCA II | $H_2O$ + 1 ng hCA II | $3.44 \times 10^{-4}$ | 0.724 | 1.00 | – | −0.00177 | 0.00246 |
| hAQP5 +100ng hCA II | hAQP5 + 1 ng hCA II | $-8.49 \times 10^{-5}$ | 0.179 | 1.00 | – | −0.00220 | 0.00203 |
| hAQP5 +100ng hCA II | $H_2O$ + 10 ng hCA II | $-5.80 \times 10^{-4}$ | 1.22 | 0.988 | – | −0.00270 | 0.00154 |
| hAQP5 +100ng hCA II | hAQP5 + 10 ng hCA II | −0.00101 | 2.13 | 0.801 | – | −0.00313 | 0.00110 |
| hAQP5 +100ng hCA II | $H_2O$ + 100 ng hCA II | $-7.62 \times 10^{-4}$ | 1.60 | 0.947 | – | −0.00288 | 0.00135 |

*Notes*: Threshold for significance a = 0.05. Standard error of the mean = $6.27 \times 10^{-4}$.
Abbreviations: DF, degrees of freedom; LCL, lower confidence limit; mean diff., mean difference for each comparison; UCL, upper confidence limit.

**Table A5A. Statistics summary for Fig. 5A: (i) descriptive statistics for data presented in the figure panel, $(dpH_S/dt)_{Max}$ upon $CO_2$ addition; (ii) overall one-way ANOVA results; (iii) Tukey's means comparison analysis**

i. Descriptive statistics

| | $n$ | Mean | SD | SEM |
|---|---|---|---|---|
| $H_2O$ + Tris | 8 | $-6.14 \times 10^{-4}$ | $1.81 \times 10^{-4}$ | $6.39 \times 10^{-5}$ |
| hAQP5 + Tris | 10 | $-0.00171$ | $2.19 \times 10^{-4}$ | $6.92 \times 10^{-5}$ |
| $H_2O$ + 1 ng h CAII | 8 | $-7.81 \times 10^{-4}$ | $3.15 \times 10^{-4}$ | $1.11 \times 10^{-4}$ |
| hAQP5 + 1 ng h CAII | 15 | $-0.00230$ | $4.30 \times 10^{-4}$ | $1.11 \times 10^{-4}$ |
| $H_2O$ + 10 ng h CAII | 10 | $-8.51 \times 10^{-4}$ | $1.48 \times 10^{-4}$ | $4.66 \times 10^{-5}$ |
| hAQP5 + 10 ng h CAII | 9 | $-0.00275$ | $6.86 \times 10^{-4}$ | $2.29 \times 10^{-4}$ |
| $H_2O$ + 100 ng h CAII | 7 | $-8.21 \times 10^{-4}$ | $2.07 \times 10^{-4}$ | $7.84 \times 10^{-5}$ |
| hAQP5 + 100 ng h CAII | 12 | $-0.00379$ | $9.53 \times 10^{-4}$ | $2.75 \times 10^{-4}$ |

ii. Overall ANOVA

| Variation | DF | Sum of squares | Mean square | $F$-value | $P$-value |
|---|---|---|---|---|---|
| Between groups | 7 | $9.44 \times 10^{-5}$ | $1.35 \times 10^{-5}$ | 52.7 | $1.27 \times 10^{-25}$ |
| Within groups | 71 | $1.82 \times 10^{-5}$ | $2.56 \times 10^{-7}$ | | |
| Total | 78 | $1.13 \times 10^{-4}$ | | | |

iii. Tukey's test

| Condition A | Condition B | Mean diff. | SEM | $Q$-value | $P$-value | Sig. | LCL | UCL |
|---|---|---|---|---|---|---|---|---|
| hAQP5 + Tris | $H_2O$ + Tris | $-0.00110$ | $2.40 \times 10^{-4}$ | 6.49 | $4.79 \times 10^{-4}$ | ✓ | $-0.00214$ | 0.00209 |
| $H_2O$ + 1 ng hCA II | $H_2O$ + Tris | $-1.67 \times 10^{-4}$ | $2.53 \times 10^{-4}$ | 0.933 | 0.998 | – | $-3.22 \times 10^{-4}$ | 0.00391 |
| $H_2O$ + 1 ng hCA II | hAQP5 + Tris | $9.34 \times 10^{-4}$ | $2.40 \times 10^{-4}$ | 5.50 | 0.00518 | ✓ | $-2.95 \times 10^{-4}$ | 0.00394 |
| hAQP5 + 1 ng hCA II | $H_2O$ + Tris | $-0.00168$ | $2.21 \times 10^{-4}$ | 10.8 | $4.73 \times 10^{-8}$ | ✓ | $1.07 \times 10^{-4}$ | 0.00434 |
| hAQP5 + 1 ng hCA II | hAQP5 + Tris | $-5.84 \times 10^{-4}$ | $2.06 \times 10^{-4}$ | 4.00 | 0.104 | – | $1.34 \times 10^{-4}$ | 0.00437 |
| hAQP5 + 1 ng hCA II | $H_2O$ + 1ng hCA II | $-0.00152$ | $2.21 \times 10^{-4}$ | 9.69 | $7.84 \times 10^{-8}$ | ✓ | $-0.00169$ | 0.00255 |
| $H_2O$ + 10 ng hCA II | $H_2O$ + Tris | $-2.37 \times 10^{-4}$ | $2.40 \times 10^{-4}$ | 1.40 | 0.975 | – | $6.02 \times 10^{-4}$ | 0.00483 |
| $H_2O$ + 10 ng hCA II | hAQP5 + Tris | $8.64 \times 10^{-4}$ | $2.26 \times 10^{-4}$ | 5.40 | 0.00655 | ✓ | $6.29 \times 10^{-4}$ | 0.00486 |
| $H_2O$ + 10 ng hCA II | $H_2O$ + 1 ng hCA II | $-7.00 \times 10^{-5}$ | $2.40 \times 10^{-4}$ | 0.413 | 1.000 | – | $-0.00119$ | 0.00304 |
| $H_2O$ + 10 ng hCA II | hAQP5 + 1 ng hCA II | 0.00145 | $2.06 \times 10^{-4}$ | 9.92 | $4.77 \times 10^{-8}$ | ✓ | $-0.00162$ | 0.00261 |
| hAQP5 + 10 ng hCA II | $H_2O$ + Tris | $-0.00214$ | $2.46 \times 10^{-4}$ | 12.3 | $3.42 \times 10^{-8}$ | ✓ | 0.00103 | 0.00527 |
| hAQP5 + 10 ng hCA II | hAQP5 + Tris | $-0.00104$ | $2.32 \times 10^{-4}$ | 6.32 | $7.41 \times 10^{-4}$ | ✓ | 0.00106 | 0.00529 |
| hAQP5 + 10 ng hCA II | $H_2O$ + 1 ng hCA II | $-0.00197$ | $2.46 \times 10^{-4}$ | 11.3 | $4.18 \times 10^{-8}$ | ✓ | $-7.59 \times 10^{-4}$ | 0.00347 |
| hAQP5 + 10 ng hCA II | hAQP5 + 1 ng hCA II | $-4.54 \times 10^{-4}$ | $2.13 \times 10^{-4}$ | 3.01 | 0.408 | – | $-0.00119$ | 0.00304 |
| hAQP5 + 10 ng hCA II | $H_2O$ + 10ng hCA II | $-0.00190$ | $2.32 \times 10^{-4}$ | 11.6 | $4.03 \times 10^{-8}$ | ✓ | $-0.00168$ | 0.00255 |
| $H_2O$ + 100 ng hCA II | $H_2O$ + Tris | $-2.07 \times 10^{-4}$ | $2.62 \times 10^{-4}$ | 1.12 | 0.993 | – | $7.84 \times 10^{-4}$ | 0.00502 |
| $H_2O$ + 100ng hCA II | hAQP5 + Tris | $8.94 \times 10^{-4}$ | $2.49 \times 10^{-4}$ | 5.07 | 0.0134 | ✓ | $8.11 \times 10^{-4}$ | 0.00504 |
| $H_2O$ + 100 ng hCA II | $H_2O$ + 1 ng hCA II | $-3.97 \times 10^{-5}$ | $2.62 \times 10^{-4}$ | 0.215 | 1.000 | – | $-0.00101$ | 0.00322 |
| $H_2O$ + 100 ng hCA II | hAQP5 + 1ng hCA II | 0.00148 | $2.32 \times 10^{-4}$ | 9.03 | $4.89 \times 10^{-7}$ | ✓ | $-0.00144$ | 0.00279 |
| $H_2O$ + 100ng hCA II | $H_2O$ + 10ng hCA II | $3.03 \times 10^{-5}$ | $2.49 \times 10^{-4}$ | 0.172 | 1.000 | – | $-0.00193$ | 0.00230 |
| $H_2O$ + 100 ng hCA II | hAQP5 + 10 ng hCA II | 0.00193 | $2.55 \times 10^{-4}$ | 10.7 | $4.78 \times 10^{-8}$ | ✓ | $-0.00237$ | 0.00187 |
| hAQP5 +100 ng hCA II | $H_2O$ + Tris | $-0.00317$ | $2.31 \times 10^{-4}$ | 19.4 | 0.00 | ✓ | $2.19 \times 10^{-5}$ | 0.00425 |
| hAQP5 +100 ng hCA II | hAQP5 + Tris | $-0.00207$ | $2.17 \times 10^{-4}$ | 13.5 | $2.85 \times 10^{-8}$ | ✓ | $4.94 \times 10^{-5}$ | 0.00428 |
| hAQP5 +100 ng hCA II | $H_2O$ + 1 ng hCA II | $-0.00301$ | $2.31 \times 10^{-4}$ | 18.4 | 0.00 | ✓ | $-0.00177$ | 0.00246 |
| hAQP5 +100 ng hCA II | hAQP5 + 1 ng hCA II | $-0.00149$ | $1.96 \times 10^{-4}$ | 10.7 | $4.74 \times 10^{-8}$ | ✓ | $-0.00220$ | 0.00203 |
| hAQP5 +100 ng hCA II | $H_2O$ + 10 ng hCA II | $-0.00294$ | $2.17 \times 10^{-4}$ | 19.2 | 0.00 | ✓ | $-0.00270$ | 0.00154 |
| hAQP5 +100 ng hCA II | hAQP5 + 10 ng hCA II | $-0.00104$ | $2.23 \times 10^{-4}$ | 6.56 | $3.95 \times 10^{-4}$ | ✓ | $-0.00313$ | 0.00110 |
| hAQP5 +100 ng hCA II | $H_2O$ + 100 ng hCA II | $-0.00297$ | $2.41 \times 10^{-4}$ | 17.4 | 0.00 | ✓ | $-0.00288$ | 0.00135 |

*Notes*: Threshold for significance a = 0.05. Where the displayed P-values is 0.00, this indicates a values less than $<2.22 \times 10^{-308}$. $2.22 \times 10^{-308}$ is the smallest possible value for double type data that a 64-bit system is able to distinguish.
Abbreviations: DF, degrees of freedom; LCL, lower confidence limit; mean diff., mean difference for each comparison; UCL, upper confidence limit.

**Table A5B. Statistics summary for Fig. 5*B*: (i) descriptive statistics for data presented in the figure panel, $(dpH_S/dt)_{Max}$ upon $CO_2$ removal; (ii) overall one-way ANOVA results for data presented in Fig. 5*A*; (iii) Tukey's means comparison analysis**

i. Descriptive statistics

|  | $n$ | Mean | SD | SEM |
|---|---|---|---|---|
| $H_2O$ + Tris | 8 | $4.13\times10^{-4}$ | $2.42\times10^{-4}$ | $8.55\times10^{-5}$ |
| hAQP5 + Tris | 10 | 0.00136 | $8.19\times10^{-4}$ | $2.59\times10^{-4}$ |
| $H_2O$ + 1 ng hCA II | 8 | $5.84\times10^{-4}$ | $3.49\times10^{-4}$ | $1.23\times10^{-4}$ |
| hAQP5 + 1 ng hCA II | 15 | 0.00151 | $6.52\times10^{-4}$ | $1.68\times10^{-4}$ |
| $H_2O$ + 10 ng hCA II | 10 | $6.17\times10^{-4}$ | $1.45\times10^{-4}$ | $4.60\times10^{-5}$ |
| hAQP5 + 10 ng hCA II | 9 | 0.00157 | $6.00\times10^{-4}$ | $2.00\times10^{-4}$ |
| $H_2O$ + 100 ng hCA II | 7 | $6.33\times10^{-4}$ | $1.66\times10^{-4}$ | $6.28\times10^{-5}$ |
| hAQP5 + 100 ng hCA II | 12 | 0.00224 | $7.43\times10^{-4}$ | $2.14\times10^{-4}$ |

ii. Overall ANOVA

| Variation | DF | Sum of squares | Mean square | $F$-value | $P$-value |
|---|---|---|---|---|---|
| Between groups | 7 | $2.96\times10^{-5}$ | $4.23\times10^{-6}$ | 13.3 | $7.77\times10^{-11}$ |
| Within groups | 71 | $2.26\times10^{-5}$ | $3.18\times10^{-7}$ |  |  |
| Total | 78 | $5.22\times10^{-5}$ |  |  |  |

iii. Tukey's test

| Condition A | Condition B | Mean diff. | SEM | $Q$-value | $P$-value | Sig. | LCL | UCL |
|---|---|---|---|---|---|---|---|---|
| hAQP5 + Tris | $H_2O$ + Tris | $9.49\times10^{-4}$ | $2.67\times10^{-4}$ | 5.02 | 0.0151 | ✓ | $1.14\times10^{-4}$ | 0.00178 |
| $H_2O$ + 1 ng hCA II | $H_2O$ + Tris | $1.71\times10^{-4}$ | $2.82\times10^{-4}$ | 0.859 | 0.999 | – | $-7.09\times10^{-4}$ | 0.00105 |
| $H_2O$ + 1 ng hCA II | hAQP5 + Tris | $-7.78\times10^{-4}$ | $2.67\times10^{-4}$ | 4.11 | 0.0862 | – | $-0.00161$ | $5.76\times10^{-5}$ |
| hAQP5 + 1 ng hCA II | $H_2O$ + Tris | 0.00110 | $2.47\times10^{-4}$ | 6.30 | $7.67\times10^{-4}$ | ✓ | $3.29\times10^{-4}$ | 0.00187 |
| hAQP5 + 1 ng hCA II | hAQP5 + Tris | $1.51\times10^{-4}$ | $2.30\times10^{-4}$ | 0.928 | 0.998 | – | $-5.68\times10^{-4}$ | $8.70\times10^{-4}$ |
| hAQP5 + 1 ng hCA II | $H_2O$ + 1ng hCA II | $9.29\times10^{-4}$ | $2.47\times10^{-4}$ | 5.32 | 0.00783 | ✓ | $1.58\times10^{-4}$ | 0.00170 |
| $H_2O$ + 10 ng hCA II | $H_2O$ + Tris | $2.04\times10^{-4}$ | $2.67\times10^{-4}$ | 1.08 | 0.995 | – | $-6.31\times10^{-4}$ | 0.00104 |
| $H_2O$ + 10 ng hCA II | hAQP5 + Tris | $-7.45\times10^{-4}$ | $2.52\times10^{-4}$ | 4.18 | 0.0768 | – | $-0.00153$ | $4.23\times10^{-5}$ |
| $H_2O$ + 10 ng hCA II | $H_2O$ + 1 ng hCA II | $3.25\times10^{-5}$ | $2.67\times10^{-4}$ | 0.172 | 1.000 | – | $-8.03\times10^{-4}$ | $8.68\times10^{-4}$ |
| $H_2O$ + 10 ng hCA II | hAQP5 + 1 ng hCA II | $-8.96\times10^{-4}$ | $2.30\times10^{-4}$ | 5.51 | 0.00516 | ✓ | $-0.00161$ | $-1.77\times10^{-4}$ |
| hAQP5 + 10 ng hCA II | $H_2O$ + Tris | 0.00116 | $2.74\times10^{-4}$ | 5.99 | 0.00166 | ✓ | $3.04\times10^{-4}$ | 0.00202 |
| hAQP5 + 10 ng hCA II | hAQP5 + Tris | $2.11\times10^{-4}$ | $2.59\times10^{-4}$ | 1.15 | 0.992 | – | $-5.98\times10^{-4}$ | 0.00102 |
| hAQP5 + 10 ng hCA II | $H_2O$ + 1 ng hCA II | $9.89\times10^{-4}$ | $2.74\times10^{-4}$ | 5.11 | 0.0125 | ✓ | $1.33\times10^{-4}$ | 0.00184 |
| hAQP5 + 10 ng hCA II | hAQP5 + 1 ng hCA II | $6.02\times10^{-5}$ | $2.38\times10^{-4}$ | 0.358 | 1.000 | – | $-6.82\times10^{-4}$ | $8.03\times10^{-4}$ |
| hAQP5 + 10 ng hCA II | $H_2O$ + 10ng hCA II | $9.56\times10^{-4}$ | $2.59\times10^{-4}$ | 5.22 | 0.00975 | ✓ | $1.47\times10^{-4}$ | 0.00177 |
| $H_2O$ + 100 ng hCA II | $H_2O$ + Tris | $2.20\times10^{-4}4$ | $2.92\times10^{-4}$ | 1.07 | 0.995 | – | $-6.91\times10^{-4}$ | 0.00113 |
| $H_2O$ + 100ng hCA II | hAQP5 + Tris | $-7.28\times10^{-4}$ | $2.78\times10^{-4}$ | 3.71 | 0.165 | – | $-0.00160$ | $1.39\times10^{-4}$ |
| $H_2O$ + 100 ng hCA II | $H_2O$ + 1 ng hCA II | $4.91\times10^{-5}$ | $2.92\times10^{-4}$ | 0.238 | 1.000 | – | $-8.62\times10^{-4}$ | $9.60\times10^{-4}$ |
| $H_2O$ + 100 ng hCA II | hAQP5 + 1ng hCA II | $-8.79\times10^{-4}$ | $2.58\times10^{-4}$ | 4.82 | 0.0228 | ✓ | $-0.00169$ | $-7.36E-5$ |
| $H_2O$ + 100ng hCA II | $H_2O$ + 10ng hCA II | $1.66\times10^{-5}$ | $2.78\times10^{-4}$ | 0.0847 | 1.00 | – | $-8.51\times10^{-4}$ | $8.84\times10^{-5}$ |
| $H_2O$ + 100 ng hCA II | hAQP5 + 10 ng hCA II | $-9.40\times10^{-4}$ | $2.84\times10^{-4}$ | 4.68 | 0.0303 | ✓ | $-0.00183$ | $-5.25\times10^{-5}$ |
| hAQP5 +100 ng hCA II | $H_2O$ + Tris | 0.00183 | $2.57\times10^{-4}$ | 10.0 | $3.68\times10^{-8}$ | ✓ | 0.00102 | 0.00263 |
| hAQP5 +100 ng hCA II | hAQP5 + Tris | $8.79\times10^{-4}$ | $2.41\times10^{-4}$ | 5.15 | 0.0114 | ✓ | $1.25\times10^{-4}$ | 0.00163 |
| hAQP5 +100 ng hCA II | $H_2O$ + 1 ng hCA II | 0.00166 | $2.57\times10^{-4}$ | 9.11 | $4.03\times10^{-7}$ | ✓ | $8.53\times10^{-4}$ | 0.00246 |
| hAQP5 +100 ng hCA II | hAQP5 + 1 ng hCA II | $7.28\times10^{-4}$ | $2.18\times10^{-4}$ | 4.72 | 0.0280 | ✓ | $4.63\times10^{-5}$ | 0.00141 |
| hAQP5 +100 ng hCA II | $H_2O$ + 10 ng hCA II | 0.00162 | $2.41\times10^{-4}$ | 9.52 | $1.21\times10^{-7}$ | ✓ | $8.70\times10^{-4}$ | 0.00238 |
| hAQP5 +100 ng hCA II | hAQP5 + 10 ng hCA II | $6.68\times10^{-4}$ | $2.49\times10^{-4}$ | 3.80 | 0.144 | – | $-1.08\times10^{-4}$ | 0.00144 |
| hAQP5 +100 ng hCA II | $H_2O$ + 100 ng hCA II | 0.00161 | $2.68\times10^{-4}$ | 8.48 | $2.16\times10^{-6}$ | – | $7.70\times10^{-4}$ | 0.00244 |

*Notes*: Threshold for significance a = 0.05. Where the displayed P-values is 0.00, this indicates a values less than $<2.22\times10^{-308}$. $2.22\times10^{-308}$ is the smallest possible value for double type data that a 64-bit system is able to distinguish.

Abbreviations: LCL, lower confidence limit; mean diff., mean difference for each comparison; UCL, upper confidence limit

**Table A6A. Statistics summary for Fig. 6A: (i) descriptive statistics for data presented in the figure panel, initial pH$_i$; (ii) overall one-way ANOVA results; (iii) Tukey's means comparison analysis**

i. Descriptive statistics

|  | n | Mean | SD | SEM |
|---|---|---|---|---|
| H$_2$O + Tris | 8 | 7.27 | 0.0486 | 0.0172 |
| hAQP5 + Tris | 8 | 7.31 | 0.0512 | 0.0181 |
| H$_2$O + 1 ng hCA II | 8 | 7.30 | 0.0614 | 0.0217 |
| hAQP5 + 1 ng hCA II | 8 | 7.32 | 0.0512 | 0.0181 |
| H$_2$O + 10 ng hCA II | 8 | 7.26 | 0.0452 | 0.0160 |
| hAQP5 + 10 ng hCA II | 8 | 7.31 | 0.0352 | 0.0124 |
| H$_2$O + 100 ng hCA II | 8 | 7.26 | 0.0614 | 0.0217 |
| hAQP5 + 100 ng hCA II | 8 | 7.29 | 0.0593 | 0.0210 |

ii. Overall ANOVA

| Variation | DF | Sum of squares | Mean square | *F*-value | *P*-value |
|---|---|---|---|---|---|
| Between groups | 7 | 0.0324 | 0.00462 | 1.68 | 0.131 |
| Within groups | 56 | 0.154 | 0.00274 |  |  |
| Total | 63 | 0.186 |  |  |  |

iii. Tukey's test

| Condition A | Condition B | Mean diff. | *Q*-value | *P*-value | Sig. | LCL | UCL |
|---|---|---|---|---|---|---|---|
| hAQP5 + Tris | H$_2$O + Tris | 0.0440 | 2.37 | 0.701 | − | −0.0385 | 0.126 |
| H$_2$O + 1 ng hCA II | H$_2$O + Tris | 0.0264 | 1.43 | 0.971 | − | −0.0561 | 0.109 |
| H$_2$O + 1 ng hCA II | hAQP5 + Tris | −0.0176 | 0.948 | 0.997 | − | −0.100 | 0.0649 |
| hAQP5 + 1 ng hCA II | H$_2$O + Tris | 0.0515 | 2.78 | 0.512 | − | −0.0309 | 0.134 |
| hAQP5 + 1 ng hCA II | hAQP5 + Tris | 0.00759 | 0.410 | 1.000 | − | −0.0749 | 0.0900 |
| hAQP5 + 1 ng hCA II | H$_2$O + 1 ng hCA II | 0.0252 | 1.36 | 0.978 | − | −0.0573 | 0.108 |
| H$_2$O + 10 ng hCA II | H$_2$O + Tris | −0.00884 | 0.477 | 1.000 | − | −0.0913 | 0.0736 |
| H$_2$O + 10 ng hCA II | hAQP5 + Tris | −0.0528 | 2.85 | 0.481 | − | −0.135 | 0.0297 |
| H$_2$O + 10 ng hCA II | H$_2$O + 1 ng hCA II | −0.0352 | 1.90 | 0.877 | − | −0.118 | 0.0472 |
| H$_2$O + 10 ng hCA II | hAQP5 + 1 ng hCA II | −0.0604 | 3.26 | 0.309 | − | −0.143 | 0.0221 |
| hAQP5 + 10 ng hCA II | H$_2$O + Tris | 0.0439 | 2.37 | 0.703 | − | −0.0386 | 0.126 |
| hAQP5 + 10 ng hCA II | hAQP5 + Tris | $-1.08 \times 10^{-4}$ | 0.00586 | 1.00 | − | −0.0826 | 0.0824 |
| hAQP5 + 10 ng hCA II | H$_2$O + 1 ng hCA II | 0.0175 | 0.943 | 0.998 | − | −0.0650 | 0.0999 |
| hAQP5 + 10 ng hCA II | hAQP5 + 1 ng hCA II | −0.00770 | 0.416 | 1.000 | − | −0.0902 | 0.0748 |
| hAQP5 + 10 ng hCA II | H$_2$O + 10ng hCA II | 0.0527 | 2.84 | 0.484 | − | −0.0298 | 0.135 |
| H$_2$O + 100 ng hCA II | H$_2$O + Tris | −0.00531 | 0.287 | 1.000 | − | −0.0878 | 0.0772 |
| H$_2$O + 100ng hCA II | hAQP5 + Tris | −0.0493 | 2.66 | 0.569 | − | −0.132 | 0.0332 |
| H$_2$O + 100 ng hCA II | H$_2$O + 1 ng hCA II | −0.0317 | 1.71 | 0.925 | − | −0.114 | 0.0508 |
| H$_2$O + 100 ng hCA II | hAQP5 + 1 ng hCA II | −0.0569 | 3.07 | 0.385 | − | −0.139 | 0.0256 |
| H$_2$O + 100ng hCA II | H$_2$O + 10 ng hCA II | 0.00353 | 0.191 | 1.000 | − | −0.0789 | 0.0860 |
| H$_2$O + 100 ng hCA II | hAQP5 + 10 ng hCA II | −0.0492 | 2.65 | 0.572 | − | −0.132 | 0.0333 |
| hAQP5 +100 ng hCA II | H$_2$O + Tris | 0.0177 | 0.957 | 0.997 | − | −0.0647 | 0.100 |
| hAQP5 +100 ng hCA II | hAQP5 + Tris | −0.0262 | 1.42 | 0.972 | − | −0.109 | 0.0562 |
| hAQP5 +100 ng hCA II | H$_2$O + 1 ng hCA II | −0.00866 | 0.468 | 1.000 | − | −0.0911 | 0.0738 |
| hAQP5 +100 ng hCA II | hAQP5 + 1 ng hCA II | −0.0338 | 1.83 | 0.898 | − | −0.116 | 0.0486 |
| hAQP5 +100 ng hCA II | H$_2$O + 10 ng hCA II | 0.0266 | 1.43 | 0.970 | − | −0.0559 | 0.109 |
| hAQP5 +100 ng hCA II | hAQP5 + 10 ng hCA II | −0.0261 | 1.41 | 0.973 | − | −0.109 | 0.0563 |
| hAQP5 +100 ng hCA II | H$_2$O + 100 ng hCA II | 0.0230 | 1.24 | 0.987 | − | −0.0594 | 0.105 |

*Notes*: Threshold for significance a = 0.05. SEM = 0.0262.
Abbreviations: DF, degrees of freedom; LCL, lower confidence limit; mean diff., mean difference for each comparison; UCL, upper confidence limit.

**Table A6B. Statistics summary for Fig. 6*B*: (i) descriptive statistics for data presented in the figure panel, $\Delta pH_i$; (ii) overall one-way ANOVA results; (iii) Tukey's means comparison analysis**

i. Descriptive statistics

|  | *n* | Mean | SD | SEM |
|---|---|---|---|---|
| $H_2O$ + Tris | 8 | 0.251 | 0.0220 | 0.00778 |
| hAQP5 + Tris | 8 | 0.257 | 0.0253 | 0.00894 |
| $H_2O$ + 1 ng hCA II | 8 | 0.252 | 0.0296 | 0.0105 |
| hAQP5 + 1 ng hCA II | 8 | 0.274 | 0.0234 | 0.00828 |
| $H_2O$ + 10 ng hCA II | 8 | 0.264 | 0.0209 | 0.00740 |
| hAQP5 + 10 ng hCA II | 8 | 0.292 | 0.0238 | 0.00841 |
| $H_2O$ + 100 ng hCA II | 8 | 0.252 | 0.0254 | 0.00898 |
| hAQP5 + 100 ng hCA II | 8 | 0.278 | 0.0422 | 0.0149 |

ii. Overall ANOVA table

| Variation | DF | Sum of squares | Mean square | *F*-value | *P*-value |
|---|---|---|---|---|---|
| Between groups | 7 | 0.0125 | 0.00178 | 2.39 | 0.0329 |
| Within groups | 56 | 0.0419 | $7.47\times10^{-4}$ |  |  |
| Total | 63 | 0.0543 |  |  |  |

iii. Tukey's test

| Condition A | Condition B | Mean diff. | *Q*-value | *P*-value | Sig. | LCL | UCL |
|---|---|---|---|---|---|---|---|
| hAQP5 + Tris | $H_2O$ + Tris | 0.00615 | 0.636 | 1.000 | – | −0.0369 | 0.0492 |
| $H_2O$ + 1 ng hCA II | $H_2O$ + Tris | 0.00173 | 0.179 | 1.000 | – | −0.0413 | 0.0448 |
| $H_2O$ + 1 ng hCA II | hAQP5 + Tris | −0.00442 | 0.457 | 1.000 | – | −0.0475 | 0.0386 |
| hAQP5 + 1 ng hCA II | $H_2O$ + Tris | 0.0234 | 2.42 | 0.679 | – | −0.0196 | 0.0664 |
| hAQP5 + 1 ng hCA II | hAQP5 + Tris | 0.0173 | 1.79 | 0.909 | – | −0.0258 | 0.0603 |
| hAQP5 + 1 ng hCA II | $H_2O$ + 1 ng hCA II | 0.0217 | 2.24 | 0.757 | – | −0.0214 | 0.0647 |
| $H_2O$ + 10 ng hCA II | $H_2O$ + Tris | 0.0129 | 1.34 | 0.980 | – | −0.0301 | 0.0560 |
| $H_2O$ + 10 ng hCA II | hAQP5 + Tris | 0.00677 | 0.700 | 1.000 | – | −0.0363 | 0.0498 |
| $H_2O$ + 10 ng hCA II | $H_2O$ + 1 ng hCA II | 0.0112 | 1.16 | 0.991 | – | −0.0319 | 0.0542 |
| $H_2O$ + 10 ng hCA II | hAQP5 + 1 ng hCA II | −0.0105 | 1.09 | 0.994 | – | −0.0535 | 0.0325 |
| hAQP5 + 10 ng hCA II | $H_2O$ + Tris | 0.0409 | 4.23 | 0.0737 | – | −0.00212 | 0.0840 |
| hAQP5 + 10 ng hCA II | hAQP5 + Tris | 0.0348 | 3.60 | 0.199 | – | −0.00828 | 0.0778 |
| hAQP5 + 10 ng hCA II | $H_2O$ + 1 ng hCA II | 0.0392 | 4.05 | 0.0995 | – | −0.00386 | 0.0822 |
| hAQP5 + 10 ng hCA II | hAQP5 + 1 ng hCA II | 0.0175 | 1.81 | 0.902 | – | −0.0255 | 0.0605 |
| hAQP5 + 10 ng hCA II | $H_2O$ + 10ng hCA II | 0.0280 | 2.90 | 0.460 | – | −0.0150 | 0.0710 |
| $H_2O$ + 100 ng hCA II | $H_2O$ + Tris | 0.00154 | 0.160 | 1.000 | – | −0.0415 | 0.0446 |
| $H_2O$ + 100ng hCA II | hAQP5 + Tris | −0.00461 | 0.477 | 1.000 | – | −0.0476 | 0.0384 |
| $H_2O$ + 100 ng hCA II | $H_2O$ + 1 ng hCA II | $-1.90\times10^{-4}$ | 0.0197 | 1.00 | – | −0.0432 | 0.0428 |
| $H_2O$ + 100 ng hCA II | hAQP5 + 1ng hCA II | −0.0219 | 2.26 | 0.748 | – | −0.0649 | 0.0212 |
| $H_2O$ + 100ng hCA II | $H_2O$ + 10ng hCA II | −0.0114 | 1.18 | 0.990 | – | −0.0544 | 0.0317 |
| $H_2O$ + 100 ng hCA II | hAQP5 + 10 ng hCA II | −0.0394 | 4.07 | 0.0963 | – | −0.0824 | 0.00367 |
| hAQP5 +100 ng hCA II | $H_2O$ + Tris | 0.0276 | 2.85 | 0.480 | – | −0.0155 | 0.0706 |
| hAQP5 +100 ng hCA II | hAQP5 + Tris | 0.0214 | 2.22 | 0.767 | – | −0.0216 | 0.0645 |
| hAQP5 +100 ng hCA II | $H_2O$ + 1 ng hCA II | 0.0258 | 2.67 | 0.563 | – | −0.0172 | 0.0689 |
| hAQP5 +100 ng hCA II | hAQP5 + 1 ng hCA II | 0.00417 | 0.431 | 1.000 | – | −0.0389 | 0.0472 |
| hAQP5 +100 ng hCA II | $H_2O$ + 10 ng hCA II | 0.0147 | 1.52 | 0.960 | – | −0.0284 | 0.0577 |
| hAQP5 +100 ng hCA II | hAQP5 + 10 ng hCA II | −0.0133 | 1.38 | 0.976 | – | −0.0564 | 0.0297 |
| hAQP5 +100 ng hCA II | $H_2O$ + 100 ng hCA II | 0.0260 | 2.69 | 0.554 | – | −0.0170 | 0.0691 |

*Notes*: Threshold for significance a = 0.05. SEM = 0.0137.
Abbreviations: DF, degrees of freedom; LCL, lower confidence limit; mean diff., mean difference for each comparison; UCL, upper confidence limit.

**Table A6C. Statistics summary for Fig. 6C: (i) descriptive statistics for data presented in the figure panel, $\beta_i$. (ii) overall one-way ANOVA results; (iii) Tukey's means comparison analysis**

### i. Descriptive statistics

|  | *n* | Mean | SD | SEM |
|---|---|---|---|---|
| $H_2O$ + Tris | 8 | 13.4 | 2.20 | 0.779 |
| hAQP5 + Tris | 8 | 14.2 | 2.14 | 0.758 |
| $H_2O$ + 1 ng hCA II | 8 | 14.1 | 2.07 | 0.731 |
| hAQP5 + 1 ng hCA II | 8 | 12.9 | 1.21 | 0.427 |
| $H_2O$ + 10 ng hCA II | 8 | 12.1 | 2.03 | 0.718 |
| hAQP5 + 10 ng hCA II | 8 | 11.5 | 1.23 | 0.435 |
| $H_2O$ + 100 ng hCA II | 8 | 13.1 | 2.26 | 0.799 |
| hAQP5 + 100 ng hCA II | 8 | 11.8 | 1.57 | 0.556 |

### ii. Overall ANOVA table

| Variation | DF | Sum of squares | Mean square | *F*-value | *P*-value |
|---|---|---|---|---|---|
| Between groups | 7 | 57.6 | 8.22 | 2.32 | 0.0379 |
| Within groups | 56 | 199 | 3.55 | | |
| Total | 63 | 256 | | | |

### iii. Tukey's test

| Condition A | Condition B | Mean diff. | Q-value | P-value | Sig. | LCL | UCL |
|---|---|---|---|---|---|---|---|
| hAQP5 + Tris | $H_2O$ + Tris | 0.873 | 1.31 | 0.982 | – | −2.09 | 3.84 |
| $H_2O$ + 1 ng hCA II | $H_2O$ + Tris | 0.699 | 1.05 | 0.995 | – | −2.27 | 3.67 |
| $H_2O$ + 1 ng hCA II | hAQP5 + Tris | −0.173 | 0.260 | 1.000 | – | −3.14 | 2.79 |
| hAQP5 + 1 ng hCA II | $H_2O$ + Tris | −0.414 | 0.621 | 1.000 | – | −3.38 | 2.55 |
| hAQP5 + 1 ng hCA II | hAQP5 + Tris | −1.29 | 1.93 | 0.869 | – | −4.25 | 1.68 |
| hAQP5 + 1 ng hCA II | $H_2O$ + 1 ng hCA II | −1.11 | 1.67 | 0.934 | – | −4.08 | 1.85 |
| $H_2O$ + 10 ng hCA II | $H_2O$ + Tris | −1.27 | 1.91 | 0.876 | – | −4.24 | 1.70 |
| $H_2O$ + 10 ng hCA II | hAQP5 + Tris | −2.14 | 3.22 | 0.326 | – | −5.11 | 0.824 |
| $H_2O$ + 10 ng hCA II | $H_2O$ + 1 ng hCA II | −1.97 | 2.96 | 0.434 | – | −4.93 | 0.997 |
| $H_2O$ + 10 ng hCA II | hAQP5 + 1 ng hCA II | −0.856 | 1.28 | 0.984 | – | −3.82 | 2.11 |
| hAQP5 + 10 ng hCA II | $H_2O$ + Tris | −1.87 | 2.80 | 0.504 | – | −4.83 | 1.10 |
| hAQP5 + 10 ng hCA II | hAQP5 + Tris | −2.74 | 4.11 | 0.0905 | – | −5.70 | 0.227 |
| hAQP5 + 10 ng hCA II | $H_2O$ + 1 ng hCA II | −2.57 | 3.85 | 0.137 | – | −5.53 | 0.401 |
| hAQP5 + 10 ng hCA II | hAQP5 + 1 ng hCA II | −1.45 | 2.18 | 0.782 | – | −4.42 | 1.51 |
| hAQP5 + 10 ng hCA II | $H_2O$ + 10ng hCA II | −0.597 | 0.895 | 0.998 | – | −3.56 | 2.37 |
| $H_2O$ + 100 ng hCA II | $H_2O$ + Tris | −0.261 | 0.392 | 1.000 | – | −3.23 | 2.71 |
| $H_2O$ + 100ng hCA II | hAQP5 + Tris | −1.13 | 1.70 | 0.928 | – | −4.10 | 1.83 |
| $H_2O$ + 100 ng hCA II | $H_2O$ + 1 ng hCA II | −0.960 | 1.44 | 0.970 | – | −3.93 | 2.01 |
| $H_2O$ + 100 ng hCA II | hAQP5 + 1ng hCA II | 0.153 | 0.229 | 1.000 | – | −2.81 | 3.12 |
| $H_2O$ + 100ng hCA II | $H_2O$ + 10ng hCA II | 1.01 | 1.51 | 0.960 | – | −1.96 | 3.97 |
| $H_2O$ + 100 ng hCA II | hAQP5 + 10 ng hCA II | 1.60 | 2.41 | 0.685 | – | −1.36 | 4.57 |
| hAQP5 +100 ng hCA II | $H_2O$ + Tris | −1.55 | 2.33 | 0.720 | – | −4.52 | 1.41 |
| hAQP5 +100 ng hCA II | hAQP5 + Tris | −2.42 | 3.64 | 0.188 | – | −5.39 | 0.542 |
| hAQP5 +100 ng hCA II | $H_2O$ + 1 ng hCA II | −2.25 | 3.38 | 0.267 | – | −5.22 | 0.715 |
| hAQP5 +100 ng hCA II | hAQP5 + 1 ng hCA II | −1.14 | 1.71 | 0.926 | – | −4.10 | 1.83 |
| hAQP5 +100 ng hCA II | $H_2O$ + 10 ng hCA II | −0.282 | 0.424 | 1.000 | – | −3.25 | 2.68 |
| hAQP5 +100 ng hCA II | hAQP5 + 10 ng hCA II | 0.314 | 0.472 | 1.000 | – | −2.65 | 3.28 |
| hAQP5 +100 ng hCA II | $H_2O$ + 100 ng hCA II | −1.29 | 1.94 | 0.867 | – | −4.26 | 1.68 |

*Notes*: Threshold for significance a = 0.05. SEM = 0.942.
Abbreviations: DF, degrees of freedom; LCL, lower confidence limit; mean diff., mean difference for each comparison; UCL, upper confidence limit.

**Table A6D. Statistics summary for Fig. 6*D*: (i) descriptive statistics for data presented in the figure panel, $P_f$; (ii) overall one-way ANOVA results; (iii) Tukey's means comparison analysis**

**i. Descriptive statistics**

|  | $n$ | Mean | SD | SEM |
|---|---|---|---|---|
| $H_2O$ + Tris | 8 | $1.60\times10^{-4}$ | $9.95\times10^{-5}$ | $3.52\times10^{-5}$ |
| hAQP5 + Tris | 8 | 0.00373 | $4.41\times10^{-4}$ | $1.56\times10^{-4}$ |
| $H_2O$ + 1 ng hCA II | 8 | $1.90\times10^{-4}$ | $1.10\times10^{-4}$ | $3.90\times10^{-5}$ |
| hAQP5 + 1 ng hCA II | 8 | 0.00376 | $4.48\times10^{-4}$ | $1.58\times10^{-4}$ |
| $H_2O$ + 10 ng hCA II | 8 | $2.35\times10^{-4}$ | $2.01\times10^{-4}$ | $7.11\times10^{-5}$ |
| hAQP5 + 10 ng hCA II | 8 | 0.00388 | $5.32\times10^{-4}$ | $1.88\times10^{-4}$ |
| $H_2O$ + 100 ng hCA II | 8 | $1.82\times10^{-4}$ | $1.50\times10^{-4}$ | $5.31\times10^{-5}$ |
| hAQP5 + 100 ng hCA II | 8 | 0.00378 | $3.80\times10^{-4}$ | $1.34\times10^{-4}$ |

**ii. Overall ANOVA table**

| Variation | DF | Sum of squares | Mean square | *F*-value | *P*-value |
|---|---|---|---|---|---|
| Between groups | 7 | $2.07\times10^{-4}$ | $2.96\times10^{-5}$ | 261 | $2.53\times10^{-4}$ |
| Within groups | 56 | $6.35\times10^{-6}$ | $1.13\times10^{-7}$ | | |
| Total | 63 | $2.13\times10^{-4}$ | | | |

**iii. Tukey's test**

| Condition A | Condition B | Mean diff. | Q-value | P-value | Sig. | LCL | UCL |
|---|---|---|---|---|---|---|---|
| hAQP5 + Tris | $H_2O$ + Tris | 0.00357 | 30.0 | $6.86\times10^{-8}$ | ✓ | 0.00304 | 0.00410 |
| $H_2O$ + 1 ng hCA II | $H_2O$ + Tris | $3.02\times10^{-5}$ | 0.254 | 1.00 | – | $-5.00\times10^{-4}$ | $5.60\times10^{-4}$ |
| $H_2O$ + 1 ng hCA II | hAQP5 + Tris | −0.00354 | 29.8 | $8.93\times10^{-7}$ | ✓ | −0.00407 | −0.00301 |
| hAQP5 + 1 ng hCA II | $H_2O$ + Tris | 0.00360 | 30.2 | $1.87\times10^{-6}$ | ✓ | 0.00307 | 0.00413 |
| hAQP5 + 1 ng hCA II | hAQP5 + Tris | $2.33\times10^{-5}$ | 0.196 | 1.00 | – | $-5.07\times10^{-4}$ | $5.53\times10^{-4}$ |
| hAQP5 + 1 ng hCA II | $H_2O$ + 1ng hCA II | 0.00357 | 30.0 | $2.79\times10^{-7}$ | ✓ | 0.00304 | 0.00410 |
| $H_2O$ + 10 ng hCA II | $H_2O$ + Tris | $7.45\times10^{-5}$ | 0.626 | 1.00 | – | $-4.56\times10^{-4}$ | $6.05\times10^{-4}$ |
| $H_2O$ + 10 ng hCA II | hAQP5 + Tris | −0.00350 | 29.4 | $8.76\times10^{-7}$ | ✓ | −0.00403 | −0.00297 |
| $H_2O$ + 10 ng hCA II | $H_2O$ + 1 ng hCA II | $4.43\times10^{-5}$ | 0.372 | 1.00 | – | $-4.86\times10^{-4}$ | $5.74\times10^{-4}$ |
| $H_2O$ + 10 ng hCA II | hAQP5 + 1 ng hCA II | −0.00352 | 29.6 | $8.85\times10^{-7}$ | ✓ | −0.00405 | −0.00299 |
| hAQP5 + 10 ng hCA II | $H_2O$ + Tris | 0.00372 | 31.2 | $7.30\times10^{-8}$ | ✓ | 0.00319 | 0.00425 |
| hAQP5 + 10 ng hCA II | hAQP5 + Tris | $1.46\times10^{-4}$ | 1.23 | 0.988 | – | $-3.84\times10^{-4}$ | $6.77\times10^{-4}$ |
| hAQP5 + 10 ng hCA II | $H_2O$ + 1 ng hCA II | 0.00369 | 31.0 | $7.23\times10^{-8}$ | ✓ | 0.00316 | 0.00422 |
| hAQP5 + 10 ng hCA II | hAQP5 + 1 ng hCA II | $1.23\times10^{-4}$ | 1.03 | 0.996 | – | $-4.07\times10^{-4}$ | $6.53\times10^{-4}$ |
| hAQP5 + 10 ng hCA II | $H_2O$ + 10ng hCA II | 0.00364 | 30.6 | $7.16\times10^{-8}$ | ✓ | 0.00311 | 0.00418 |
| $H_2O$ + 100 ng hCA II | $H_2O$ + Tris | $2.18\times10^{-5}$ | 0.183 | 1.000 | – | $-5.08\times10^{-4}$ | $5.52\times10^{-4}$ |
| $H_2O$ + 100ng hCA II | hAQP5 + Tris | −0.00355 | 29.8 | $8.96\times10^{-7}$ | ✓ | −0.00408 | −0.00302 |
| $H_2O$ + 100 ng hCA II | $H_2O$ + 1 ng hCA II | $-8.37\times10^{-6}$ | 0.0703 | 1.00 | – | $-5.38\times10^{-4}$ | $5.22\times10^{-4}$ |
| $H_2O$ + 100 ng hCA II | hAQP5 + 1 ng hCA II | −0.00357 | 30.0 | $6.86\times10^{-8}$ | ✓ | −0.00410 | −0.00304 |
| $H_2O$ + 100ng hCA II | $H_2O$ + 10 ng hCA II | $-5.27\times10^{-5}$ | 0.442 | 1.00 | – | $-5.83\times10^{-4}$ | $4.77\times10^{-4}$ |
| $H_2O$ + 100 ng hCA II | hAQP5 + 10 ng hCA II | −0.00370 | 31.1 | $7.25\times10^{-8}$ | ✓ | −0.00423 | −0.00317 |
| hAQP5 +100 ng hCA II | $H_2O$ + Tris | 0.00362 | 30.4 | $7.08\times10^{-8}$ | ✓ | 0.00309 | 0.00415 |
| hAQP5 +100 ng hCA II | hAQP5 + Tris | $4.36\times10^{-5}$ | 0.366 | 1.00 | – | $-4.87\times10^{-4}$ | $5.74\times10^{-4}$ |
| hAQP5 +100 ng hCA II | $H_2O$ + 1 ng hCA II | 0.00359 | 30.1 | $1.86\times10^{-6}$ | ✓ | 0.00306 | 0.00412 |
| hAQP5 +100 ng hCA II | hAQP5 + 1 ng hCA II | $2.03\times10^{-5}$ | 0.170 | 1.00 | – | $-5.10\times10^{-4}$ | $5.50\times10^{-4}$ |
| hAQP5 +100 ng hCA II | $H_2O$ + 10 ng hCA II | 0.00354 | 29.7 | $8.93\times10^{-7}$ | ✓ | 0.00301 | 0.00407 |
| hAQP5 +100 ng hCA II | hAQP5 + 10 ng hCA II | $-1.03\times10^{-4}$ | 0.864 | 0.999 | – | $-6.33\times10^{-4}$ | $4.27\times10^{-4}$ |
| hAQP5 +100 ng hCA II | $H_2O$ + 100 ng hCA II | 0.00359 | 30.2 | $1.87\times10^{-6}$ | ✓ | 0.00306 | 0.00412 |

*Notes*: Threshold for significance a = 0.05. SEM = $1.68 \times 10^{-4}$.
Abbreviations: DF, degrees of freedom; LCL, lower confidence limit; mean diff., mean difference for each comparison; UCL, upper confidence limit.

**Table A8A. Statistics summary for Fig. 8A: (i) descriptive statistics for data presented in the figure panel, $\triangle pH_S$ upon $CO_2$ addition; (ii) overall one-way ANOVA results; (iii) Tukey's means comparison analysis**

i. Descriptive statistics

|  | $n$ | Mean | SD | SEM |
|---|---|---|---|---|
| $H_2O$ + Tris | 6 | 0.235 | 0.235 | 0.0112 |
| hAQP5 + Tris | 6 | 0.236 | 0.236 | 0.00784 |
| $H_2O$ + 1ng hCA II | 6 | 0.324 | 0.324 | 0.0196 |
| hAQP5 + 1ng hCA II | 6 | 0.347 | 0.347 | 0.0230 |

ii. Overall ANOVA table

| Variation | DF | Sum of squares | Mean square | $F$-value | $P$-value |
|---|---|---|---|---|---|
| Between groups | 3 | 0.0614 | 0.0205 | 12.4 | $8.35 \times 10^{-5}$ |
| Within groups | 20 | 0.0330 | 0.00165 |  |  |
| Total | 23 | 0.0944 |  |  |  |

iii. Tukey's test

| Condition A | Condition B | Mean diff. | $Q$-value | $P$-value | Sig. | LCL | UCL |
|---|---|---|---|---|---|---|---|
| hAQP5 + Tris | $H_2O$ + Tris | 0.00134 | 0.0810 | 1.000 | – | −0.0643 | 0.0670 |
| $H_2O$ + 1 ng hCA II | $H_2O$ + Tris | 0.0889 | 5.36 | 0.00582 | ✓ | 0.0232 | 0.155 |
| $H_2O$ + 1 ng hCA II | hAQP5 + Tris | 0.0875 | 5.28 | 0.00663 | ✓ | 0.0219 | 0.153 |
| hAQP5 + 1 ng hCA II | $H_2O$ + Tris | 0.112 | 6.76 | $6.13 \times 10^{-4}$ | ✓ | 0.0464 | 0.178 |
| hAQP5 + 1 ng hCA II | hAQP5 + Tris | 0.111 | 6.68 | $6.99 \times 10^{-4}$ | ✓ | 0.0451 | 0.176 |
| hAQP5 + 1 ng hCA II | $H_2O$ + 1 ng hCA II | 0.0232 | 1.40 | 0.757 | – | −0.0425 | 0.0888 |

*Notes*: Threshold for significance a = 0.05. SEM = 0.0235.
Abbreviations: DF, degrees of freedom; LCL, lower confidence limit; mean diff., mean difference for each comparison; UCL, upper confidence limit.

**Table A8B. Statistics summary for Fig. 8B: (i) descriptive statistics for data presented in the figure panel, $\triangle pH_S$ upon $CO_2$ removal; (ii) overall one-way ANOVA results; (iii) Tukey's means comparison analysis**

i. Descriptive statistics

|  | $n$ | Mean | SD | SEM |
|---|---|---|---|---|
| $H_2O$ + Tris | 6 | −0.335 | 0.0523 | 0.0213 |
| hAQP5 + Tris | 6 | −0.341 | 0.0314 | 0.0128 |
| $H_2O$ + 1 ng hCA II | 6 | −0.418 | 0.0546 | 0.0223 |
| hAQP5 + 1 ng hCA II | 6 | −0.450 | 0.0531 | 0.0217 |

ii. Overall ANOVA table

| Variation | DF | Sum of squares | Mean square | $F$-value | $P$-value |
|---|---|---|---|---|---|
| Between groups | 3 | 0.0590 | 0.0197 | 8.26 | $9.03 \times 10^{-4}$ |
| Within groups | 20 | 0.0476 | 0.00238 | | |
| Total | 23 | 0.107 | | | |

iii. Tukey's test

| Condition A | Condition B | Mean diff. | $Q$-value | $P$-value | Sig. | LCL | UCL |
|---|---|---|---|---|---|---|---|
| hAQP5 + Tris | $H_2O$ + Tris | −0.00633 | 0.318 | 0.996 | − | −0.0852 | 0.0725 |
| $H_2O$ + 1 ng hCA II | $H_2O$ + Tris | −0.0836 | 4.20 | 0.0352 | ✓ | −0.162 | −0.00478 |
| $H_2O$ + 1 ng hCA II | hAQP5 + Tris | −0.0773 | 3.88 | 0.0560 | − | −0.156 | 0.00156 |
| hAQP5 + 1 ng hCA II | $H_2O$ + Tris | −0.116 | 5.80 | 0.00287 | ✓ | −0.194 | −0.0367 |
| hAQP5 + 1 ng hCA II | hAQP5 + Tris | −0.109 | 5.48 | 0.00478 | ✓ | −0.188 | −0.0304 |
| hAQP5 + 1 ng hCA II | $H_2O$ + 1 ng hCA II | −0.0319 | 1.60 | 0.674 | − | −0.111 | 0.0469 |

*Notes*: Threshold for significance a = 0.05. SEM = 0.0282.
Abbreviations: DF, degrees of freedom; LCL, lower confidence limit; mean diff., mean difference for each comparison; UCL, upper confidence limit.

**Table A8C. Statistics summary for Fig. 8C: (i) descriptive statistics for data presented in the figure panel, $(dpH_i/dt)_{Max}$ upon $CO_2$ addition; (ii) overall one-way ANOVA results; (iii) Tukey's means comparison analysis**

i. Descriptive statistics

|  | $n$ | Mean | SD | SEM |
|---|---|---|---|---|
| $H_2O$ + Tris | 6 | −0.00250 | $7.09 \times 10^{-4}$ | $2.89 \times 10^{-4}$ |
| hAQP5 + Tris | 6 | −0.00451 | 0.00110 | $4.48 \times 10^{-4}$ |
| $H_2O$ + 1 ng hCA II | 6 | −0.00742 | $9.98 \times 10^{-4}$ | $4.08 \times 10^{-4}$ |
| hAQP5 + 1 ng hCA II | 6 | −0.00670 | 0.00165 | $6.73 \times 10^{-4}$ |

ii. Overall ANOVA table

| Variation | DF | Sum of squares | Mean square | $F$-value | $P$-value |
|---|---|---|---|---|---|
| Between groups | 3 | $8.95 \times 10^{-5}$ | $2.98 \times 10^{-5}$ | 22.0 | $1.50 \times 10^{-6}$ |
| Total | 23 | $1.17 \times 10^{-4}$ |  |  |  |

iii. Tukey's test

| Condition A | Condition B | Mean diff. | $Q$-value | $P$-value | Sig. | LCL | UCL |
|---|---|---|---|---|---|---|---|
| hAQP5 + Tris | $H_2O$ + Tris | −0.00201 | 4.24 | 0.0331 | ✓ | −0.00389 | $-1.33 \times 10^{-4}$ |
| $H_2O$ + 1 ng hCA II | $H_2O$ + Tris | −0.00492 | 10.4 | $2.42 \times 10^{-6}$ | ✓ | −0.00680 | −0.00304 |
| $H_2O$ + 1 ng hCA II | hAQP5 + Tris | −0.00291 | 6.12 | 0.00171 | ✓ | −0.00479 | −0.00103 |
| hAQP5 + 1 ng hCA II | $H_2O$ + Tris | −0.00420 | 8.84 | $2.33 \times 10^{-5}$ | ✓ | −0.00608 | −0.00232 |
| hAQP5 + 1 ng hCA II | hAQP5 + Tris | −0.00218 | 4.60 | 0.0192 | ✓ | −0.00407 | $-3.05 \times 10^{-4}$ |
| hAQP5 + 1 ng hCA II | $H_2O$ + 1 ng hCA II | $7.23 \times 10^{-4}$ | 1.52 | 0.707 | − | −0.00116 | 0.00260 |

*Notes*: Threshold for significance a = 0.05. SEM = $6.72 \times 10^{-4}$.

Abbreviations: DF, degrees of freedom; LCL, lower confidence limit; mean diff., mean difference for each comparison; UCL, upper confidence limit.

**Table A8D. Statistics summary for Fig. 8D: (i) descriptive statistics for data presented in the figure panel, $(dpH_i/dt)_{Max}$ upon $CO_2$ removal; (ii) overall one-way ANOVA results; (iii) Tukey's means comparison analysis**

i. Descriptive statistics

|  | *n* | Mean | SD | SEM |
|---|---|---|---|---|
| $H_2O$ + Tris | 6 | 0.00161 | $6.45\times10^{-4}$ | $2.63\times10^{-4}$ |
| hAQP5 + Tris | 6 | 0.00229 | $5.65\times10^{-4}$ | $2.31\times10^{-4}$ |
| $H_2O$ + 1ng hCA II | 6 | 0.00498 | $9.28\times10^{-4}$ | $3.79\times10^{-4}$ |
| hAQP5 + 1ng hCA II | 6 | 0.00406 | 0.00129 | $5.25\times10^{-4}$ |

ii. Overall ANOVA table

| Variation | DF | Sum of squares | Mean square | *F*-value | *P*-value |
|---|---|---|---|---|---|
| Between groups | 3 | $4.36\times10^{-5}$ | $1.45\times10^{-5}$ | 17.9 | $6.92\times10^{-7}$ |
| Within groups | 20 | $1.62\times10^{-5}$ | $8.12\times10^{-7}$ |  |  |
| Total | 23 | $5.99\times10^{-5}$ |  |  |  |

iii. Tukey's test

| Condition A | Condition B | Mean diff. | *Q*-value | *P*-value | Sig. | LCL | UCL |
|---|---|---|---|---|---|---|---|
| hAQP5 + Tris | $H_2O$ + Tris | $6.78\times10^{-4}$ | 1.84 | 0.571 | – | $-7.78\times10^{-4}$ | 0.00213 |
| $H_2O$ + 1 ng hCA II | $H_2O$ + Tris | 0.00337 | 9.16 | $1.42\times10^{-5}$ | ✓ | 0.00192 | 0.00483 |
| $H_2O$ + 1 ng hCA II | hAQP5 + Tris | 0.00269 | 7.32 | $2.49\times10^{-4}$ | ✓ | 0.00124 | 0.00415 |
| hAQP5 + 1 ng hCA II | $H_2O$ + Tris | 0.00245 | 6.67 | $7.08\times10^{-4}$ | ✓ | $9.97\times10^{-4}$ | 0.00391 |
| hAQP5 + 1 ng hCA II | hAQP5 + Tris | 0.00178 | 4.82 | 0.0135 | ✓ | $3.19\times10^{-4}$ | 0.00323 |
| hAQP5 + 1 ng hCA II | $H_2O$ + 1 ng hCA II | $-9.18\times10^{-4}$ | 2.50 | 0.318 | – | $-0.00237$ | $5.38\times10^{-4}$ |

*Notes*: Threshold for significance a = 0.05. SEM = $5.20\times10^{-4}$.
Abbreviations: DF, degrees of freedom; LCL, lower confidence limit; mean diff., mean difference for each comparison; UCL, upper confidence limit.

**Table A9A. Statistics summary for Fig. 9*A*: (i) descriptive statistics for data presented in the figure panel, $(dpH_S/d_t)_{Max}$ upon $CO_2$ addition; (ii) overall one-way ANOVA results. (iii) Tukey's means comparison analysis**

i. Descriptive statistics

|  | $n$ | Mean | SD | SEM |
|---|---|---|---|---|
| $H_2O$ + Tris | 6 | −0.00502 | 0.00137 | $5.58\times10^{-4}$ |
| hAQP5 + Tris | 6 | −0.00541 | 0.00185 | $7.53\times10^{-4}$ |
| $H_2O$ + 1 ng hCA II | 6 | −0.00910 | $9.11\times10^{-4}$ | $3.72\times10^{-4}$ |
| hAQP5 + 1 ng hCA II | 6 | −0.00843 | 0.00133 | $5.43\times10^{-4}$ |

ii. Overall ANOVA table

| Variation | DF | Sum of squares | Mean square | $F$-value | $P$-value |
|---|---|---|---|---|---|
| Between groups | 3 | $7.74\times10^{-5}$ | $2.58\times10^{-5}$ | 13.1 | $5.83\times10^{-5}$ |
| Within groups | 20 | $3.94\times10^{-5}$ | $1.97\times10^{-5}$ | | |
| Total | 23 | $1.17\times10^{-5}$ | | | panel |

iii. Tukey's test

| Condition A | Condition B | Mean diff. | $Q$-value | $P$-value | Sig. | LCL | UCL |
|---|---|---|---|---|---|---|---|
| hAQP5 + Tris | $H_2O$ + Tris | $-3.98\times10^{-4}$ | 0.695 | 0.960 | − | −0.00267 | 0.00187 |
| $H_2O$ + 1 ng hCA II | $H_2O$ + Tris | −0.00408 | 7.13 | $3.38\times10^{-4}$ | ✓ | −0.00635 | −0.00182 |
| $H_2O$ + 1 ng hCA II | hAQP5 + Tris | −0.00369 | 6.43 | 0.00103 | ✓ | −0.00595 | −0.00142 |
| hAQP5 + 1 ng hCA II | $H_2O$ + Tris | −0.00342 | 5.96 | 0.00221 | ✓ | −0.00568 | −0.00115 |
| hAQP5 + 1 ng hCA II | hAQP5 + Tris | −0.00302 | 5.27 | 0.00674 | ✓ | −0.00528 | $-7.50\times10^{-4}$ |
| hAQP5 + 1 ng hCA II | $H_2O$ + 1 ng hCA II | $6.68\times10^{-4}$ | 1.17 | 0.842 | − | −0.00160 | 0.00294 |

*Notes*: Threshold for significance a = 0.05. SEM = $8.10\times10^{-4}$.

Abbreviations: DF, degrees of freedom; LCL, lower confidence limit; mean diff., mean difference for each comparison; UCL, upper confidence limit.

**Table A9B. Statistics summary for Fig. 9*B*: (i) descriptive statistics for data presented in the figure panel, $(dpH_S/dt)_{Max}$ upon $CO_2$ removal; (ii) overall one-way ANOVA results; (iii) Tukey's means comparison analysis.**

### i. Descriptive statistics

|  | *n* | Mean | SD | SEM |
|---|---|---|---|---|
| $H_2O$ + Tris | 6 | 0.00271 | $9.14 \times 10^{-4}$ | $3.73 \times 10^{-4}$ |
| hAQP5 + Tris | 6 | 0.00507 | 0.00180 | $7.34 \times 10^{-4}$ |
| $H_2O$ + 1 ng hCA II | 6 | 0.00364 | 0.00127 | $5.18 \times 10^{-4}$ |
| hAQP5 + 1 ng hCA II | 6 | 0.00479 | 0.00120 | $4.91 \times 10^{-4}$ |

### ii. Overall ANOVA table

| Variation | DF | Sum of squares | Mean square | *F*-value | *P*-value |
|---|---|---|---|---|---|
| Between groups | 3 | $2.13 \times 10^{-5}$ | $7.12 \times 10^{-6}$ | 3.99 | 0.0222 |
| Within groups | 20 | $3.56 \times 10^{-5}$ | $1.78 \times 10^{-6}$ | | |
| Total | 23 | $5.70 \times 10^{-5}$ | | | |

### iii. Tukey's test

| Condition A | Condition B | Mean diff. | *Q*-value | *P*-value | Sig. | LCL | UCL |
|---|---|---|---|---|---|---|---|
| hAQP5 + Tris | $H_2O$ + Tris | 0.00236 | 4.33 | 0.0288 | ✓ | $2.04 \times 10^{-4}$ | 0.00452 |
| $H_2O$ + 1 ng hCA II | $H_2O$ + Tris | $9.36 \times 10^{-4}$ | 1.72 | 0.625 | – | −0.00122 | 0.00309 |
| $H_2O$ + 1 ng hCA II | hAQP5 + Tris | −0.00143 | 2.62 | 0.281 | – | −0.00358 | $7.32 \times 10^{-4}$ |
| hAQP5 + 1 ng hCA II | $H_2O$ + Tris | 0.00209 | 3.83 | 0.0604 | – | $−7.19 \times 10^{-5}$ | 0.00424 |
| hAQP5 + 1 ng hCA II | hAQP5 + Tris | $−2.76 \times 10^{-4}$ | 0.506 | 0.984 | – | −0.00243 | 0.00188 |
| hAQP5 + 1 ng hCA II | $H_2O$ + 1 ng hCA II | 0.00115 | 2.11 | 0.461 | – | −0.00101 | 0.00331 |

*Notes*: Threshold for significance a = 0.05. SEM = $7.71 \times 10^{-4}$.
Abbreviations: DF, degrees of freedom; LCL, lower confidence limit; mean diff., mean difference for each comparison; UCL, upper confidence limit.

**Table A10A. Statistics summary for Fig. 10*A*: (i) descriptive statistics for data presented in the figure panel, initial $pH_i$ values; (ii) overall one-way ANOVA results; (iii) Tukey's means comparison analysis**

i. Descriptive statistics

| | $n$ | Mean | SD | SEM |
|---|---|---|---|---|
| $H_2O$ + Tris | 6 | 7.29 | 0.0530 | 0.0216 |
| hAQP5 + Tris | 6 | 7.32 | 0.0741 | 0.0303 |
| $H_2O$ + 1 ng hCA II | 6 | 7.28 | 0.0685 | 0.0280 |
| hAQP5 + 1 ng hCA II | 6 | 7.31 | 0.0441 | 0.0180 |

ii. Overall ANOVA table

| Variation | DF | Sum of squares | Mean square | *F*-value | *P*-value |
|---|---|---|---|---|---|
| Between groups | 3 | 0.00556 | 0.00185 | 0.497 | 0.689 |
| Within groups | 20 | 0.0747 | 0.00373 | | |
| Total | 23 | 0.0802 | | | |

iii. Tukey's test

| Condition A | Condition B | Mean diff. | *Q*-value | *P*-value | Sig. | LCL | UCL |
|---|---|---|---|---|---|---|---|
| hAQP5 + Tris | $H_2O$ + Tris | 0.0352 | 1.41 | 0.752 | – | −0.0635 | 0.134 |
| $H_2O$ + 1 ng hCA II | $H_2O$ + Tris | −0.00157 | 0.0629 | 1.000 | – | −0.100 | 0.0972 |
| $H_2O$ + 1 ng hCA II | hAQP5 + Tris | −0.0368 | 1.47 | 0.727 | – | −0.136 | 0.0620 |
| hAQP5 + 1 ng hCA II | $H_2O$ + Tris | 0.0203 | 0.813 | 0.939 | – | −0.0785 | 0.119 |
| hAQP5 + 1 ng hCA II | hAQP5 + Tris | −0.0149 | 0.599 | 0.974 | – | −0.114 | 0.0838 |
| hAQP5 + 1 ng hCA II | $H_2O$ + 1 ng hCA II | 0.0218 | 0.876 | 0.925 | – | −0.0769 | 0.121 |

*Notes*: Threshold for significance a = 0.05. SEM = 0.0353.
Abbreviations: DF, degrees of freedom; LCL, lower confidence limit; mean diff., mean difference for each comparison; UCL, upper confidence limit.

**Table A10B. Statistics summary for Fig. 10B: (i) descriptive statistics for data presented in the figure, $\Delta pH_i$ values; (ii) overall one-way ANOVA results; (iii) Tukey's means comparison analysis**

i. Descriptive statistics

|  | n | Mean | SD | SEM |
|---|---|---|---|---|
| $H_2O$ + Tris | 6 | −0.275 | 0.0274 | 0.0112 |
| hAQP5 + Tris | 6 | −0.291 | 0.0117 | 0.00477 |
| $H_2O$ + 1 ng hCA II | 6 | −0.275 | 0.0374 | 0.0153 |
| hAQP5 + 1 ng hCA II | 6 | −0.301 | 0.0383 | 0.0157 |

ii. Overall ANOVA table

| Variation | DF | Sum of squares | Mean square | F-value | P-value |
|---|---|---|---|---|---|
| Between groups | 3 | 0.00297 | $9.90 \times 10^{-4}$ | 1.05 | 0.391 |
| Within groups | 20 | 0.0188 | $9.40 \times 10^{-4}$ | | |
| Total | 23 | 0.0218 | | | |

iii. Tukey's test

| Condition A | Condition B | Mean diff. | Q-value | P-value | Sig. | LCL | UCL |
|---|---|---|---|---|---|---|---|
| hAQP5 + Tris | $H_2O$ + Tris | −0.0161 | 1.29 | 0.798 | – | −0.0334 | 0.0657 |
| $H_2O$ + 1 ng hCA II | $H_2O$ + Tris | $6.38 \times 10^{-4}$ | 0.0510 | 1.000 | – | −0.0502 | 0.0489 |
| $H_2O$ + 1 ng hCA II | hAQP5 + Tris | 0.0168 | 1.34 | 0.779 | – | −0.0663 | 0.0327 |
| hAQP5 + 1 ng hCA II | $H_2O$ + Tris | −0.0256 | 2.05 | 0.486 | – | −0.0239 | 0.0752 |
| hAQP5 + 1 ng hCA II | hAQP5 + Tris | −0.00948 | 0.758 | 0.949 | – | −0.0401 | 0.0590 |
| hAQP5 + 1 ng hCA II | $H_2O$ + 1ng hCA II | −0.0263 | 2.10 | 0.465 | – | −0.0233 | 0.0758 |

*Notes*: Threshold for significance a = 0.05. SEM = 0.0177.
Abbreviations: DF, degrees of freedom; LCL, lower confidence limit; mean diff., mean difference for each comparison; UCL, upper confidence limit.

**Table A10C. Statistics summary for Fig. 10C: (i) descriptive statistics for data presented in the figure panel, intrinsic buffering power ($\beta_I$); (ii) overall one-way ANOVA results; (iii) Tukey's means comparison analysis**

i. Descriptive statistics

| | n | Mean | SD | SEM |
|---|---|---|---|---|
| $H_2O$ + Tris | 6 | 12.0 | 2.10 | 0.855 |
| hAQP5 + Tris | 6 | 11.7 | 1.63 | 0.664 |
| $H_2O$ + 1 ng hCA II | 6 | 12.0 | 1.87 | 0.762 |
| hAQP5 + 1 ng hCA II | 6 | 10.8 | 1.43 | 0.586 |

ii. Overall ANOVA table

| Variation | DF | Sum of squares | Mean square | F-value | P-value |
|---|---|---|---|---|---|
| Between groups | 3 | 5.77 | 1.92 | 0.612 | 0.615 |
| Within groups | 20 | 62.9 | 3.14 | | |
| Total | 23 | 68.6 | | | |

iii. Tukey's test

| Condition A | Condition B | Mean diff. | Q-value | P-value | Sig. | LCL | UCL |
|---|---|---|---|---|---|---|---|
| hAQP5 + Tris | $H_2O$ + Tris | −0.265 | 0.366 | 0.994 | − | −3.13 | 2.60 |
| $H_2O$ + 1 ng hCA II | $H_2O$ + Tris | −0.0111 | 0.0154 | 1.000 | − | −2.88 | 2.85 |
| $H_2O$ + 1 ng hCA II | hAQP5 + Tris | 0.253 | 0.350 | 0.994 | − | −2.61 | 3.12 |
| hAQP5 + 1 ng hCA II | $H_2O$ + Tris | −1.20 | 1.65 | 0.652 | − | −4.06 | 1.67 |
| hAQP5 + 1 ng hCA II | hAQP5 + Tris | −0.932 | 1.29 | 0.799 | − | −3.80 | 1.93 |
| hAQP5 + 1 ng hCA II | $H_2O$ + 1 ng hCA II | −1.19 | 1.64 | 0.659 | − | −4.05 | 1.68 |

*Notes*: Threshold for significance a = 0.05. SEM = 1.02.
Abbreviations: DF, degrees of freedom; LCL, lower confidence limit; mean diff., mean difference for each comparison; UCL, upper confidence limit.

**Table A10D. Statistics summary for Fig. 10*D*: (i) descriptive statistics for data presented in the figure panel, $P_f$; (ii) overall one-way ANOVA results; (iii) Tukey's means comparison analysis**

**i. Descriptive statistics**

|  | *n* | Mean | SD | SEM |
|---|---|---|---|---|
| $H_2O$ + Tris | 8 | $1.91 \times 10^{-4}$ | $2.19 \times 10^{-4}$ | $7.74 \times 10^{-5}$ |
| hAQP5 + Tris | 7 | $1.67 \times 10^{-4}$ | $1.18 \times 10^{-4}$ | $4.47 \times 10^{-5}$ |
| $H_2O$ + 1 ng hCA II | 7 | 0.00389 | $3.34 \times 10^{-4}$ | $1.26 \times 10^{-4}$ |
| hAQP5 + 1 ng hCA II | 8 | 0.00391 | $4.82 \times 10^{-4}$ | $1.70 \times 10^{-4}$ |

**ii. Overall ANOVA table**

| Variation | DF | Sum of squares | Mean square | *F*-value | *P*-value |
|---|---|---|---|---|---|
| Between groups | 3 | $1.04 \times 10^{-4}$ | $3.46 \times 10^{-5}$ | 331 | $7.92 \times 10^{-21}$ |
| Within groups | 26 | $2.72 \times 10^{-6}$ | $1.04 \times 10^{-7}$ |  |  |
| Total | 29 | $1.07 \times 10^{-4}$ |  |  |  |

**iii. Tukey's test**

| Condition A | Condition B | Mean diff. | SEM | *Q*-value | *P*-value | Sig. | LCL | UCL |
|---|---|---|---|---|---|---|---|---|
| $H_2O$ + 1 ng hCA II | $H_2O$ + Tris | $-2.44 \times 10^{-5}$ | $1.67 \times 10^{-4}$ | 0.206 | 0.999 | – | $-4.83 \times 10^{-4}$ | $4.34 \times 10^{-4}$ |
| hAQP5 + Tris | $H_2O$ + Tris | 0.00370 | $1.67 \times 10^{-4}$ | 31.3 | $2.10 \times 10^{-12}$ | ✓ | 0.00324 | 0.00416 |
| $H_2O$ + 1 ng hCA II | hAQP5 + Tris | 0.00372 | $1.73 \times 10^{-4}$ | 30.5 | $2.10 \times 10^{-12}$ | ✓ | 0.00325 | 0.00420 |
| hAQP5 + 1 ng hCA II | $H_2O$ + Tris | 0.00372 | $1.62 \times 10^{-4}$ | 32.6 | $2.10 \times 10^{-12}$ | ✓ | 0.00328 | 0.00416 |
| hAQP5 + 1 ng hCA II | $H_2O$ + 1 ng hCA II | 0.00375 | $1.67 \times 10^{-4}$ | 31.7 | $2.10 \times 10^{-12}$ | ✓ | 0.00329 | 0.00420 |
| hAQP5 + 1 ng hCA II | hAQP5 + Tris | $2.32 \times 10^{-5}$ | $1.67 \times 10^{-4}$ | 0.196 | 0.999 | – | $-4.36 \times 10^{-4}$ | $4.82 \times 10^{-4}$ |

*Note*: Threshold for significance a = 0.05. SEM for Tukey's test.

Abbreviations: DF, degrees of freedom; LCL, lower confidence limit; mean diff., mean difference for each comparison; UCL, upper confidence limit.

**Table A11A. Statistics summary for Fig. 11A: (i) descriptive statistics for data presented in the figure panel, which are the $\triangle pH_S$ upon $CO_2$ addition data rearranged from Fig. 3A to juxtapose two hAQP5 absent bars and two heterologously expressed hAQP5 bars; (ii) overall one-way ANOVA results for only data bars presented in Fig. 11A extracted from Fig. 3A; (iii) Tukey's means comparison analysis for all data presented in Fig. 11A; (iv) descriptive statistics for data presented in Fig. 11A to analyse $\triangle\triangle_{Base}$ vs. $\triangle\triangle_{AQP5}$ when adding 1 ng hCA II on the 'trans' side of the membrane to the $pH_S$ measurement; (v) one-way ANOVA results for $\triangle\triangle_{Base}$ vs. $\triangle\triangle_{AQP5}$ data presented in Fig. 11A; (vi) Tukey's means comparison analysis for $\triangle\triangle_{Base}$ vs. $\triangle\triangle_{AQP5}$ data presented in Fig. 11A**

### i. Descriptive statistics for $\triangle pH_S$ on $CO_2$ addition

|  | *n* | Mean | SD | SEM |
|---|---|---|---|---|
| $H_2O$ + Tris | 8 | 0.0285 | 0.00895 | 0.00316 |
| hAQP5 + Tris | 8 | 0.0787 | 0.0181 | 0.00641 |
| $H_2O$ + 1 ng hCA II | 8 | 0.0411 | 0.0115 | 0.00406 |
| hAQP5 + 1 ng hCA II | 8 | 0.127 | 0.0246 | 0.00869 |

### ii. Overall ANOVA table for $\triangle pH_S$ on $CO_2$ addition

| Variation | DF | Sum of squares | Mean square | *F*-value | *P*-value |
|---|---|---|---|---|---|
| Between groups | 3 | 0.0471 | 0.0157 | 54.8 | $7.63\times10^{-12}$ |
| Within groups | 28 | 0.00802 | $2.86\times10^{-4}$ |  |  |
| Total | 31 | 0.0552 |  |  |  |

### iii. Tukey's test for $\triangle pH_S$ on $CO_2$ addition

| Condition A | Condition B | Mean diff. | SEM | *Q*-value | *P*-value | Sig. | LCL | UCL |
|---|---|---|---|---|---|---|---|---|
| hAQP5 + Tris | $H_2O$ + Tris | 0.0502 | 0.00846 | 8.39 | $1.25\times10^{-5}$ | ✓ | 0.0271 | 0.0733 |
| $H_2O$ + 1 ng hCA II | $H_2O$ + Tris | 0.0126 | 0.00846 | 2.11 | 0.455 | − | −0.0105 | 0.0357 |
| $H_2O$ + 1 ng hCA II | hAQP5 + Tris | −0.0376 | 0.00846 | 6.28 | $7.00\times10^{-4}$ | ✓ | −0.0607 | −0.0145 |
| hAQP5 + 1 ng hCA II | $H_2O$ + Tris | 0.0986 | 0.00846 | 16.5 | 0.00 | ✓ | 0.0755 | 0.122 |
| hAQP5 + 1 ng hCA II | hAQP5 + Tris | 0.0484 | 0.00846 | 8.09 | $2.20\times10^{-5}$ | ✓ | 0.0253 | 0.0715 |
| hAQP5 + 1 ng hCA II | $H_2O$ + 1 ng hCA II | 0.0860 | 0.00846 | 14.4 | $6.50\times10^{-7}$ | ✓ | 0.0629 | 0.109 |

### iv. Descriptive statistics for $\triangle\triangle pH_S$ on $CO_2$ addition

|  | *n* | Mean | SD | SEM |
|---|---|---|---|---|
| $\triangle\triangle_{Base}$ | 8 | 0.0126 | 0.0115 | 0.00406 |
| $\triangle\triangle_{hAQP5}$ | 8 | 0.0484 | 0.0246 | 0.00869 |

### v. Overall ANOVA table for $\triangle\triangle pH_S$ on $CO_2$ addition

| Variation | DF | Sum of squares | Mean square | *F*-value | *P*-value |
|---|---|---|---|---|---|
| Between groups | 1 | 0.00513 | 0.00513 | 13.9 | 0.00223 |
| Within groups | 14 | 0.00516 | $3.68\times10^{-4}$ |  |  |
| Total | 15 | 0.0103 |  |  |  |

### vi. Tukey's test for $\triangle\triangle pH_S$ on $CO_2$ addition

| Condition A | Condition B | Mean diff. |  |  |  |  |  |
|---|---|---|---|---|---|---|---|
|  |  | SEM | *Q*-value | *P*-value | Sig. | LCL | UCL |
| $\triangle\triangle_{Base}$ | $\triangle\triangle_{AQP5}$ | 0.0358 | 0.00960 | 5.28 | 0.00224 | ✓ | 0.0152 | 0.0564 |

*Notes*: Threshold for significance a = 0.05. SEM for Tukey's test.
Abbreviations: DF, degrees of freedom; LCL, lower confidence limit; mean diff., mean difference for each comparison; UCL, upper confidence limit.

**Table A11B. Statistics summary for Fig. 11*B*: (i) descriptive statistics for data presented in Fig. 11*B*, which are the $\triangle pH_S$ upon $CO_2$ removal data rearranged from Fig. 3*B* to juxtapose two hhAQP5 absent bars and two heterologously expressed hAQP5 bars; (ii) overall one-way ANOVA results for only data bars presented in Fig. 11*B* extracted from Fig. 3*B*; (iii) Tukey's means comparison analysis for all data presented in Fig. 11*B*; (iv) descriptive statistics for data presented in Fig. 11*B* to analyse $\triangle\triangle_{Base}$ *vs.* $\triangle\triangle_{AQP5}$ when adding 1 ng hCA II on the 'trans' side of the membrane to the $pH_S$ measurement; (v) one-way ANOVA results for $\triangle\triangle_{Base}$ *vs.* $\triangle\triangle_{AQP5}$ data presented in Fig. 11*B*; (vi) Tukey's means comparison analysis for $\triangle\triangle_{Base}$ *vs.* $\triangle\triangle_{AQP5}$ data presented in Fig. 11*B***

### i. Descriptive statistics for $\triangle pH_S$ on $CO_2$ addition

|  | n | Mean | SD | SEM |
|---|---|---|---|---|
| $H_2O$ + Tris | 8 | 0.0285 | 0.00895 | 0.00316 |
| hAQP5 + Tris | 8 | 0.0787 | 0.0181 | 0.00641 |
| $H_2O$ + 100 ng hCA II | 8 | 0.0416 | 0.00948 | 0.00335 |
| hAQP5 + 100 ng hCA II | 8 | 0.160 | 0.0478 | 0.0169 |

### ii. *Overall ANOVA table for $\triangle pH_S$ on $CO_2$ addition*

| Variation | DF | Sum of squares | Mean square | *F*-value | *P*-value |
|---|---|---|---|---|---|
| Between groups | 3 | 0.0845 | 0.0282 | 40.4 | $2.61 \times 10^{-10}$ |
| Within groups | 28 | 0.0195 | $6.97 \times 10^{-4}$ |  |  |
| Total | 31 | 0.104 |  |  |  |

### iii. Tukey's test for $\triangle pH_S$ on $CO_2$ addition

| Condition A | Condition B | Mean diff. | SEM | *Q*-value | *P*-value | Sig. | LCL | UCL |
|---|---|---|---|---|---|---|---|---|
| hAQP5 + Tris | $H_2O$ + Tris | 0.0502 | 0.0132 | 5.38 | 0.00374 | ✓ | 0.0142 | 0.0862 |
| $H_2O$ + 100 ng hCA II | $H_2O$ + Tris | 0.0131 | 0.0132 | 1.41 | 0.754 | – | −0.0229 | 0.0492 |
| $H_2O$ + 100 ng hCA II | hAQP5 + Tris | −0.0371 | 0.0132 | 3.97 | 0.0419 | ✓ | −0.0731 | −0.00104 |
| hAQP5 + 100 ng hCA II | $H_2O$ + Tris | 0.132 | 0.0132 | 14.1 | $2.23 \times 10^{-7}$ | ✓ | 0.0959 | 0.168 |
| hAQP5 + 100 ng hCA II | hAQP5 + Tris | 0.0817 | 0.0132 | 8.76 | $6.27 \times 10^{-6}$ | ✓ | 0.0457 | 0.118 |
| hAQP5 + 100 ng hCA II | $H_2O$ + 100 ng hCA II | 0.119 | 0.0132 | 12.7 | 0.00 | ✓ | 0.0828 | 0.155 |

### iv. Descriptive statistics for $\triangle\triangle pH_S$ on $CO_2$ addition

|  | *n* | Mean | SD | SEM |
|---|---|---|---|---|
| $\triangle\triangle_{Base}$ | 8 | 0.0131 | 0.00948 | 0.00335 |
| $\triangle\triangle_{hAQP5}$ | 8 | 0.0817 | 0.0478 | 0.0169 |

### v. Overall ANOVA table

| Variation | DF | Sum of squares | Mean square | *F*-value | *P*-value |
|---|---|---|---|---|---|
| Between groups | 1 | 0.0188 | 0.0188 | 15.8 | 0.00137 |
| Within groups | 14 | 0.0166 | 0.00119 |  |  |
| Total | 15 | 0.0355 |  |  |  |

### vi. Tukey's test

| Condition A | Condition B | Mean diff. | SEM | *Q*-value | *P*-value | Sig. | LCL | UCL |
|---|---|---|---|---|---|---|---|---|
| $\triangle\triangle_{Base}$ | $\triangle\triangle_{AQP5}$ | 0.0686 | 0.0172 | 5.63 | 0.00137 | ✓ | 0.0316 | 0.106 |

*Notes*: Threshold for significance a = 0.05. SEM for Tukey's test.
Abbreviations: DF, degrees of freedom; LCL, lower confidence limit; mean diff., mean difference for each comparison; UCL, upper confidence limit.

**Table A12A. Statistics summary for Fig. 12A:** (i) descriptive statistics for data presented in Fig. 12A for comparison of select $-(dpH_i/dt)_{Max}$ bars extracted from Fig. 3C and Fig. 8C to analyse $-(dpH_i/dt)_{Max}$ upon $CO_2$ addition in the absence or presence of bCA in bECF on the 'trans' side of the membrane to the $pH_i$ measurement; (ii) overall one-way ANOVA results for $-(dpH_i/dt)_{Max}$ data presented in Fig. 12A; (iii) Tukey's means comparison analysis for $-(dpH_i/dt)_{Max}$ was performed on the subset of data presented in Fig. 12A; (iv) descriptive statistics for data presented in Fig. 12A to analyse $\Delta_{Base}$ *vs.* $\Delta_{AQP5}$; (v) one-way ANOVA results for $\Delta_{Base}$ *vs.* $\Delta_{AQP5}$ data presented in Fig. 12A; (vi) Tukey's means comparison analysis for $\Delta\Delta_{Base}$ *vs.* $\Delta\Delta_{AQP5}$ data presented in Fig. 12A

**i. Descriptive statistics for $-(dpH_i/dt)_{Max}$ on $CO_2$ addition**

|  | *n* | Mean | SD | SEM |
|---|---|---|---|---|
| $H_2O$ + Tris | 8 | −0.00126 | $2.08\times10^{-4}$ | $7.35\times10^{-5}$ |
| $H_2O$ + Tris + $bCA_{bECF}$ | 6 | −0.00250 | $7.09\times10^{-4}$ | $2.89\times10^{-4}$ |
| hAQP5 + Tris | 8 | −0.00122 | $1.57\times10^{-4}$ | $5.55\times10^{-5}$ |
| hAQP5 + Tris+ $bCA_{bECF}$ | 6 | −0.00451 | 0.00110 | $4.48\times10^{-4}$ |

**ii. Overall ANOVA table for $-(dpH_i/dt)_{Max}$ on $CO_2$ addition**

| Variation | DF | Sum of squares | Mean square | *F*-value | *P*-value |
|---|---|---|---|---|---|
| Between groups | 3 | $4.73\times10^{-5}$ | $1.58\times10^{-5}$ | 42.0 | $1.04\times10^{-9}$ |
| Within groups | 24 | $9.00\times10^{-6}$ | $3.75\times10^{-7}$ |  |  |
| Total | 27 | $5.63\times10^{-5}$ |  |  |  |

**iii. Tukey's test for $-(dpH_i/dt)_{Max}$ on $CO_2$ addition**

| Condition A | Condition B | Mean diff. | SEM | *Q*-value | *P*-value | Sig. | LCL | UCL |
|---|---|---|---|---|---|---|---|---|
| $H_2O$ + Tris + $bCA_{bECF}$ | $H_2O$ + Tris | −0.00124 | $3.31\times10^{-4}$ | 5.30 | 0.00513 | ✓ | −0.00215 | $-3.27\times10^{-4}$ |
| hAQP5 + Tris | $H_2O$ + Tris | $3.53\times10^{-5}$ | $3.06\times10^{-4}$ | 0.163 | 0.999 | − | $-8.09\times10^{-4}$ | $8.80\times10^{-4}$ |
| hAQP5 + Tris | $H_2O$ + Tris + $bCA_{bECF}$ | 0.00127 | $3.31\times10^{-4}$ | 5.45 | 0.00396 | ✓ | $3.62\times10^{-4}$ | 0.00219 |
| hAQP5 + Tris+ $bCA_{bECF}$ | $H_2O$ + Tris | −0.00325 | $3.31\times10^{-4}$ | 13.9 | $4.43\times10^{-7}$ | ✓ | −0.00417 | −0.00234 |
| hAQP5 + Tris+ $bCA_{bECF}$ | $H_2O$ + Tris + $bCA_{bECF}$ | −0.00201 | $3.54\times10^{-4}$ | 8.05 | $4.07\times10^{-5}$ | ✓ | −0.00299 | −0.00104 |
| hAQP5 + Tris+ $bCA_{bECF}$ | hAQP5 + Tris | −0.00329 | $3.31\times10^{-4}$ | 14.1 | $4.47\times10^{-7}$ | ✓ | −0.00420 | −0.00238 |

**iv. Descriptive statistics for $\Delta_{hAQP5}$ *vs.* $\Delta_{Base}$ for $-(dpH_i/dt)_{Max}$ on $CO_2$ addition**

|  | *n* | Mean | SD | SEM |
|---|---|---|---|---|
| $\Delta_{Base}$ | 6 | −0.00271 | $7.09\times10^{-4}$ | $2.89\times10^{-4}$ |
| $\Delta_{AQP5}$ | 6 | −0.00467 | 0.00110 | $4.48\times10^{-4}$ |

**v. Overall ANOVA table**

| Variation | DF | Sum of squares | Mean square | *F*-value | *P*-value |
|---|---|---|---|---|---|
| Between groups | 1 | $1.16\times10^{-5}$ | $1.16\times10^{-5}$ | 13.6 | 0.00424 |
| Within groups | 10 | $8.53\times10^{-6}$ | $8.53\times10^{-7}$ |  |  |
| Total | 11 | $2.01\times10^{-5}$ |  |  |  |

**vi. Tukey's test**

| Condition A | Condition B | Mean diff. | SEM | *Q*-value | *P*-value | Sig. | LCL | UCL |
|---|---|---|---|---|---|---|---|---|
| $\Delta_{Base}$ | $\Delta_{AQP5}$ | −0.00196 | $5.33\times10^{-4}$ | 5.21 | 0.00424 | ✓ | −0.00315 | $-7.75\times10^{-4}$ |

*Note*: Threshold for significance a = 0.05. SEM for Tukey's test.
Abbreviations: DF, degrees of freedom; LCL, lower confidence limit; mean diff., mean difference for each comparison; UCL, upper confidence limit.

**Table A12B. Statistics summary for Fig. 12*B*: (i)** descriptive statistics for data presented in Fig. 12*B*, for comparison of select $-(dpH_i/dt)_{Max}$ bars extracted from Fig. 3*D* and Fig. 8*D* to analyse $+(dpH_i/dt)_{Max}$ upon $CO_2$ removal in the absence or presence of bCA in bECF on the 'trans' side of the membrane to the $pH_i$ measurement; **(ii)** overall one-way ANOVA results for $+(dpH_i/dt)_{Max}$ data presented in Fig. 12*B*; **(iii)** Tukey's means comparison analysis for $+(dpH_i/dt)_{Max}$ data presented in Fig. 12*B*; **(iv)** descriptive statistics for data presented in Fig. 12*B* to analyse $\Delta_{Base}$ *vs.* $\Delta_{AQP5}$; **(v)** one-way ANOVA results for $\Delta_{Base}$ *vs.* $\Delta_{AQP5}$ data presented in Fig. 12*B*; **(vi)** Tukey's means comparison analysis for $\Delta\Delta_{Base}$ *vs.* $\Delta\Delta_{AQP5}$ data presented in Fig. 12*B*

**i. Descriptive statistics for $+(dpH_i/dt)_{Max}$ on $CO_2$ removal**

|  | *n* | Mean | SD | SEM |
|---|---|---|---|---|
| $H_2O$ + Tris | 8 | $8.35 \times 10^{-4}$ | $1.34 \times 10^{-4}$ | $4.74 \times 10^{-5}$ |
| $H_2O$ + Tris + $bCA_{bECF}$ | 6 | 0.00161 | $6.45 \times 10^{-4}$ | $2.63 \times 10^{-4}$ |
| hAQP5 + Tris | 8 | $8.07 \times 10^{-4}$ | $6.85 \times 10^{-5}$ | $2.42 \times 10^{-5}$ |
| hAQP5 + Tris+ $bCA_{bECF}$ | 6 | 0.00229 | $5.65 \times 10^{-4}$ | $2.31 \times 10^{-4}$ |

**ii. Overall ANOVA table for $+(dpH_i/dt)_{Max}$ on $CO_2$ removal**

| Variation | DF | Sum of squares | Mean square | *F*-value | *P*-value |
|---|---|---|---|---|---|
| Between groups | 3 | $1.01 \times 10^{-5}$ | $3.35 \times 10^{-6}$ | 21.0 | $6.82 \times 10^{-7}$ |
| Within groups | 24 | $3.84 \times 10^{-6}$ | $1.60 \times 10^{-7}$ |  |  |
| Total | 27 | $1.39 \times 10^{-5}$ |  |  |  |

**iii. Tukey's test for $+(dpH_i/dt)_{Max}$ on $CO_2$ removal**

| Condition A | Condition B | Mean diff. | SEM | *Q*-value | *P*-value | Sig. | LCL | UCL |
|---|---|---|---|---|---|---|---|---|
| $H_2O$ + Tris + $bCA_{bECF}$ | $H_2O$ + Tris | $7.72 \times 10^{-4}$ | $2.16 \times 10^{-4}$ | 5.06 | 0.00778 | ✓ | $1.76 \times 10^{-4}$ | 0.00137 |
| hAQP5 + Tris | $H_2O$ + Tris | $-2.75 \times 10^{-5}$ | $2.00 \times 10^{-4}$ | 0.195 | 0.999 | – | $-5.79 \times 10^{-4}$ | $5.24 \times 10^{-4}$ |
| hAQP5 + Tris | $H_2O$ + Tris + $bCA_{bECF}$ | $-7.99 \times 10^{-4}$ | $2.16 \times 10^{-4}$ | 5.24 | 0.00572 | ✓ | $-0.00140$ | $-2.04 \times 10^{-4}$ |
| hAQP5 + Tris+ $bCA_{bECF}$ | $H_2O$ + Tris | 0.00145 | $2.16 \times 10^{-4}$ | 9.50 | $3.43 \times 10^{-6}$ | ✓ | $8.55 \times 10^{-4}$ | 0.00205 |
| hAQP5 + Tris+ $bCA_{bECF}$ | $H_2O$ + Tris + $bCA_{bECF}$ | $6.78 \times 10^{-4}$ | $2.31 \times 10^{-4}$ | 4.16 | 0.0338 | ✓ | $4.15 \times 10^{-5}$ | 0.00131 |
| hAQP5 + Tris+ $bCA_{bECF}$ | hAQP5 + Tris | 0.00148 | $2.16 \times 10^{-4}$ | 9.68 | $2.54 \times 10^{-6}$ | ✓ | $8.82 \times 10^{-4}$ | 0.00207 |

**iv. Descriptive statistics for $\Delta_{AQP5}$ *vs.* $\Delta_{Base}$ for $-(dpH_i/dt)_{Max}$ on $CO_2$ removal**

|  | *n* | Mean | SD | SEM |
|---|---|---|---|---|
| $\Delta_{Base}$ | 6 | $7.72 \times 10^{-4}$ | $6.45 \times 10^{-4}$ | $2.63 \times 10^{-4}$ |
| $\Delta_{AQP5}$ | 6 | 0.00148 | $5.65 \times 10^{-4}$ | $2.31 \times 10^{-4}$ |

**v. Overall ANOVA table**

| Variation | DF | Sum of squares | Mean square | *F*-value | *P*-value |
|---|---|---|---|---|---|
| Between groups | 1 | $1.49 \times 10^{-6}$ | $1.49 \times 10^{-6}$ | 4.06 | 0.0715 |
| Within groups | 10 | $3.68 \times 10^{-6}$ | $3.68 \times 10^{-7}$ |  |  |
| Total | 11 | $5.17 \times 10^{-6}$ |  |  |  |

**vi. Tukey's test**

| Condition A | Condition B | Mean diff. | SEM | *Q*-value | *P*-value | Sig. | LCL | UCL |
|---|---|---|---|---|---|---|---|---|
| $\Delta_{Base}$ | $\Delta_{AQP5}$ | $7.06 \times 10^{-4}$ | $3.50 \times 10^{-4}$ | 2.85 | 0.0715 | ✓ | $-7.43 \times 10^{-5}$ | 0.00149 |

*Notes*: Threshold for significance a = 0.05. SEM for Tukey's test.
Abbreviations: DF, degrees of freedom; LCL, lower confidence limit; mean diff., mean difference for each comparison; UCL, upper confidence limit.

**Table A13. Statistics summary for Fig. *A*1: (i) descriptive statistics for data presented in Fig. *A*1 for summarizing $V_m$ values for oocytes used in Fig. 3; (ii) overall one-way ANOVA results for $V_m$ data presented in Fig. *A*1; (iii) Tukey's means comparison analysis for $V_m$ data presented in Fig. *A*1**

### i. Descriptive statistics

|  | *n* | Mean | SD | SEM |
|---|---|---|---|---|
| $H_2O$ + Tris | 8 | −57.8 | 6.77 | 2.39 |
| hAQP5 + Tris | 8 | −45.5 | 2.90 | 1.02 |
| hAQP5 + 1 ng hCA II | 8 | −51.8 | 8.44 | 2.98 |
| $H_2O$ + 1 ng hCA II | 8 | −43.0 | 6.55 | 2.32 |
| $H_2O$ + 10 ng hCA II | 8 | −55.1 | 7.14 | 2.52 |
| hAQP5 + 10 ng hCA II | 8 | −44.9 | 7.05 | 2.49 |
| $H_2O$ + 100 ng hCA II | 8 | −57.6 | 6.25 | 2.21 |
| hAQP5 + 100 ng hCA II | 8 | −46.8 | 4.39 | 1.55 |

### ii. Overall ANOVA table

| Variation | DF | Sum of squares | Mean square | *F*-value | *P*-value |
|---|---|---|---|---|---|
| Between groups | 7.00 | $2.01 \times 10^3$ | 287 | 7.01 | $5.30 \times 10^{-6}$ |
| Within groups | 56.0 | $2.29 \times 10^3$ | 40.9 |  |  |
| Total | 63.0 | $4.30 \times 10^3$ |  |  |  |

### iii. Tukey's test

| Condition A | Condition B | Mean diff. | *Q*-value | *P*-value | Sig. | LCL | UCL |
|---|---|---|---|---|---|---|---|
| hAQP5 + Tris | $H_2O$ + Tris | 12.2 | 5.42 | 0.00738 | √ | 2.18 | 22.3 |
| $H_2O$ + 1 ng hCA II | $H_2O$ + Tris | 5.96 | 2.63 | 0.581 | – | −4.11 | 16.0 |
| $H_2O$ + 1 ng hCA II | hAQP5 + Tris | −6.29 | 2.78 | 0.513 | – | −16.4 | 3.78 |
| hAQP5 + 1 ng hCA II | $H_2O$ + Tris | 14.7 | 6.51 | $6.06 \times 10^{-4}$ | √ | 4.66 | 24.8 |
| hAQP5 + 1 ng hCA II | hAQP5 + Tris | 2.48 | 1.10 | 0.994 | – | −7.59 | 12.6 |
| hAQP5 + 1 ng hCA II | $H_2O$ + 1ng hCA II | 8.77 | 3.88 | 0.132 | – | −1.30 | 18.8 |
| $H_2O$ + 10 ng hCA II | $H_2O$ + Tris | 2.65 | 1.17 | 0.991 | – | −7.43 | 12.7 |
| $H_2O$ + 10 ng hCA II | hAQP5 + Tris | −9.60 | 4.25 | 0.0720 | – | −19.7 | 0.467 |
| $H_2O$ + 10 ng hCA II | $H_2O$ + 1ng hCA II | −3.31 | 1.46 | 0.967 | – | −13.4 | 6.76 |
| $H_2O$ + 10 ng hCA II | hAQP5 + 1ng hCA II | −12.1 | 5.34 | 0.00863 | √ | −22.2 | −2.01 |
| hAQP5 + 10 ng hCA II | $H_2O$ + Tris | 12.9 | 5.70 | 0.00396 | √ | 2.82 | 23.0 |
| hAQP5 + 10 ng hCA II | hAQP5 + Tris | 0.646 | 0.286 | 1.000 | – | −9.43 | 10.7 |
| hAQP5 + 10 ng hCA II | $H_2O$ + 1 ng hCA II | 6.94 | 3.07 | 0.386 | – | −3.13 | 17.0 |
| hAQP5 + 10 ng hCA II | hAQP5 + 1 ng hCA II | −1.83 | 0.810 | 0.999 | – | −11.9 | 8.24 |
| hAQP5 + 10 ng hCA II | $H_2O$ + 10 ng hCA II | 10.3 | 4.53 | 0.0433 | √ | 0.179 | 20.3 |
| $H_2O$ + 100 ng hCA II | $H_2O$ + Tris | 0.208 | 0.0921 | 1.00 | – | −9.86 | 10.3 |
| $H_2O$ + 100 ng hCA II | hAQP5 + Tris | −12.0 | 5.32 | 0.00897 | √ | −22.1 | −1.97 |
| $H_2O$ + 100 ng hCA II | $H_2O$ + 1 ng hCA II | −5.75 | 2.54 | 0.624 | – | −15.8 | 4.32 |
| $H_2O$ + 100 ng hCA II | hAQP5 + 1 ng hCA II | −14.5 | 6.42 | $7.56 \times 10^{-4}$ | √ | −24.6 | −4.45 |
| $H_2O$ + 100 ng hCA II | $H_2O$ + 10 ng hCA II | −2.44 | 1.08 | 0.994 | – | −12.5 | 7.63 |
| $H_2O$ + 100 ng hCA II | hAQP5 + 10 ng hCA II | −12.7 | 5.61 | 0.00485 | √ | −22.8 | −2.62 |
| hAQP5 +100 ng hCA II | $H_2O$ + Tris | 10.9 | 4.84 | 0.0241 | √ | 0.877 | 21.0 |
| hAQP5 +100 ng hCA II | hAQP5 + Tris | −1.30 | 0.576 | 1.000 | – | −11.4 | 8.77 |
| hAQP5 +100 ng hCA II | $H_2O$ + 1 ng hCA II | 4.99 | 2.21 | 0.771 | – | −5.08 | 15.1 |
| hAQP5 +100 ng hCA II | hAQP5 + 1 ng hCA II | −3.78 | 1.67 | 0.934 | – | −13.9 | 6.29 |
| hAQP5 +100 ng hCA II | $H_2O$ + 10 ng hCA II | 8.30 | 3.67 | 0.179 | – | −1.77 | 18.4 |
| hAQP5 +100 ng hCA II | hAQP5 + 10 ng hCA II | −1.95 | 0.861 | 0.999 | – | −12.0 | 8.12 |
| hAQP5 +100 ng hCA II | $H_2O$ + 100 ng hCA II | 10.7 | 4.75 | 0.0288 | √ | 0.668 | 20.8 |

*Notes*: Threshold for significance a = 0.05. SEM for Tukey's test = 3.20.
Abbreviations: DF, degrees of freedom; LCL, lower confidence limit; mean diff., mean difference for each comparison; UCL, upper confidence limit.

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

## Additional information

### Data availability statement

All raw data are deposited into the NIH-supported Zenodo data repository in common and open formats and are accessible via the persistent identifier; https://doi.org/10.5281/zenodo.16950950

### Competing interests

All authors declare no conflict of interests.

### Author contributions

D.K.W. contributed to the conception and design of the research, performed the experiments, analysed the data and interpreted the results, wrote the first draft of the manuscript, prepared the figures and edited the manuscript. F.J.M. contributed to the conception and design of the research, interpreted the results, performed statistical analyses and edited the figures and manuscript. W.F.B. contributed to the conception and design of the research, interpreted the results and edited the figures and manuscript. All authors approved the final version of the

manuscript; all qualify for authorship, and all those who qualify for authorship are listed.

## Funding

This work was supported by NIH grants HL160857 and DK128315; by Office of Naval Research (ONR) grant N00014-11-1-0889, N00014-14-1-0716 and N00014-15-1-2060; and by a Multidisciplinary University Research Initiative (MURI) grant N00014-16-1-2535 from the Department of Defense (to W.F.B.). The contributions of W.F.B. and F.J.M. to this work were also supported in part by a Department of Defense, Air Force Research Laboratory 711th Human Performance Wing, Studies and Analysis funding 21–023.

## Acknowledgements

We thank Dale E. Huffman for computer support. We acknowledge the assistance of Gerald T. Babcock in his role as a laboratory manager. W.F.B. gratefully acknowledges the support of the Myers/Scarpa endowed chair.

## Keywords

cell-surface pH, Fick's law, gas channels, intracellular pH, pH-sensitive microelectrodes

## Supporting information

Additional supporting information can be found online in the Supporting Information section at the end of the HTML view of the article. Supporting information files available:

**Peer Review History**

