## [Peer Review History · The Journal of Physiology]

Synergistic roles of Aquaporin 5 and Intra- and Extracellular Carbonic Anhydrases in promoting CO₂ Diffusion across the *Xenopus* Oocyte Plasma Membrane

Deng-Ke Wang, Fraser J Moss, and Walter F Boron
DOI: 10.1113/JP289145

Corresponding author(s): Walter Boron (walter.boron@case.edu)

Review Timeline:

Submission Date:	23-Apr-2025
Editorial Decision:	04-Jul-2025
Revision Received:	28-Aug-2025
Accepted:	11-Sep-2025

Senior Editor: Peking Fong

Reviewing Editor: Peking Fong

Transaction Report:

Dear Dr Boron,

Re: JP-RP-2025-289145 "Synergistic roles of Aquaporin 5 and Intra- and Extracellular Carbonic Anhydrases in promoting CO₂ Diffusion across the *Xenopus* Oocyte Plasma Membrane" by Deng-Ke Wang, Fraser J Moss, and Walter F Boron

Thank you for submitting your manuscript to The Journal of Physiology. It has been assessed by a Reviewing Editor and by 2 expert referees and we are pleased to tell you that it is acceptable for publication following satisfactory revision.

LANGUAGE EDITING AND SUPPORT FOR PUBLICATION: If you would like help with English language editing, or other article preparation support, Wiley Editing Services offers expert help, including English Language Editing, as well as translation, manuscript formatting, and figure formatting at www.wileyauthors.com/eoo/preparation. You can also find resources for Preparing Your Article for general guidance about writing and preparing your manuscript at www.wileyauthors.com/eoo/prepresources.

REVISION CHECKLIST:

We look forward to receiving your revised submission.

Yours sincerely,

Peying Fong
Senior Editor
The Journal of Physiology

REQUIRED ITEMS

- Author photo and profile. First or joint first authors are asked to provide a short biography (no more than 100 words for one author or 150 words in total for joint first authors) and a portrait photograph. These should be uploaded and clearly labelled together in a Word document with the revised version of the manuscript. See Information for Authors for further details.

- You must start the Methods section with a paragraph headed Ethical approval (https://jp.msubmit.net/cgi-bin/main.plex?form_type=display_requirements#methods).

Research must comply with The Journal's policies regarding animal experiments (<https://physoc.onlinelibrary.wiley.com/hub/animal-experiments>) and adherence to these policies must be stated in the manuscript.

Authors should confirm in their Methods section that their experiments were carried out according to the guidelines laid down by their institution's animal welfare committee, including an ethics approval reference number. The Methods section must contain a statement about access to food, water and housing, details of the anaesthetic regime: anaesthetic used, dose and route of administration, and method of killing the experimental animals.

- The Journal of Physiology funds authors of provisionally accepted papers to use the premium BioRender site to create high resolution schematic figures. Follow this link and enter your details and the manuscript number to create and download figures. Upload these as the figure files for your revised submission. If you choose not to take up this offer, we require figures to be of similar quality and resolution. If you are opting out of this service to authors, state this in the Comments section on the Detailed Information page of the submission form. The link provided should only be used for the purposes of this submission. Authors will be charged for figures created on this premium BioRender account if they are not related to this manuscript submission.

- Please ensure that any tables are editable and in Word format, and wherever possible, embedded in the article file itself.

- Please ensure that the Article File you upload is a Word file.

- A Data Availability Statement is required for all papers reporting original data. This must be in the Additional Information section of the manuscript itself. It must have the paragraph heading 'Data Availability Statement'. All data supporting the results in the paper must be either: in the paper itself; uploaded as Supporting Information for Online Publication; or archived in an appropriate public repository. The statement needs to describe the availability or the absence of shared data. Authors must include in their statement: a link to the repository they have used, or a statement that it is available as Supporting Information; reference the data in the appropriate sections(s) of their manuscript; and cite the data they have shared in the References section. Whenever possible, the scripts and other artefacts used to generate the analyses presented in the paper should also be publicly archived. If sharing data compromises ethical standards or legal requirements then authors are not expected to share it, but must note this in their statement. For more information, see our Statistics Policy.

- Please include an Abstract Figure file, as well as the Figure Legend text within the main article file. The Abstract Figure is a piece of artwork designed to give readers an immediate understanding of the research and should summarise the main conclusions. If possible, the image should be easily 'readable' from left to right or top to bottom. It should show the physiological relevance of the manuscript so readers can assess the importance and content of its findings. Abstract Figures should not merely recapitulate other figures in the manuscript. Please try to keep the diagram as simple as possible and without superfluous information that may distract from the main conclusion(s). Abstract Figures must be provided by authors no later than the revised manuscript stage and should be uploaded as a separate file during online submission labelled as File Type 'Abstract Figure'. Please also ensure that you include the figure legend in the main article file. All Abstract Figures should be created using BioRender. Authors should use The Journal's premium BioRender account to export high-resolution images. Details on how to use and access the premium account are included as part of this email.

EDITOR COMMENTS

Reviewing Editor:

Two experts in the field have reviewed the manuscript. They each agree that the manuscript has the potential to have a significant impact on the field.

Please ensure the revised manuscript complies with statistical policy https://jp.msubmit.net/cgi-bin/main.plex?form_type=display_requirements#statistics, including reporting of P-values in figures.

Senior Editor:

Your manuscript, "Synergistic roles of Aquaporin 5 and Intra- and Extracellular Carbonic Anhydrases in promoting CO₂ Diffusion across the Xenopus Oocyte Plasma Membrane" was assessed by two Expert Referees, whose feedback was summarized by the Reviewing Editor. Both Referees expressed enthusiasm about the manuscript's potential to influence the field significantly. Please note that their comments, while uniformly positive, also offered several points for your consideration. Many, if not most, focus on matters of clarity. Referee 2 does highlight points requiring deeper exploration and contemplation, particularly the comments pertaining to figures 5 and 7.

I encourage you to address all comments completely in your revised manuscript.

The extremely thorough statistical analyses performed in your study is commendable. Of note, the resultant information appears not only as tables within the manuscript proper but also are enhanced by supplemental statistical tables. In considering that these supplemental tables are not typical of what the Journal includes as supporting information files for online publication (video and audio files, 3D structures and program codes), it seems that perhaps they might better be incorporated within an Appendix, which can be included within the body of the revised manuscript proper.

For additional information, please refer to:

https://jp.msubmit.net/cgi-bin/main.plex?form_type=display_requirements#suppinfo.

Thank you for contributing your work to The Journal of Physiology. We look forward to receiving your revised manuscript.

REFEREE COMMENTS

Referee #1:

This is an outstanding, well-written and comprehensive exploration of the synergistic roles of an aquaporin and intra and extracellular CA activity on the transmembrane diffusion of CO₂ across oocytes. The authors have explored the permutations of adding in the CO₂ channel AQ5, extracellular CA activity and intracellular CA activity using measurements of changes in magnitude and speed in extracellular pH at the membrane and intracellular pH and deduce changes in CO₂ flux rates. They find that all three proteins increase CO₂ flux, but that there is a synergistic effect with the combination of all three. The work and data add nicely to the story of how CO₂ flux across membranes is dependent on the presence of a gas channel protein and both intra and extracellular CA activity. The work is original, the methods are state of the art and the data are novel and

compelling. The conclusions are well-supported by the data.

I have only minor concerns to raise.

1. Does the addition of small amounts of CA on either side of the membrane influence buffering capacity in the respective compartments to any significant extent?
2. How are the oocytes and electrodes maintained in a secure way when bathing conditions are changed?
3. Are AQP5 mediated H₂O fluxes also important to the augmentation of CO₂ fluxes?
4. page 4. I would add also the evidence that some aquaporins are NO gas channels
5. page 37 line 12, please add 'not yet peer reviewed' after 'results, since all three papers are Biorxiv postings. This applies elsewhere any time a Biorxiv post is cited.
6. page 41, line 23. should the word 'or' be 'on'?

Referee #2:

The study on the "Synergistic roles of Aquaporin 5 and Intra- and Extracellular Carbonic Anhydrases in promoting CO₂ Diffusion across the Xenopus Oocyte Plasma Membrane" is a detailed study that analyzes the interaction of catalysts (intracellular and extracellular CA) with the diffusion of CO₂ across cellular membrane in the absence and presence of membrane channels (AQP5 in this case) that also contribute to CO₂ transport. The study is meticulously detailed and executed, and the manuscript is very well written. The study builds up on previous studies by the same group and extends it to include intracellular and extracellular CA and AQP5 in relation to CO₂ transport. The multiple measurements in multiple conditions, although accurately informative, makes it difficult to read. The PI's group is one of few labs that can address such a study with experiments that are demanding and difficult both to do and interpret.

Although most of my concerns are not major, there are several points that need to be addressed as outlined below.

Abstract: the second half of the abstract needs to be rewritten. It is confusing because several terms and interactions are not yet described.

In Methods (P12, 2nd paragraph): The calibration points in ND96 & 1.5% CO₂ seem to be changing upon switching solutions. Is this why a new calibration point is needed every time? If so, what is the difference in measured pHs? Also, is there a difference between pHs in bulk solution & pHs when electrode is touching the membrane? Is there a difference in pHs (in bulk solution) between ND96 (at pH 7.5) and 1.5% CO₂?

P. 14: Shouldn't the equation read $10^{\text{pHi}-6.8}$ and not $10^{\text{pHi}-7.5}$?

Fig 1. Differences between "arrowheads" are difficult to recognize.

Fig 2: what is the membrane potential of oocytes in each condition (injected or expressing CA and/or AQP5)? This is one parameter that is indicative of the health status of the oocyte.

Fig 3 legend: The explanation of the star/bar significance is completely incomprehensible. Please rewrite in a better format easy to understand.

P. 23 last 2 lines: Remove "(vs 0ng) until 10 & 100ng"

P 25-26, Exemplar data & Fig 4: not needed.

P. 25: Use of $(dpH/dt)_{max}$ is not necessarily better than $1/\tau$ as done by Musa-Aziz et al., and Occhipinti et al. If I understand correctly, the conclusion is that CO₂ influx (or efflux) is increased with expression of CAII and remains so during re-equilibration at the surface. This can be determined by a measure of $1/\tau$ or by $(dpH/dt)_{max}$. However, my concern is that exponential decay could vary depending on the parameters mentioned by the author (tip of electrode, location on the membrane etc). This shows a potential source of error.

Fig 5: Fig 5A shows that $(dpH/dt)_{max}$ increases with increased CAII. However, in Fig. 5B $(dpH/dt)_{max}$ increased significantly only with 100 CAII (after the initial increase compared to CAII). Is it expected that the changes in $(dpH/dt)_{max}$ should mimic Fig 5A?

Fig 6: Same as before, the star/bar significance is very difficult to understand.

Fig 7: In Fig 7A, there is significant decrease in pHs when the electrode touches the membrane (gray bar). Any explanation for this?

In Figs 7A & 7B, the authors assert that no significant change in pHs because extracellular CO₂ is rate limiting. This means that if $P_m \cdot CO_2$ is increased due to AQP5, it will not affect the rate of CO₂ formation at the surface. Do the authors expect that at a lesser content of external CO₂ (e.g. 0.5% CO₂ rather than 1.5% CO₂), AQP5 expression may cause detectable change in pHs?

Fig 11 and text: using pHs for comparisons is a very good tool but the need to use double exponential for fitting is an overkill and probably does not make a difference.

The dependence of the measured changes on different levels of CO₂, in all conditions is very important.

Discussion:

P. 37, paragraph before last: informative but not needed. What is "mobile CA"?

What is expected by varying external CO₂ at constant pH. Please comment.

What is the logic behind including hCAIV with extracellular bCAII?

END OF COMMENTS

Dear Dr Boron,

Re: JP-RP-2025-289145 "Synergistic roles of Aquaporin 5 and Intra- and Extracellular Carbonic Anhydrases in promoting CO₂ Diffusion across the *Xenopus* Oocyte Plasma Membrane" by Deng-Ke Wang, Fraser J Moss, and Walter F Boron

Thank you for submitting your manuscript to The Journal of Physiology. It has been assessed by a Reviewing Editor and by 2 expert referees and we are pleased to tell you that it is acceptable for publication following satisfactory revision.

Your revised manuscript should be submitted online using the link in your Author Tasks <https://jp.msubmit.net/cgi-bin/main.plex?el=A1JS5HOM2A7OuW1F1A9ftdeBMNFMzLLqQE2RO3BxUogZ>. This link is accessible via your account as Corresponding Author; it is not available to your co-authors. If this presents a problem, please contact journal staff (jp@physoc.org). Image files from the previous version are retained on the system. Please ensure you replace or remove any files that are being revised.

LANGUAGE EDITING AND SUPPORT FOR PUBLICATION: If you would like help with English language editing, or other article preparation support, Wiley Editing Services offers expert help, including English Language Editing, as well as translation, manuscript formatting, and figure formatting at www.wileyauthors.com/eo/preparation. You can also find resources for

Preparing Your Article for general guidance about writing and preparing your manuscript at www.wileyauthors.com/eo/prepresources.

REVISION CHECKLIST:

'Potential Cover Art' for consideration as the issue's cover image

- Appropriate Supporting Information (Video, audio or data set: see https://jp.msubmit.net/cgi-bin/main.plex?form_type=display_requirements#supp).

We look forward to receiving your revised submission.

Yours sincerely,

Peying Fong

Senior Editor

The Journal of Physiology

REQUIRED ITEMS

- Author photo and profile. First or joint first authors are asked to provide a short biography (no more than 100 words for one author or 150 words in total for joint first authors) and a portrait photograph. These should be uploaded and clearly labelled together in a Word document with the revised version of the manuscript. See Information for Authors for further details.

- You must start the Methods section with a paragraph headed Ethical approval (https://jp.msubmit.net/cgi-bin/main.plex?form_type=display_requirements#methods).

Research must comply with The Journal's policies regarding animal experiments (<https://physoc.onlinelibrary.wiley.com/hub/animal-experiments>) and adherence to these policies must be stated in the manuscript.

Authors should confirm in their Methods section that their experiments were carried out according to the guidelines laid down by their institution's animal welfare committee, including an ethics approval reference number. The Methods section must contain a statement about access to food, water and housing, details of the anaesthetic regime: anaesthetic used, dose and route of administration, and method of killing the experimental animals.

- The Journal of Physiology funds authors of provisionally accepted papers to use the premium BioRender site to create high resolution schematic figures. Follow this link and enter your details and the manuscript number to create and download figures. Upload these as the figure files for your revised submission. If you choose not to take up this offer, we require figures to be of similar quality and resolution. If you are opting out of this service to authors, state this in the Comments section on the Detailed Information page of the submission form. The link provided should only be used for the purposes of this submission. Authors will be charged for figures created on this premium BioRender account if they are not related to this manuscript submission.

- Please ensure that any tables are editable and in Word format, and wherever possible, embedded in the article file itself.

- Please ensure that the Article File you upload is a Word file.

- A Data Availability Statement is required for all papers reporting original data. This must be in the Additional Information section of the manuscript itself. It must have the paragraph heading 'Data Availability Statement'. All data supporting the results in the paper must be either: in the paper itself; uploaded as Supporting Information for Online Publication; or archived in an appropriate public repository. The statement needs to describe the availability or the absence of shared data. Authors must include in their statement: a link to the repository they have used, or a statement that it is available as Supporting Information; reference the data in the appropriate sections(s) of their manuscript; and cite the data they have shared in the References section. Whenever possible, the scripts and other artefacts used to generate the analyses presented in the paper should also be publicly archived. If sharing data compromises ethical standards or legal requirements then authors are not expected to share it, but must note this in their statement. For more information, see our Statistics Policy.

- Please include an Abstract Figure file, as well as the Figure Legend text within the main article file. The Abstract Figure is a piece of artwork designed to give readers an immediate understanding of the research and should summarise the main conclusions. If possible, the image should be easily 'readable' from left to right or top to bottom. It should show the physiological relevance of the manuscript so readers can assess the importance and content of its findings. Abstract Figures should not merely recapitulate other figures in the manuscript. Please try to keep the diagram as simple as possible and without superfluous information that may distract from the main conclusion(s). Abstract Figures must be provided by authors no later than the revised manuscript stage and should be uploaded as a separate file during online submission labelled as File Type 'Abstract Figure'. Please also ensure that you include the figure legend in the main article file. All Abstract Figures should be created using BioRender. Authors should use The Journal's premium BioRender account to export high-resolution images. Details on how to use and access the premium account are included as part of this email.-----

Note to the Editor and Reviewers: “line #” callouts in blue refer to the ms, revised with track changes “on” and “No Markup” selected, whereas red “line #” callouts refer to the clean ms version. In the proper environment (all files in the same directory identified as the “hyperlink base”), clicking a hyperlink will take one directly to the relevant portion of the ms.

EDITOR COMMENTS

Reviewing Editor:

Two experts in the field have reviewed the manuscript. They each agree that the manuscript has the potential to have a significant impact on the field.

Please ensure the revised manuscript complies with statistical policy https://jp.msubmit.net/cgi-bin/main.plex?form_type=display_requirements#statistics, including reporting of P-values in figures.

Senior Editor (Peying Fong):

Your manuscript, "Synergistic roles of Aquaporin 5 and Intra- and Extracellular Carbonic Anhydrases in promoting CO₂ Diffusion across the Xenopus Oocyte Plasma Membrane" was assessed by two Expert Referees, whose feedback was summarized by the Reviewing Editor. Both Referees expressed enthusiasm about the manuscript's potential to influence the field significantly. Please note that their comments, while uniformly positive, also offered several points for your consideration. Many, if not most, focus on matters of clarity. Referee 2 does highlight points requiring deeper exploration and contemplation, particularly the comments pertaining to figures 5 and 7.

I encourage you to address all comments completely in your revised manuscript.

The extremely thorough statistical analyses performed in your study is commendable. Of note, the resultant information appears not only as tables within the manuscript proper but also are enhanced by supplemental statistical tables. In considering that these supplemental tables are not typical of what the Journal includes as supporting information files for online publication (video and audio files, 3D structures and program codes), it seems that perhaps they might better be incorporated within an Appendix, which can be included within the body of the revised manuscript proper.

Response: Thank you for the good news!

Action: We moved these tables to a new Appendix (see line #1712/1656).

For additional information, please refer to:

https://jp.msubmit.net/cgi-bin/main.plex?form_type=display_requirements#suppinfo.

Thank you for contributing your work to The Journal of Physiology. We look forward to receiving your revised manuscript.

Response: Below we also list our responses to the additional email from Madeleine Hatfield on 8/26/2025 regarding the JP-RP-2025-289145R1

Ethical approval section: Please include the ethics reference number within the main manuscript text. Please also explicitly describe the food, water, housing, anaesthesia (agent, route, dose) and method of killing the experimental animals.

"Housing, anesthesia and euthanasia were as described by Moss & Boron, (2020)" is not sufficient and details must be stated explicitly.

Response: We have addressed this request.

Action: We have written a new Ethical approval section including our approved IACUC protocol number and addressing all of the requested points (see line # 156–170/152-167)

A Data Availability Statement is required for all papers reporting original data.

Response: We have included a Data Availability Statement in the revised manuscript in the Additional information section as requested.

Action: We have added a Data Availability Statement (see line # 1301–1304/1291-1293), stating that “All raw data are deposited into the NIH-supported Zenodo data repository in common and open formats and are accessible via the persistent identifier; DOI: [10.5281/zenodo.16950950](https://doi.org/10.5281/zenodo.16950950)”

In the process of uploading the raw data to Zenodo we noticed several cut-and-paste errors from the analysis files to Appendix Tables A12A iv, v and vi. These have been fixed in the revised manuscript. Furthermore, as a consequence, the P value displayed in Figure 12A has been corrected. This modification did not change the outcome of the analysis for Δ_{Base} vs. Δ_{AQP5} .

REFeree COMMENTS

Referee #1:

This is an outstanding, well-written and comprehensive exploration of the synergistic roles of an aquaporin and intra and extracellular CA activity on the transmembrane diffusion of CO₂ across oocytes. The authors have explored the permutations of adding in the CO₂ channel AQ5, extracellular CA activity and intracellular CA activity using measurements of changes in magnitude and speed in extracellular pH at the membrane and intracellular pH and deduce changes in CO₂ flux rates. They find that all three proteins increase CO₂ flux, but that there is a synergistic effect with the combination of all three. The work and data add nicely to the story of how CO₂ flux across membranes is dependent on the presence of a gas channel protein and both intra and extracellular CA activity. The work is original, the methods are state of the art and the data are novel and compelling. The conclusions are well-supported by the data.

Response: Thank you! We very much appreciate your very positive comments.

I have only minor concerns to raise.

1. Does the addition of small amounts of CA on either side of the membrane influence buffering capacity in the respective compartments to any significant extent?

Response: This is a thoughtful question but not one that is not easy to answer succinctly. One way of analyzing the “chemical buffering” in and around our oocytes is to divide it into (1) CO₂/HCO₃⁻ buffering and (2) non-CO₂/HCO₃⁻ buffering. When changes in [CO₂] produce acid or alkaline loads, the CO₂/HCO₃⁻ buffer system by definition does not participate in the buffering of CO₂-induced pH changes. CA will greatly speed these pH changes, but no buffering takes place in the classic sense of “buffering power.” It is the non-CO₂/HCO₃⁻ buffers that are responsible for limiting the size of CO₂-induced pH changes. These, of course, are not influenced by CAs. So, no, the CAs—either inside or outside the oocyte—do not affect buffering power.

Action: In the Discussion, have added a passage to address this subject briefly (see line # 937–947/929-939).

2. How are the oocytes and electrodes maintained in a secure way when bathing conditions are changed?

Response: Thanks for your concern. Stabilizing the oocyte in the chamber is critical for both pH_s and pH_i measurements. As noted in Methods, (1) We glued X-shaped nylon fibers downstream of the oocyte to hold the cell in place against the flowing solution. (2) We impaled the oocyte with both the V_m and pH_i electrodes after fixing the oocyte against the X-shaped nylon fibers. (3) We adjusted the solution volume in the chamber by a vacuum-driven suction needle to minimize the vibration when solutions are switched between ND96 and 1.5% CO₂/10 mM NaHCO₃.

Action: In Methods (see line # 244–249/241-245) we have attempted to clarify how the solution flowing through the chamber pins the oocyte against X-fibers glued into the channel of the chamber, just downstream from the oocyte. We also cite a trio of Musa-Aziz papers that include illustrative cartoons.

3. Are AQP5 mediated H₂O fluxes also important to the augmentation of CO₂ fluxes?

Response: Thanks for your insightful question—one that we have thought about for years. It is conceivable that the traffic of H₂O molecules through the monomeric pore of AQP5 will affect CO₂ diffusion to the extent that it occurs through the monomeric water pores of AQP5. To test this hypothesis, we could in principle measure the pH_s change under hypertonic (H₂O exiting via AQP5) or hypotonic conditions (H₂O entering), rather than isotonic used in the paper. In fact, we constructed a rudimentary cell-volume clamp that would make it possible to stabilize oocyte volume during an exposure to a hypertonic saline.

4. page 4. I would add also the evidence that some aquaporins are NO gas channels

Response: Thanks for the reminder.

Action: We added references to two papers by Herrera and Garvin to the beginning of the 2nd paragraph of the Introduction (see lines #79–80/76-77).

5. page 37 line 12, please add 'not yet peer reviewed' after 'results, since all three papers are Biorxiv postings. This applies elsewhere any time a Biorxiv post is cited.

Response: We had written “preliminary results”, which could in fact apply to an abstract. The suggested revision is a substantially stronger statement, inasmuch as the 3 *bioRxiv* papers comprise dozens of figures and tables, and >100 pages of text.

Action: We have made the suggested change in the 2nd paragraph of the Discussion (see line # 884/876)

6. page 41, line 23. should the word 'or' be 'on'?

Response: We cannot find the “or” to which the Referee refers. We would be happy to make the correction if the Referee could refer to the line #s in the margin.

Referee #2:

The study on the "Synergistic roles of Aquaporin 5 and Intra- and Extracellular Carbonic Anhydrases in promoting CO₂ Diffusion across the Xenopus Oocyte Plasma Membrane" is a detailed study that analyzes the interaction of catalysts (intracellular and extracellular CA) with the diffusion of CO₂ across cellular membrane in the absence and presence of membrane channels (AQP5 in this case) that also contribute to CO₂ transport. The study is meticulously detailed and executed, and the manuscript is very well written. The study builds up on previous studies by the same group and extends it to include intracellular and extracellular CA and AQP5 in relation to CO₂ transport. The multiple measurements in multiple conditions, although accurately informative, makes it difficult to read. The PI's group is one of few labs that can address such a study with experiments that are demanding and difficult both to do and interpret.

Although most of my concerns are not major, there are several points that need to be addressed as outlined below.

Abstract: the second half of the abstract needs to be rewritten. It is confusing because several terms and interactions are not yet described.

Response: Agreed.

Action: We have revised the entire Abstract, particularly the 2nd half. We removed all reference to "trans" and "cis"—but better explained the concept without using the term "trans". We also deleted " $\Delta\Delta\text{pH}_S$ ". These changes reduced the word count enough for us to add back some words deleted in an earlier draft. The result, we hope, is a more smoothly reading Abstract.

In Methods (P12, 2nd paragraph): The calibration points in ND96 & 1.5% CO₂ seem to be changing upon switching solutions. Is this why a new calibration point is needed every time? If so, what is the difference in measured pHs? Also, is there a difference between pHs in bulk solution & pHs when electrode is touching the membrane? Is there a [real] difference in pHs (in bulk solution) between ND96 (at pH 7.5) and 1.5% CO₂?

Response: Thanks for goading us into action ... these, for us, are standard techniques. However, we understand that they may not be so clear to the general reader ...

First, the "gold standard" for measuring pH in physiological solutions is the Ross electrode (with a glass pH sensor) and a reference electrode with a flowing 3M KCl junction. The pH microelectrode is based on an organic "liquid membrane" in which CO₂ has a finite solubility. (Kaila has exploited this in a double-barreled conformation to measure P_{CO₂}.) Thus, our approach is to use a benchtop pH meter to titrate the pH of our solutions to 7.50 and then use this calibrated solution to "define" pH_o as "7.50" in the chamber, as measured by the pH_S electrode.

Indeed, when the pH_S electrode is in ND96 (pH_o \equiv 7.50) and then switched to CO₂/HCO₃⁻, the apparent value may shift to ~7.52. However, we define the pH_o of the CO₂/HCO₃⁻ solution (previously titrated to 7.50 using the Ross electrode) as 7.50. This small difference in calibration between ND96 and CO₂/HCO₃⁻ is why we typically perform three 1-point calibrations during the experiment.

The pH_S on the membrane surface is nearly always slightly lower than in the bulk ECF, likely because of the efflux of CO₂ produced by the oocyte metabolism. The phenomenon is more

obvious when AQP5 is expressed in the membrane and carbonic anhydrase is added in the perfusing solution (i.e., enhancing the outflow of CO_2 and its conversion to $\text{HCO}_3^- + \text{H}^+$).

Finally, the question of whether there is a real difference between ND96 and $\text{CO}_2/\text{HCO}_3^-$ solutions is difficult to answer (e.g., does the benchtop system have a bias?). We do our best to vigorously equilibrate the $\text{CO}_2/\text{HCO}_3^-$ solution with the CO_2 gas mixture, to store it in syringes with negligible CO_2 permeability, and to deliver it to the chamber via low-permeability tubing.

Action: We made multiple changes to the Methods section, including a description of the benchtop pH meter/electrode in Methods>Physiological Solutions (lines #173–175/170-172), the chamber reference electrode (line #255/251), and improvements to the section Calibration of pH microelectrodes (lines #267-270/263-266).

In Results, we edited the legend of Figure 1C, and added a reference to Methods near line #425/420.

P. 14: Shouldn't the equation read $10^{\text{pHi}-6.8}$ and not $10^{\text{pHi}-7.5}$?

Response: Thanks for pointing out the embarrassing error! “6.8” would have been correct if that were the pK of the $\text{CO}_2/\text{HCO}_3^-$ equilibrium (but that pK is ~6.1).

Action: We revised the equation (line #357/352) to express $[\text{HCO}_3^-]_i$ in terms of $[\text{HCO}_3^-]_o$ and pHi .

Fig 1. Differences between "arrowheads" are difficult to recognize.

Action: We modified the figure to make the differences in arrowheads easier to recognize, and also edited the description in the figure legend.

Fig 2: what is the membrane potential of oocytes in each condition (injected or expressing CA and/or AQP5)? This is one parameter that is indicative of the health status of the oocyte.

Response: Sorry, we were remiss in not listing our criteria for accepting an oocyte as healthy.

V_m is not the “be all and end all” for assessing oocyte health, inasmuch as expression of proteins can alter V_m even in healthy oocytes (e.g., NBCn1, with its intrinsic Na^+ conductance, typically “lowers” V_m to -25 to -30 mV). We use a series of criteria: visual in the initial selection of oocytes, V_m upon impalement, and also membrane stiffness as we push the pH_s electrode up against the membrane.

Action: We explain the above in a new paragraph in Methods under “Expression ...” (lines #215–221/211-218).

In addition, we have added a new bar chart (Appendix Figure A1) in which we summarize final (i.e., near the end of the experiment) V_m values that are from the oocytes used in Fig 3. We introduce this figure on (lines #577-579/571-573) under the new inline heading “ \$V_m\$ ”.

Fig 3 legend: The explanation of the star/bar significance is completely incomprehensible. Please rewrite is a better format easy to understand.

Response: In our statistical analysis (lauded by the Senior Editor), we compared each bar to every other bar, resulting in an extremely large number of comparisons. Rather than explicitly stating, for example, that B is different from A and that A is (obviously) different from B, we stopped after the former, leaving the last half unwritten (and thereby omitting half the statistics symbols).

Action: We have attempted to clarify our nomenclature by thoroughly rewriting the end of the figure legend (see new paragraph titled Fig 3CD conclusions in legend of Fig 3).

P. 23 last 2 lines: Remove "(vs 0ng) until 10 & 100ng"

Response: Sorry, it was garbled.

Action: We reworded the passage under Fig 3CD/conclusions (see lines #562-564/557-558).

P. 25: Use of $(dpH/dt)_{max}$ is not necessarily better than $1/\tau$ as done by Musa-Aziz et al., and Occhipinti et al. If I understand correctly, the conclusion is that CO₂ influx (or efflux) is increased with expression of CAII and remains so during re-equilibration at the surface. This can be determined by a measure of $1/\tau$ or by $(dpH/dt)_{max}$. However, my concern is that exponential decay could vary depending on the parameters mentioned by the author (tip of electrode, location on the membrane etc). This shows a potential source of error.

Response: In the present study, it was impossible to use a single " $1/\tau$ " because the pH_S decays were double-exponential—i.e., they had two time constants. In the case of the Musa-Aziz, with somewhat smaller electrode tips and a somewhat different placement, the decays were nicely fit by single-exponential functions, in which case " $1/\tau$ " was informative. We don't think that this difference reflects an "error" in either the previous or present studies, but a divergence in technique (such as what could easily occur between 2 laboratories) that, in the end (and with technique-appropriate analyses), provides comparable answers. As noted in the original text, mathematical modeling suggests that the pH_S decays should not be perfect exponential functions, anyway. No model is "correct", and the models assume perfect radial/spherical symmetry, which is not true under real-world conditions.

Perhaps we could have focused on the faster of the 2 exponentials, but the effect slower τ would have gone unaccounted-for. Our approach of using $(dpH_S/dt)_{Max}$ has the advantage of capturing the contribution of the slower exponential and is thus less biased.

We are rather proud of our novel analyses in Fig 4, and thought that they would be of value to future investigators.

Action: We heavily edited the paragraph that starts with "In the present study, we note ..." (see lines #600-608/594-601) to reflect our "Response."

Fig 5: Fig 5A shows that $(dpH/dt)_{max}$ increases with increased CAII. However, in Fig. 5B $(dpH/dt)_{max}$ increased significantly only with 100 CAII (after the initial increase compared to CAII). Is it expected that the changes in $(dpH/dt)_{max}$ should mimick Fig 5A?

Response: First, for reasons that we do not fully understand, the removal of CO₂/HCO₃⁻ does not yield results so close to the theoretical as does the addition of CO₂/HCO₃⁻. Part of the issue may be the inherent asymmetry of adding CO₂/HCO₃⁻ from an infinite reservoir as the solution spreads evenly across the "front" of the oocyte vs. removing CO₂/HCO₃⁻ from the finite reservoir of an oocyte that generates eddies in the shadow of the solution flow.

Second, note that although the darker-colored bars in Fig 5A steadily increase in magnitude from left to right, only the dark-red and dark-green bars differ from the black bar (i.e., the dark-blue bar does not differ from the other 3 dark bars). In Fig 5B, it is the dark-red bar that does not differ from the other darker bars. But now the dark-blue bar differs from the dark-green bar. Really, the panels are not so different in their messages.

Fig 6: Same as before, the star/bar significance is very difficult to understand.

Response: OK.

Action: For the legends of Figs 5, 6, 8, 9, and 10 as well as the new Appendix Figure A, we (a) reference the lengthy presentation noted above for Fig 3 and (b) delete most of the rest of the star/bar presentation. In Figs 11 and 12 (special figures from the Discussion), we also reference Fig 3, but leave the rest of the legend intact.

Fig 7: In Fig 7A, there is significant decrease in pHs when the electrode touches the membrane (gray bar). Any explanation for this?

Response: This is probably due to metabolically produced acid ...

Action: Already addressed above (see "Methods, p 12") in comment by Referee #2, just after Abstract.

In Figs 7A & 7B, the authors assert that no significant change in Δ pHs because extracellular CO_2 is rate limiting. This means that if P_{M,CO_2} is increased due to AQP5, it will not affect the rate of CO_2 formation at the surface. Do the authors expect that at a lesser content of external CO_2 (e.g. 0.5% CO_2 rather than 1.5% CO_2), AQP5 expression may cause detectable change in Δ pHs?

Response: We realize that this is a dense paper on a difficult subject ... regarding the first sentence of this statement:

No, we stated that intracellular CA activity is rate limiting. In the "Interpretation" paragraph, see line #688/679. Of course, extracellular CO_2 would not be rate limiting because we have an excess of bCA in the ECF.

Fig 11 and text: using $\Delta\Delta$ pHs for comparisons is a very good tool but the need to use double exponential for fitting is an overkill and probably does not make a difference.

Response: The double-exponential curve fits have nothing to do with this $\Delta\Delta$ pHs analysis, which simply involves assessing differences in spike heights.

The dependence of the measured changes on different levels of CO_2 , in all conditions is very important.

Response: We agree that it would have been nice—as was done by Musa-Aziz—to examine the system over a range of $[\text{CO}_2]_o$ values (they used 3). They varied (a) $[\text{CO}_2]_o$, (b) intracellular CA, (c) and extracellular CA, and (d) $[\text{HEPES}]_o$. We varied (a) intracellular CA, (b) extracellular CA, and (c) AQP5. If we also varied $[\text{CO}_2]_o$ (or $[\text{HEPES}]_o$), the data set would have exploded in size.

Note that we had addressed the $[\text{CO}_2]_o$ matrix in the Discussion as part of Advances and Limitations > Limitations (see lines #1063-1070/1054-1060, especially lines #1064–1066/1054-1056).

Action: We added an explanatory sentence to the Discussion (see lines #1066–1068/1056-1058)

Discussion:

P. 37, paragraph before last: informative but not needed. What is "mobile CA"?

Response: (A) We like the paragraph on O_2 fluxes. Our purpose was not so much to be "informative" but rather to emphasize the broader principle that the "sink" or "source" for an

intracellular gas can be very important. (B) Here, “mobile” means (more or less) freely diffusible (as opposed to a membrane-linked CA, like CA IV).

Action: On line #891/883, we clarified the meaning of “mobile”.

What is expected by varying external CO₂ at constant pH. Please comment.

Response: This comment extends the one comment previously regarding the Musa-Aziz/Occhipinti trio of papers in which they examine 1.5% CO₂/10 mM HCO₃⁻, 5% CO₂/33 mM HCO₃⁻, and 10% CO₂/66 mM HCO₃⁻. This comparison was valuable for Musa-Aziz/Occhipinti because it informed their mathematical model that, in turn, provided insight into the influence of the CAs at different CO₂/HCO₃⁻ combinations (at a fixed pH_o).

Action: As noted above, we added an explanatory sentence to the Discussion (see lines #1066–1068/1056-1058)

What is the logic behind including hCAIV with extracellular bCAII?

Response: We did not use CA IV in the present paper. That was one of the experiments performed by Musa-Aziz and Occhipinti. They added extracellular bCA to oocytes already expressing hCA IV to test the hypothesis that it is advantageous to the CO₂ fluxes if the extracellular CA is not confined to the external surface of the membrane (CA IV), but extended for many μm away from the membrane (the added bCA). The hypothesis was correct! That is, the CA is more effective if it is present not just at the absolute membrane surface but through the unconvected layer.

Action: We further edited the first paragraph of “Limitations” to include the above material.

END OF COMMENTS

The Physiological Society is a company limited by guarantee. Registered in England and Wales, No. 00323575. Registered Office: Hodgkin Huxley House, 30 Farringdon Lane, London, EC1R 3AW, UK. Registered Charity No. 211585. The Physiological Society and The Journal of Physiology are registered trademarks.

This email and any files transmitted with it are confidential and intended solely for the use of the individual or entity to whom they are addressed. If you have received this email in error please notify the sender. If you are not the named addressee you should not disseminate, distribute or copy this e-mail. The Physiological Society may monitor email traffic data.

The Physiological Society has taken reasonable precautions to ensure no viruses are present in this email, however does not accept responsibility for any loss or damage arising from the use of this email or attachments.

Dear Professor Boron,

Re: JP-RP-2025-289145R1 "Synergistic roles of Aquaporin 5 and Intra- and Extracellular Carbonic Anhydrases in promoting CO₂ Diffusion across the *Xenopus* Oocyte Plasma Membrane" by Deng-Ke Wang, Fraser J Moss, and Walter F Boron

We are pleased to tell you that your paper has been accepted for publication in The Journal of Physiology.

Yours sincerely,

Peying Fong
Senior Editor
The Journal of Physiology

If you would like to receive our 'Research Roundup', a monthly newsletter highlighting the cutting-edge research published in The Physiological Society's family of journals (The Journal of Physiology, Experimental Physiology, Physiological Reports, The Journal of Nutritional Physiology and The Journal of Precision Medicine: Health and Disease), please click this link, fill in your name and email address and select 'Research Roundup':
<https://www.physoc.org/journals-and-media/membernews>

- You can help your research get the attention it deserves! Check out Wiley's free Promotion Guide for best-practice recommendations for promoting your work at: www.wileyauthors.com/eeo/guide. You can learn more about Wiley Editing Services which offers professional video, design, and writing services to create shareable video abstracts, infographics, conference posters, lay summaries, and research news stories for your research at: www.wileyauthors.com/eeo/promotion.

EDITOR COMMENTS

Reviewing Editor:

Thank you for your considered and comprehensive response in addressing points raised by two Expert Referees in their initial reviews of your manuscript.

The attached reports indicate their satisfaction with the present, revised version of your manuscript. You will see that their comments are terse, largely because no substantive remaining details require your attention! Well done.

Senior Editor:

Review of your revised manuscript, "Synergistic roles of Aquaporin 5 and Intra- and Extracellular Carbonic Anhydrases in promoting CO₂ Diffusion across the Xenopus Oocyte Plasma Membrane" is now complete. As you will read from their attached reports, no further issues require your attention.

Thank you for your continued contributions to The Journal of Physiology. Congratulations on a well-executed and rigorous study!

REFEREE COMMENTS

Referee #1:

all concerns have been addressed

Referee #2:

All my concerns were answered. Very nice work.